# The best of both worlds: stochastic and adversarial episodic MDPs with unknown transition

**Tiancheng Jin**
University of Southern California
tiancheng.jin@usc.edu

**Longbo Huang**
Tsinghua University
longbohuang@tsinghua.edu.cn

**Haipeng Luo**
University of Southern California
haipengl@usc.edu

## Abstract

We consider the best-of-both-worlds problem for learning an episodic Markov Decision Process through $T$ episodes, with the goal of achieving $\widetilde{\mathcal{O}}(\sqrt{T})$ regret when the losses are adversarial and simultaneously $\mathcal{O}(\text{polylog}(T))$ regret when the losses are (almost) stochastic. Recent work by [Jin and Luo, 2020] achieves this goal when the fixed transition is known, and leaves the case of unknown transition as a major open question. In this work, we resolve this open problem by using the same Follow-the-Regularized-Leader (FTRL) framework together with a set of new techniques. Specifically, we first propose a loss-shifting trick in the FTRL analysis, which greatly simplifies the approach of [Jin and Luo, 2020] and already improves their results for the known transition case. Then, we extend this idea to the unknown transition case and develop a novel analysis which upper bounds the transition estimation error by (a fraction of) the regret itself in the stochastic setting, a key property to ensure $\mathcal{O}(\text{polylog}(T))$ regret.

## 1 Introduction

We study the problem of learning finite-horizon Markov Decision Processes (MDPs) with unknown transition through $T$ episodes. In each episode, the learner starts from a fixed initial state and repeats the following for a fixed number of steps: select an available action, incur some loss, and transit to the next state according to a fixed but unknown transition function. The goal of the learner is to minimize her regret, which is the difference between her total loss and that of the optimal stationary policy in hindsight.

When the losses are stochastically generated, [Simchowitz and Jamieson, 2019, Yang et al., 2021] show that $\mathcal{O}(\log T)$ regret is achievable (ignoring dependence on some gap-dependent quantities for simplicity). On the other hand, even when the losses are adversarially generated, [Rosenberg and Mansour, 2019a, Jin et al., 2020] show that $\widetilde{\mathcal{O}}(\sqrt{T})$ regret is achievable.[1] Given that the existing algorithms for these two worlds are substantially different, Jin and Luo [2020] asked the natural question of whether one can achieve the *best of both worlds*, that is, enjoying (poly)logarithmic regret in the stochastic world while simultaneously ensuring some worst-case robustness in the adversarial world. Taking inspiration from the bandit literature and using the classic Follow-the-regularized-Leader (FTRL) framework with a novel regularizer, they successfully achieved this goal, albeit under a strong restriction that the transition has to be known ahead of time. Since it is highly unclear how

---

[1] Throughout the paper, we use $\widetilde{\mathcal{O}}(\cdot)$ to hide polylogarithmic terms.

35th Conference on Neural Information Processing Systems (NeurIPS 2021).

to ensure that the transition estimation error is only $\mathcal{O}(\text{polylog}(T))$, extending their results to the unknown transition case is highly challenging and was left as a key open question.

In this work, we resolve this open question and propose the first algorithm with such a best-of-both-worlds guarantee under unknown transition. Specifically, our algorithm enjoys $\widetilde{\mathcal{O}}(\sqrt{T})$ regret always, and simultaneously $\mathcal{O}(\log^2 T)$ regret if the losses are i.i.d. samples of a fixed distribution. More generally, our polylogarithmic regret holds under a general condition similar to that of [Jin and Luo, 2020], which requires neither independence nor identical distributions. For example, it covers the corrupted i.i.d. setting where our algorithm achieves $\widetilde{\mathcal{O}}(\sqrt{C})$ regret with $C \leq T$ being the total amount of corruption.

**Techniques**   Our results are achieved via three new techniques. First, we propose a new *loss-shifting* trick for the FTRL analysis when applied to MDPs. While similar ideas have been used for the special case of multi-armed bandits (e.g., [Wei and Luo, 2018, Zimmert and Seldin, 2019, Lee et al., 2020b, Zimmert and Seldin, 2021]), its extension to MDPs has eluded researchers, which is also the reason why [Jin and Luo, 2020] resorts to a different approach with a highly complex analysis involving analyzing the inverse of the non-diagonal Hessian of a complicated regularizer. Instead, inspired by the well-known performance difference lemma, we design a key shifting function in the FTRL analysis, which helps reduce the variance of the stability term and eventually leads to an adaptive bound with a certain self-bounding property known to be useful for the stochastic world. To better illustrate this idea, we use the known transition case as a warm-up example in Section 3, and show that the simple Tsallis entropy regularizer (with a diagonal Hessian) is already enough to achieve the best-of-both-worlds guarantee. This not only greatly simplifies the approach of Jin and Luo [2020] (paving the way for extension to unknown transition), but also leads to bounds with better dependence on some parameters, which on its own is a notable result already.

Our second technique is a new framework to deal with unknown transition under adversarial losses, which is important for incorporating the loss-shifting trick mentioned above. Specifically, when the transition is unknown, prior works [Rosenberg and Mansour, 2019a,b, Jin et al., 2020, Lee et al., 2020a] perform FTRL over the set of all plausible occupancy measures according to a confident set of the true transition, which can be seen as a form of optimism encouraging exploration. Since our loss-shifting trick requires a fixed transition, we propose to move the optimism from the decision set of FTRL to the losses fed to FTRL. More specifically, we perform FTRL over the empirical transition in some doubling epoch schedule, and add (negative) bonuses to the loss functions so that the algorithm is optimistic and never underestimates the quality of a policy, an idea often used in the stochastic setting (e.g., [Azar et al., 2017]). See Section 4 for the details of our algorithm.

Finally, we develop a new analysis to show that the transition estimation error of our algorithm is only polylogarithmic in $T$, overcoming the most critical obstacle in achieving best-of-both-worlds. An important aspect of our analysis is to make use of the amount of underestimation of the optimal policy, a term that is often ignored since it is nonpositive for optimistic algorithms. We do so by proposing a novel decomposition of the regret inspired by the work of Simchowitz and Jamieson [2019], and show that in the stochastic world, every term in this decomposition can be bounded by a fraction of the regret itself plus some polylogarithmic terms, which is enough to conclude the final polylogarithmic regret bound. See Section 5 for a formal summary of this idea.

**Related work**   For earlier results in each of the two worlds, we refer the readers to the systematic surveys in [Simchowitz and Jamieson, 2019, Yang et al., 2021, Jin et al., 2020]. The work closest to ours is [Jin and Luo, 2020] which assumes known transition, and as mentioned, we strictly improve their bounds and more importantly extend their results to the unknown transition case.

Two recent works [Lykouris et al., 2021, Chen et al., 2021] also consider the corrupted stochastic setting, where both the losses and the transition function can be corrupted by a total amount of $C$. This is more general than our results since we assume a fixed transition and only allow the losses to be corrupted. On the other hand, their bounds are worse than ours when specified to our setting — [Lykouris et al., 2021] ensures a gap-dependent polylogarithmic regret bound of $\mathcal{O}(C \log^3 T + C^2)$, while [Chen et al., 2021] achieves $\mathcal{O}(\log^3 T + C)$ but with a potentially larger gap-dependent quantity. Therefore, neither result provides a meaningful guarantee in the adversarial world when $C = T$, while our algorithm always ensures a robustness guarantee with $\widetilde{\mathcal{O}}(\sqrt{T})$ regret. Their algorithms are also very different from ours and are not based on FTRL.

The question of achieving best-of-both-worlds guarantees for the special case of multi-armed bandits was first proposed in [Bubeck and Slivkins, 2012]. Since then, many improvements using different approaches have been established over the years [Seldin and Slivkins, 2014, Auer and Chiang, 2016, Seldin and Lugosi, 2017, Wei and Luo, 2018, Lykouris et al., 2018, Gupta et al., 2019, Zimmert et al., 2019, Zimmert and Seldin, 2021, Lee et al., 2021]. One notable and perhaps surprising approach is to use the FTRL framework, originally designed only for the adversarial settings but later found to be able to automatically adapt to the stochastic settings as long as certain regularizers are applied [Wei and Luo, 2018, Zimmert et al., 2019, Zimmert and Seldin, 2021]. Our approach falls into this category, and our regularizer design is also based on these prior works. As mentioned, however, obtaining our results requires the new loss-shifting technique as well as the novel analysis on controlling the estimation error, both of which are critical to address the extra challenges presented in MDPs.

## 2    Preliminaries

We consider the problem of learning an episodic MDP through $T$ episodes, where the MDP is formally defined by a tuple $(S, A, L, P, \{\ell_t\}_{t=1}^T)$ with $S$ being a finite state set, $A$ being a finite action set, $L$ being the horizon, $\ell_t : S \times A \to [0, 1]$ being the loss function of episode $t$, and $P : S \times A \times S \to [0, 1]$ being the transition function so that $P(s'|s, a)$ is the probability of moving to state $s'$ after executing action $a$ at state $s$.

Without loss of generality [Jin et al., 2020], the MDP is assumed to have a layer structure, that is, the state set $S$ is partitioned into $L + 1$ subsets $S_0, S_1, \ldots, S_L$ such that the state transition is only possible from one layer to the next layer (in other words, $P(s'|s, a)$ must be zero unless $s \in S_k$ and $s' \in S_{k+1}$ for some $k \in \{0, \ldots, L-1\}$). Moreover, $S_0$ contains $s_0$ only (the initial state), and $S_L$ contains $s_L$ only (the terminal state). We use $k(s)$ to represent the layer to which state $s$ belongs.

Ahead of time, the environment decides an MDP with $P$ and $\{\ell_t\}_{t=1}^T$ unknown to the learner. The interaction proceeds through $T$ episodes. In episode $t$, the learner selects a stochastic policy $\pi_t : S \times A \to [0, 1]$ where $\pi_t(a|s)$ denotes the probability of taking action $a$ at state $s$.[2] Starting from the initial state $s_0^t = s_0$, the learner then repeatedly selects an action $a_k^t$ drawn from $\pi_t(\cdot|s_k^t)$, suffers loss $\ell_t(s_k^t, a_k^t)$, and transits to the next state $s_{k+1}^t \in S_{k+1}$ for $k = 0, \ldots, L-1$, until reaching the terminal state $s_L$. At the end of the episode, the learner receives some feedback on the loss function $\ell_t$. In the *full-information* setting, the learner observes the entire loss function $\ell_t$, while in the more challenging *bandit feedback* setting, the learner only observes the losses of those visited state-action pairs, that is, $\ell_t(s_0^t, a_0^t), \ldots, \ell_t(s_{L-1}^t, a_{L-1}^t)$.

With slight abuse of notation, we denote the expected loss of a policy $\pi$ for episode $t$ by $\ell_t(\pi) = \mathbb{E}\left[\sum_{k=0}^{L-1} \ell_t(s_k, a_k) \,\middle|\, P, \pi\right]$, where the trajectory $\{(s_k, a_k)\}_{k=0,\ldots,L-1}$ is the generated by executing policy $\pi$ under transition $P$. The regret of the learner against some policy $\pi$ is then defined as $\text{Reg}_T(\pi) = \mathbb{E}\left[\sum_{t=1}^T \ell_t(\pi_t) - \ell_t(\pi)\right]$, and we denote by $\mathring{\pi}$ one of the optimal policies in hindsight such that $\text{Reg}_T(\mathring{\pi}) = \max_\pi \text{Reg}_T(\pi)$.

**Adversarial world versus stochastic world**   We consider two different setups depending on how the loss functions $\ell_1, \ldots, \ell_T$ are generated. In the adversarial world, the environment decides the loss functions arbitrarily with knowledge of the learner's algorithm (but not her randomness). In this case, the goal is to minimize the regret against the best policy $\text{Reg}_T(\mathring{\pi})$, with the best existing upper bound being $\widetilde{\mathcal{O}}(L|S|\sqrt{|A|T})$ [Rosenberg and Mansour, 2019a, Jin et al., 2020] and the best lower bound being $\Omega(L\sqrt{|S||A|T})$ [Jin et al., 2018] (for both full-information and bandit feedback).

In the stochastic world, following [Jin and Luo, 2020] (which generalizes the bandit case of [Zimmert and Seldin, 2019, 2021]), we assume that the loss functions satisfy the following condition: there exists a deterministic policy $\pi^\star : S \to A$, a gap function $\Delta : S \times A \to \mathbb{R}_+$ and a constant $C > 0$ such that

$$\text{Reg}_T(\pi^\star) \geq \mathbb{E}\left[\sum_{t=1}^T \sum_{s \neq s_L} \sum_{a \neq \pi^\star(s)} q_t(s, a)\Delta(s, a)\right] - C, \tag{1}$$

---

[2]Note that $\pi_t(\cdot|s_L)$ is not meaningful since no action will be taken at $s_L$. For conciseness, however, we usually define functions over $S \times A$ instead of $(S \setminus \{s_L\}) \times A$.

where $q_t(s,a)$ is the probability of the learner visiting $(s,a)$ in episode $t$. This general condition covers the heavily-studied i.i.d. setting where $\ell_1, \ldots, \ell_T$ are i.i.d. samples of a fixed distribution, in which case $C = 0$, $\pi^\star$ is simply the optimal policy, and $\Delta$ is the gap function with respect to the optimal $Q$-function. More generally, the condition also covers the corrupted i.i.d. setting with $C$ being the total amount of corruption. We refer the readers to [Jin and Luo, 2020] for detailed explanation. In this stochastic world, our goal is to minimize regret against $\pi^\star$, that is, $\text{Reg}_T(\pi^\star)$.[3] With unknown transition, this general setup has not been studied before, but for specific examples such as the i.i.d. setting, regret bounds of order $\mathcal{O}(\frac{\log T}{\Delta_{\text{MIN}}})$ where $\Delta_{\text{MIN}} = \min_{s, a \neq \pi^\star(s)} \Delta(s,a)$ have been derived [Simchowitz and Jamieson, 2019, Yang et al., 2021].

**Occupancy measure and FTRL**   To solve this problem with online learning techniques, a commonly used concept is the occupancy measure. Specifically, an occupancy measure $q^{\bar{P},\pi} : S \times A \rightarrow [0,1]$ associated with a policy $\pi$ and a transition function $\bar{P}$ is such that $q^{\bar{P},\pi}(s,a)$ equals the probability of visiting state-action pair $(s,a)$ under the given policy $\pi$ and transition $\bar{P}$. Our earlier notation $q_t$ in Eq. (1) is thus simply a shorthand for $q^{P,\pi_t}$. Moreover, by definition, $\ell_t(\pi)$ can be rewritten as $\langle q^{P,\pi}, \ell_t \rangle$ by naturally treating $q^{P,\pi}$ and $\ell_t$ as vectors in $\mathbb{R}^{|S| \times |A|}$, and thus the regret $\text{Reg}_T(\pi)$ can be written as $\mathbb{E}\left[ \sum_{t=1}^T \langle q_t - q^{P,\pi}, \ell_t \rangle \right]$, connecting the problem to online linear optimization.

Given a transition function $\bar{P}$, we denote by $\Omega(\bar{P}) = \{ q^{\bar{P},\pi} : \pi \text{ is a stochastic policy} \}$ the set of all valid occupancy measures associated with the transition $\bar{P}$. It is known that $\Omega(\bar{P})$ is a simple polytope with $\mathcal{O}(|S||A|)$ constraints [Zimin and Neu, 2013]. When $P$ is unknown, our algorithm uses an estimated transition $\bar{P}$ as a proxy and searches for a "good" occupancy measure within $\Omega(\bar{P})$. More specifically, this is done by the classic Follow-the-Regularized-Leader (FTRL) framework which solves the following at the beginning of episode $t$:

$$\widehat{q}_t = \operatorname*{argmin}_{q \in \Omega(\bar{P})} \left\langle q, \sum_{\tau < t} \widehat{\ell}_\tau \right\rangle + \phi_t(q), \tag{2}$$

where $\widehat{\ell}_\tau$ is some estimator for $\ell_\tau$ and $\phi_t$ is some regularizer. The learner's policy $\pi_t$ is then defined through $\pi_t(a|s) \propto \widehat{q}_t(s,a)$. Note that we have $\widehat{q}_t = q^{\bar{P},\pi_t}$ but not necessarily $\widehat{q}_t = q_t$ unless $\bar{P} = P$.

## 3   Warm-up for Known Transition: A New Loss-shifting Technique

One of the key components of our approach is a new loss-shifting technique for analyzing FTRL applied to MDPs. To illustrate the key idea in a clean manner, in this section we focus on the known transition setting with bandit feedback, the same setting studied by Jin and Luo [2020]. As we will show, our method not only improves their bounds, but also significantly simplifies the analysis, which paves the way for extending the result to the unknown transition setting studied in following sections.

First note that when $P$ is known, one can simply take $\bar{P} = P$ (so that $\widehat{q}_t = q_t$) and use the standard importance-weighted estimator $\widehat{\ell}_\tau(s,a) = \ell_\tau(s,a)\mathbb{I}_\tau(s,a)/q_\tau(s,a)$ in the FTRL framework Eq. (2), where $\mathbb{I}_\tau(s,a)$ is 1 if $(s,a)$ is visited in episode $\tau$, and 0 otherwise. It remains to determine the regularizer $\phi_t$. While there are many choices of $\phi_t$ leading to $\sqrt{T}$-regret in the adversarial world, obtaining logarithmic regret in the stochastic world requires some special property of the regularizer. Specifically, generalizing the idea of [Zimmert and Seldin, 2019] for multi-armed bandits, [Jin and Luo, 2020] shows that it suffices to find $\phi_t$ such that the following adaptive regret bound holds

$$\text{Reg}_T(\mathring{\pi}) \lesssim \mathbb{E}\left[ \sum_{t=1}^T \sum_{s \neq s_L} \sum_{a \neq \pi^\star(s)} \sqrt{\frac{q_t(s,a)}{t}} \right], \tag{3}$$

which then automatically implies logarithmic regret under Eq. (1). This is because Eq. (3) admits a self-bounding property under Eq. (1) — one can bound the right-hand side of Eq. (3) as follows using

---

[3]Some works (such as [Jin and Luo, 2020]) still consider minimizing $\text{Reg}_T(\mathring{\pi})$ as the goal in this case. More discussions are deferred to the last paragraph of Section 4.1.

AM-GM inequality (for any $z > 0$), which can then be related to the regret itself using Eq. (1):

$$\mathbb{E}\left[\sum_{t=1}^{T}\sum_{s\neq s_L}\sum_{a\neq\pi^\star(s)}\frac{q_t(s,a)\Delta(s,a)}{2z} + \frac{z}{2t\Delta(s,a)}\right] \leq \frac{\text{Reg}_T(\mathring{\pi}) + C}{2z} + z\sum_{s\neq s_L}\sum_{a\neq\pi^\star(s)}\frac{\log T}{\Delta(s,a)}.$$
(4)

Rearranging and picking the optimal $z$ then shows a logarithmic bound for $\text{Reg}_T(\mathring{\pi})$ (see Section 2 of Jin and Luo [2020] for detailed discussions).

To achieve Eq. (3), a natural candidate of $\phi_t$ would be a direct generalization of the Tsallis-entropy regularizer of [Zimmert and Seldin, 2019], which takes the form $\phi_t(q) = -\frac{1}{\eta_t}\sum_{s,a}\sqrt{q(s,a)}$ with $\eta_t = 1/\sqrt{t}$. However, Jin and Luo [2020] argued that it is highly unclear how to achieve Eq. (3) with this natural candidate, and instead, inspired by [Zimmert et al., 2019] they ended up using a different regularizer with a complicated non-diagonal Hessian to achieve Eq. (3), which makes the analysis extremely complex since it requires analyzing the inverse of this non-diagonal Hessian.

Our first key contribution is to show that this natural and simple candidate is in fact (almost) enough to achieve Eq. (3) after all. To show this, we propose a new a loss-shifting technique in the analysis. Similar techniques have been used for multi-armed bandits, but the extension to MDPs is much less clear. Specifically, observe that for any *shifting function* $g_\tau : S \times A \to \mathbb{R}$ such that the value of $\langle q, g_\tau\rangle$ is independent of $q$ for any $q \in \Omega(\bar{P})$, we have

$$\widehat{q}_t = \operatorname*{argmin}_{q\in\Omega(\bar{P})}\left\langle q, \sum_{\tau<t}\widehat{\ell}_\tau\right\rangle + \phi_t(q) = \operatorname*{argmin}_{q\in\Omega(\bar{P})}\left\langle q, \sum_{\tau<t}(\widehat{\ell}_\tau + g_\tau)\right\rangle + \phi_t(q).$$
(5)

Therefore, we can pretend that the learner is performing FTRL over the shifted loss sequence $\{\widehat{\ell}_\tau + g_\tau\}_{\tau<t}$ (even when $g_\tau$ is unknown to the learner). The advantage of analyzing FTRL over this shifted loss sequence is usually that it helps reduce the variance of the loss functions.

For multi-armed bandits, prior works [Wei and Luo, 2018, Zimmert and Seldin, 2019] pick $g_\tau$ to be a constant such as the negative loss of the learner in episode $\tau$. For MDPs, however, this is not enough to show Eq. (3), as already pointed out by Jin and Luo [2020] (which is also the reason why they resorted to a different approach). Instead, we propose the following shifting function:

$$g_\tau(s,a) = \widehat{Q}_\tau(s,a) - \widehat{V}_\tau(s) - \widehat{\ell}_\tau(s,a), \quad \forall(s,a) \in S \times A,$$
(6)

where $\widehat{Q}_\tau$ and $\widehat{V}_\tau$ are the state-action and state value functions with respect to the transition $\bar{P}$, the loss function $\widehat{\ell}_\tau$, and the policy $\pi_\tau$, that is: $\widehat{Q}_\tau(s,a) = \widehat{\ell}_\tau(s,a) + \mathbb{E}_{s'\sim\bar{P}(\cdot|s,a)}[\widehat{V}_\tau(s')]$ and $\widehat{V}_\tau(s) = \mathbb{E}_{a\sim\pi_\tau(\cdot|s)}[\widehat{Q}_\tau(s,a)]$ (with $\widehat{V}_\tau(s_L) = 0$). This indeed satisfies the invariant condition since using a well-known performance difference lemma one can show $\langle q, g_\tau\rangle = -\widehat{V}_\tau(s_0)$ for any $q \in \Omega(\bar{P})$ (Lemma A.1.1). With this shifting function, the learner is equivalently running FTRL over the "advantage" functions ($\widehat{Q}_\tau(s,a) - \widehat{V}_\tau(s)$ is often called the advantage at $(s,a)$ in the literature).

More importantly, it turns out that when seeing FTRL in this way, a standard analysis with some direct calculation already shows Eq. (3). One caveat is that since $\widehat{Q}_\tau(s,a) - \widehat{V}_\tau(s)$ can potentially have a large magnitude, we also need to stabilize the algorithm by adding a small amount of the so-called log-barrier regularizer to the Tsallis entropy regularizer, an idea that has appeared in several prior works (see [Jin and Luo, 2020] and references therein). We defer all details including the concrete algorithm and analysis to Appendix A, and show the final results below.

**Theorem 3.1.** *When $P$ is known, Algorithm 3 (with parameter $\gamma = 1$) ensures the optimal regret $\text{Reg}_T(\mathring{\pi}) = \mathcal{O}(\sqrt{L|S||A|T})$ in the adversarial world, and simultaneously $\text{Reg}_T(\pi^\star) \leq \text{Reg}_T(\mathring{\pi}) = \mathcal{O}(U + \sqrt{UC})$ where $U = \frac{L|S|\log T}{\Delta_{\text{MIN}}} + L^4\sum_{s\neq s_L}\sum_{a\neq\pi^\star(s)}\frac{\log T}{\Delta(s,a)}$ in the stochastic world.*

Our bound for the stochastic world is even better than [Jin and Luo, 2020] (their $U$ has an extra $|A|$ factor in the first term and an extra $L$ factor in the second term). By setting the parameter $\gamma$ differently, one can also improve $L^4$ to $L^3$, matching the best existing result from [Simchowitz and Jamieson, 2019] for the i.i.d. setting with $C = 0$ (this would worsen the adversarial bound though). Besides this improvement, we emphasize again that the most important achievement of this approach is that it significantly simplifies the analysis, making the extension to the unknown transition setting possible.

# 4 Main Algorithms and Results

We are now ready to introduce our main algorithms and results for the unknown transition case, with either full-information or bandit feedback. The complete pseudocode is shown in Algorithm 1, which is built with two main components: a new framework to deal with unknown transitions and adversarial losses (important for incorporating our loss-shifting technique), and special regularizers for FTRL. We explain these two components in detail below.

**A new framework for unknown transitions and adversarial losses** When the transition is unknown, a common practice (which we also follow) is to maintain an empirical transition along with a shrinking confidence set of the true transition, usually updated in some doubling epoch schedule. More specifically, a new epoch is started whenever the total number of visits to some state-action pair is doubled (compared to the beginning of this epoch), thus resulting in at most $\mathcal{O}(|S||A|\log T)$ epochs. We denote by $i(t)$ the epoch index to which episode $t$ belongs. At the beginning of each epoch $i$, we calculate the empirical transition $\bar{P}_i$ (fixed through this epoch) as:

$$\bar{P}_i(s'|s,a) = \frac{m_i(s,a,s')}{m_i(s,a)}, \quad \forall (s,a,s') \in S_k \times A \times S_{k+1},\ k = 0, \ldots L-1, \tag{7}$$

where $m_i(s,a)$ and $m_i(s,a,s')$ are the total number of visits to $(s,a)$ and $(s,a,s')$ respectively prior to epoch $i$.[4] The confidence set of the true transition for this epoch is then defined as

$$\mathcal{P}_i = \left\{ \widehat{P} : \left| \widehat{P}(s'|s,a) - \bar{P}_i(s'|s,a) \right| \leq B_i(s,a,s'),\ \forall (s,a,s') \in S_k \times A \times S_{k+1}, k < L \right\},$$

where $B_i$ is Bernstein-style confidence width (taken from Jin et al. [2020]):

$$B_i(s,a,s') = \min \left\{ 2\sqrt{\frac{\bar{P}_i(s'|s,a)\ln\left(\frac{T|S||A|}{\delta}\right)}{m_i(s,a)}} + \frac{14\ln\left(\frac{T|S||A|}{\delta}\right)}{3m_i(s,a)}, 1 \right\} \tag{8}$$

for some confidence parameter $\delta \in (0,1)$. As [Jin et al., 2020, Lemma 2] shows, the true transition $P$ is contained in the confidence set $\mathcal{P}_i$ for all epoch $i$ with probably at least $1-4\delta$.

When dealing with adversarial losses, prior works [Rosenberg and Mansour, 2019a,b, Jin et al., 2020, Lee et al., 2020a] perform FTRL (or a similar algorithm called Online Mirror Descent) over the set of all plausible occupancy measures $\Omega(\mathcal{P}_i) = \{q \in \Omega(\widehat{P}) : \widehat{P} \in \mathcal{P}_i\}$ during epoch $i$, which can be seen as a form of optimism and encourages exploration. This framework, however, does not allow us to apply the loss-shifting trick discussed in Section 3 — indeed, our key shifting function Eq. (6) is defined in terms of some fixed transition $\bar{P}$, and the required invariant condition on $\langle q, g_\tau \rangle$ only holds for $q \in \Omega(\bar{P})$ but not $q \in \Omega(\mathcal{P}_i)$.

Inspired by this observation, we propose the following new approach. First, to directly fix the issue mentioned above, for each epoch $i$, we run a new instance of FTRL simply over $\Omega(\bar{P}_i)$. This is implemented by keeping track of the epoch starting time $t_i$ and only using the cumulative loss $\sum_{\tau=t_i}^{t-1} \widehat{\ell}_\tau$ in the FTRL update (Eq. (10)). Therefore, in each epoch, we are pretending to deal with a known transition problem, making the same loss-shifting technique discussed in Section 3 applicable.

However, this removes the critical optimism in the algorithm and does not admit enough exploration. To fix this, our second modification is to feed FTRL with optimistic losses constructed by adding some (negative) bonus term, an idea often used in the stochastic setting. More specifically, we subtract $L \cdot B_i(s,a)$ from the loss for each $(s,a)$ pair, where $B_i(s,a) = \min\left\{1, \sum_{s' \in S_{k(s)+1}} B_i(s,a,s')\right\}$; see Eq. (11). In the full-information setting, this means using $\widehat{\ell}_t(s,a) = \ell_t(s,a) - L \cdot B_i(s,a)$. In the bandit setting, note that the importance-weighted estimator discussed in Section 3 is no longer applicable since the transition is unknown (making $q_t$ also unknown), and [Jin et al., 2020] proposes to use $\frac{\ell_t(s,a) \cdot \mathbb{I}_t(s,a)}{u_t(s,a)}$ instead, where $\mathbb{I}_t(s,a)$ is again the indicator of whether $(s,a)$ is visited during episode $t$, and $u_t(s,a)$ is the so-called upper occupancy measure defined as

$$u_t(s,a) = \max_{\widehat{P} \in \mathcal{P}_{i(t)}} q^{\widehat{P},\pi_t}(s,a) \tag{9}$$

---

[4]When $m_i(s,a) = 0$, we simply let $\bar{P}_i(\cdot|s,a)$ be an arbitrary distribution.

---

**Algorithm 1** Best-of-both-worlds for Episodic MDPs with Unknown Transition

---

**Input:** confidence parameter $\delta$.

**Initialize:** epoch index $i = 1$ and epoch starting time $t_i = 1$.

**Initialize:** $\forall (s, a, s')$, set counters $m_1(s, a) = m_1(s, a, s') = m_0(s, a) = m_0(s, a, s') = 0$.

**Initialize:** empirical transition $\bar{P}_1$ and confidence width $B_1$ based on Eq. (7) and Eq. (8).

**for** $t = 1, \ldots, T$ **do**

Let $\phi_t$ be Eq. (13) for full-information feedback or Eq. (12) for bandit feedback, and compute

$$\widehat{q}_t = \underset{q \in \Omega(\bar{P}_i)}{\mathrm{argmin}} \left\langle q, \sum_{\tau = t_i}^{t-1} \widehat{\ell}_\tau \right\rangle + \phi_t(q). \tag{10}$$

Compute policy $\pi_t$ from $\widehat{q}_t$ such that $\pi_t(a|s) \propto \widehat{q}_t(s, a)$.[5]

Execute policy $\pi_t$ and obtain trajectory $(s_k^t, a_k^t)$ for $k = 0, \ldots, L - 1$.

Construct adjusted loss estimator $\widehat{\ell}_t$ such that

$$\widehat{\ell}_t(s, a) = \begin{cases} \ell_t(s, a) - L \cdot B_i(s, a), & \text{for full-information feedback,} \\ \frac{\ell_t(s, a) \cdot \mathbb{I}_t(s, a)}{u_t(s, a)} - L \cdot B_i(s, a), & \text{for bandit feedback,} \end{cases} \tag{11}$$

where $B_i(s, a) = \min \left\{ 1, \sum_{s' \in S_{k(s)+1}} B_i(s, a, s') \right\}$, $\mathbb{I}_t(s, a) = \mathbb{I}\{\exists k, (s, a) = (s_k^t, a_k^t)\}$, and $u_t$ is the upper occupancy measure defined in Eq. (9).

Increment counters: for each $k < L$, $m_i(s_k^t, a_k^t, s_{k+1}^t) \overset{+}{\leftarrow} 1$, $m_i(s_k^t, a_k^t) \overset{+}{\leftarrow} 1$.[6]

**if** $\exists k, \ m_i(s_k^t, a_k^t) \geq \max\{1, 2m_{i-1}(s_k^t, a_k^t)\}$ **then**                    ▷ entering a new epoch

Increment epoch index $i \overset{+}{\leftarrow} 1$ and set new epoch starting time $t_i = t + 1$.

Initialize new counters: $\forall (s, a, s'), m_i(s, a, s') = m_{i-1}(s, a, s'), m_i(s, a) = m_{i-1}(s, a)$.

Update empirical transition $\bar{P}_i$ and confidence width $B_i$ based on Eq. (7) and Eq. (8).

---

and can be efficiently computed via the COMP-UOB procedure of [Jin et al., 2020]. Our final adjusted loss estimator is then $\widehat{\ell}_t(s, a) = \frac{\ell_t(s, a) \cdot \mathbb{I}_t(s, a)}{u_t(s, a)} - L \cdot B_i(s, a)$. In our analysis, we show that these adjusted loss estimators indeed make sure that we only underestimate the loss of each policy, which encourages exploration.

With this new framework, it is not difficult to show $\sqrt{T}$-regret in the adversarial world using many standard choices of the regularizer $\phi_t$ (which recovers the results of [Rosenberg and Mansour, 2019a, Jin et al., 2020] with a different approach). To further ensure polylogarithmic regret in the stochastic world, however, we need some carefully designed regularizers discussed next.

**Special regularizers for FTRL**  Due to the new structure of our algorithm which uses a fixed transition $\bar{P}_i$ during epoch $i$, the design of the regularizers is basically the same as in the known transition case. Specifically, in the bandit case, we use the same Tsallis entropy regularizer:

$$\phi_t(q) = -\frac{1}{\eta_t} \sum_{s \neq s_L} \sum_{a \in A} \sqrt{q(s, a)} + \beta \sum_{s \neq s_L} \sum_{a \in A} \ln \frac{1}{q(s, a)}, \tag{12}$$

where $\eta_t = 1/\sqrt{t - t_{i(t)} + 1}$ and $\beta = 128L^4$. As discussed in Section 3, the small amount of log-barrier in the second part of Eq. (12) is used to stabilize the algorithm, similarly to [Jin and Luo, 2020].

In the full-information case, while we can still use Eq. (12) since the bandit setting is only more difficult, this leads to extra dependence on some parameters. Instead, we use the following Shannon entropy regularizer:

$$\phi_t(q) = \frac{1}{\eta_t} \sum_{s \neq s_L} \sum_{a \in A} q(s, a) \cdot \ln q(s, a). \tag{13}$$

---

[5]If $\sum_{b \in A} \widehat{q}_t(s, b) = 0$, we let $\pi_t$ be the uniform distribution.

[6]We use $x \overset{+}{\leftarrow} y$ as a shorthand for the increment operation $x \leftarrow x + y$.

Although this is a standard choice for the full-information setting, the tuning of the learning rate $\eta_t$ requires some careful thoughts. In the special case of MDPs with one layer (known as the expert problem [Freund and Schapire, 1997]), it has been shown that choosing $\eta_t$ to be of order $1/\sqrt{t}$ ensures best-of-both-worlds [Mourtada and Gaïffas, 2019, Amir et al., 2020]. However, in our general case, due to the use of the loss-shifting trick, we need to use the following data-dependent tuning (with $i$ denoting $i(t)$ for simplicity): $\eta_t = \sqrt{\frac{L\ln(|S||A|)}{64L^5\ln(|S||A|)+M_t}}$ where

$$M_t = \sum_{\tau=t_i}^{t-1} \min\left\{ \sum_{s\neq s_L}\sum_{a\in A} \widehat{q}_\tau(s,a)\widehat{\ell}_\tau(s,a)^2, \sum_{s\neq s_L}\sum_{a\in A}\widehat{q}_\tau(s,a)\left(\widehat{Q}_\tau(s,a) - \widehat{V}_\tau(s)\right)^2 \right\},$$

and similar to the discussion in Section 3, $\widehat{Q}_\tau$ and $\widehat{V}_\tau$ are the state-action and state value functions with respect to the transition $\bar{P}_i$, the adjusted loss function $\widehat{\ell}_\tau$, and the policy $\pi_\tau$, that is: $\widehat{Q}_\tau(s,a) = \widehat{\ell}_\tau(s,a)+\mathbb{E}_{s'\sim\bar{P}_i(\cdot|s,a)}[\widehat{V}_\tau(s')]$ and $\widehat{V}_\tau(s) = \mathbb{E}_{a\sim\pi_\tau(\cdot|s)}[\widehat{Q}_\tau(s,a)]$ (with $\widehat{V}_\tau(s_L) = 0$). This particular tuning makes sure that FTRL enjoys some adaptive regret bound with a self-bounding property akin to Eq. (3), which is again the key to ensure polylogarithmic regret in the stochastic world. This concludes all the algorithm design; see Algorithm 1 again for the complete pseudocode.

## 4.1 Main Best-of-both-worlds Results

We now present our main best-of-both-worlds results. As mentioned, proving $\sqrt{T}$-regret in the adversarial world is relatively straightforward. However, proving polylogarithmic regret bounds for the stochastic world is much more challenging due to the transition estimation error, which is usually of order $\sqrt{T}$. Fortunately, we are able to develop a new analysis that upper bounds some transition estimation related terms by the regret itself, establishing a self-bounding property again. We defer the proof sketch to Section 5, and state the main results in the following theorems.[7]

**Theorem 4.1.1.** *In the full-information setting, Algorithm 1 with $\delta = \frac{1}{T^2}$ guarantees $\mathrm{Reg}_T(\mathring{\pi}) = \widetilde{\mathcal{O}}\left(L|S|\sqrt{|A|T}\right)$ always, and simultaneously $\mathrm{Reg}_T(\pi^\star) = \mathcal{O}\left(U + \sqrt{UC}\right)$ under Condition (1), where $U = \mathcal{O}\left(\frac{\left(L^6|S|^2+L^5|S||A|\log(|S||A|)\right)\log T}{\Delta_{\mathrm{MIN}}} + \sum_{s\neq s_L}\sum_{a\neq\pi^\star(s)}\frac{L^6|S|\log T}{\Delta(s,a)}\right)$.*

**Theorem 4.1.2.** *In the bandit feedback setting, Algorithm 1 with $\delta = \frac{1}{T^3}$ guarantees $\mathrm{Reg}_T(\mathring{\pi}) = \widetilde{\mathcal{O}}\left((L + \sqrt{|A|})|S|\sqrt{|A|T}\right)$ always, and simultaneously $\mathrm{Reg}_T(\pi^\star) = \mathcal{O}\left(U + \sqrt{UC}\right)$ under Condition (1), where $U = \mathcal{O}\left(\frac{\left(L^6|S|^2+L^3|S|^2|A|\right)\log^2 T}{\Delta_{\mathrm{MIN}}} + \sum_{s\neq s_L}\sum_{a\neq\pi^\star(s)}\frac{\left(L^6|S|+L^4|S||A|\right)\log^2 T}{\Delta(s,a)}\right)$.*

While our bounds have some extra dependence on the parameters $L$, $|S|$, and $|A|$ compared to the best existing bounds in each of the two worlds, we emphasize that our algorithm is the first to be able to adapt to these two worlds simultaneously and achieve $\widetilde{\mathcal{O}}(\sqrt{T})$ and $\mathcal{O}(\mathrm{polylog}(T))$ regret respectively. In fact, with some extra twists (such as treating differently the state-action pairs that are visited often enough and those that are not), we can improve the dependence on these parameters, but we omit these details since they make the algorithms much more complicated.

Also, while [Jin and Luo, 2020] is able to obtain $\mathcal{O}(\log T)$ regret for the stronger benchmark $\mathrm{Reg}_T(\mathring{\pi})$ under Condition (1) and known transition (same as our Theorem 3.1), here we only achieve so for $\mathrm{Reg}_T(\pi^\star)$ due to some technical difficulty (see Section 5). However, recall that for the most interesting i.i.d. case, one simply has $\mathrm{Reg}_T(\pi^\star) = \mathrm{Reg}_T(\mathring{\pi})$ as discussed in Section 2; even for the corrupted i.i.d. case, since $\mathrm{Reg}_T(\mathring{\pi})$ is at most $C + \mathrm{Reg}_T(\pi^\star)$, our algorithms ensure $\mathrm{Reg}_T(\mathring{\pi}) = \mathcal{O}(U + C)$ (note $\sqrt{UC} \leq U + C$). Therefore, our bounds on $\mathrm{Reg}_T(\pi^\star)$ are meaningful and strong.

## 5 Analysis Sketch

In this section, we provide a proof sketch for the full-information setting (which is simpler but enough to illustrate our key ideas). The complete proofs can be found in Appendix B (full-information) and

---

[7]For simplicity, for bounds in the stochastic world, we omit some $\widetilde{\mathcal{O}}(1)$ terms that are independent of the gap function, but they can be found in the full proof.

[Appendix C](#) (bandit). We start with the following straightforward regret decomposition:

$$\text{Reg}_T(\pi) = \mathbb{E}\left[\underbrace{\sum_{t=1}^{T} V_t^{\pi_t}(s_0) - \widehat{V}_t^{\pi_t}(s_0)}_{\text{ERR}_1} + \underbrace{\sum_{t=1}^{T} \widehat{V}_t^{\pi_t}(s_0) - \widehat{V}_t^{\pi}(s_0)}_{\text{ESTREG}} + \underbrace{\sum_{t=1}^{T} \widehat{V}_t^{\pi}(s_0) - V_t^{\pi}(s_0)}_{\text{ERR}_2}\right] \quad (14)$$

for an arbitrary benchmark $\pi$, where $V_t^\pi$ is the state value function associated with the true transition $P$, the true loss $\ell_t$, and policy $\pi$, while $\widehat{V}_t^\pi$ is the state value function associated with the empirical transition $\bar{P}_{i(t)}$, the adjusted loss $\widehat{\ell}_t$, and policy $\pi$. Define the corresponding state-action value functions $Q_t^\pi$ and $\widehat{Q}_t^\pi$ similarly (our earlier notations $\widehat{V}_t$ and $\widehat{Q}_t$ are thus shorthands for $\widehat{V}_t^{\pi_t}$ and $\widehat{Q}_t^{\pi_t}$).

In the adversarial world, we bound each of the three terms in [Eq. (14)](#) as follows (see [Proposition B.1](#) for details). First, $\mathbb{E}[\text{ERR}_1]$ measures the estimation error of the loss of the learner's policy $\pi_t$, which can be bounded by $\widetilde{\mathcal{O}}(L|S|\sqrt{|A|T})$ following the analysis of [Jin et al. [2020]](#). Second, as mentioned, our adjusted losses are optimistic in the sense that it underestimates the loss of all policies (with high probability), making $\mathbb{E}[\text{ERR}_2]$ an $\mathcal{O}(1)$ term only. Finally, $\mathbb{E}[\text{ESTREG}]$ is the regret measured with $\bar{P}_{i(t)}$ and $\widehat{\ell}_t$, which is controlled by the FTRL procedure and of order $\widetilde{\mathcal{O}}(L\sqrt{|S||A|T})$. Put together, this proves the $\widetilde{\mathcal{O}}(L|S|\sqrt{|A|T})$ regret shown in [Theorem 4.1.1](#).

In the stochastic world, we fix the benchmark $\pi = \pi^\star$. To obtain polylogarithmic regret, an important observation is that we now have to make use of the potentially negative term $\text{ERR}_2$ instead of simply bounding it by $\mathcal{O}(1)$ (in expectation). Specifically, inspired by [[Simchowitz and Jamieson, 2019]](#), we propose a new decomposition on $\text{ERR}_1$ and $\text{ERR}_2$ *jointly* as follows (see [Appendix D.1](#)): $\text{ERR}_1 + \text{ERR}_2 = \text{ERRSUB} + \text{ERROPT} + \text{OCCDIFF} + \text{BIAS}$. Here,

- $\text{ERRSUB} = \sum_{t=1}^{T} \sum_{s \neq s_L} \sum_{a \neq \pi^\star(s)} q_t(s,a) \widehat{E}_t^{\pi^\star}(s,a)$ measures some estimation error contributed by the suboptimal actions, where $\widehat{E}_t^{\pi^\star}(s,a) = \ell_t(s,a) + \mathbb{E}_{s' \sim P(\cdot|s,a)}[\widehat{V}_t^{\pi^\star}(s')] - \widehat{Q}_t^{\pi^\star}(s,a)$ is a "surplus" function (a term taken from [[Simchowitz and Jamieson, 2019]](#));

- $\text{ERROPT} = \sum_{t=1}^{T} \sum_{s \neq s_L} \sum_{a = \pi^\star(s)} (q_t(s,a) - q_t^\star(s,a)) \widehat{E}_t^{\pi^\star}(s,a)$ measures some estimation error contributed by the optimal action, where $q_t^\star(s,a)$ is the probability of visiting a trajectory of the form $(s_0, \pi^\star(s_0)), (s_1, \pi^\star(s_1)), \ldots, (s_{k(s)-1}, \pi^\star(s_{k(s)-1})), (s,a)$ when executing policy $\pi_t$;

- $\text{OCCDIFF} = \sum_{t=1}^{T} \sum_{s \neq s_L} \sum_{a \in A} (q_t(s,a) - \widehat{q}_t(s,a)) \left(\widehat{Q}_t^{\pi^\star}(s,a) - \widehat{V}_t^{\pi^\star}(s)\right)$ measures the occupancy measure difference between $q_t$ and $\widehat{q}_t$;

- $\text{BIAS} = \sum_{t=1}^{T} \sum_{s \neq s_L} \sum_{a \neq \pi^\star(s)} q_t^\star(s,a) \left(\widehat{V}_t^{\pi^\star}(s) - V_t^{\pi^\star}(s)\right)$ measures some estimation error for $\pi^\star$, which, similar to $\text{ERR}_2$, is of order $\mathcal{O}(1)$ in expectation due to optimism.

The next key step is to show that the terms $\text{ERRSUB}, \text{ERROPT}, \text{OCCDIFF}$, and $\text{ESTREG}$ can all be upper bounded by some quantities that admit a certain self-bounding property similarly to the right-hand side of [Eq. (3)](#). We identify four such quantities and present them using functions $\mathbb{G}_1, \mathbb{G}_2, \mathbb{G}_3$, and $\mathbb{G}_4$, whose definitions are deferred to [Appendix D.2](#) due to space limit. Combining these bounds for each term, we obtain the following important lemma.

**Lemma 5.1.** *With $\delta = \frac{1}{T^2}$, [Algorithm 1](#) ensures that $\text{Reg}_T(\pi^\star)$ is at most $\mathcal{O}(L^4|S|^3|A|^2 \ln^2 T)$ plus:*

$$\mathbb{E}\left[\mathcal{O}\left(\underbrace{\mathbb{G}_1\left(L^4|S|\ln T\right)}_{\text{from ERRSUB}} + \underbrace{\mathbb{G}_2\left(L^4|S|\ln T\right)}_{\text{from ERROPT}} + \underbrace{\mathbb{G}_3\left(L^4 \ln T\right)}_{\text{from OCCDIFF}} + \underbrace{\mathbb{G}_4\left(L^5|S||A|\ln T \ln(|S||A|)\right)}_{\text{from ESTREG}}\right)\right].$$

Finally, as mentioned, each of the $\mathbb{G}_1, \mathbb{G}_2, \mathbb{G}_3$, and $\mathbb{G}_4$ functions can be shown to admit the following self-bounding property, such that similarly to what we argue in [Eq. (4)](#), picking the optimal values of $\alpha$ and $\beta$ and rearranging leads to the polylogarithmic regret bound shown in [Theorem 4.1.1](#).

**Lemma 5.2** (Self-bounding property). *Under Condition ([1](#)), we have for any $\alpha, \beta \in (0,1)$,*

$$\mathbb{E}[\mathbb{G}_1(J)] \leq \alpha \cdot (\text{Reg}_T(\pi^\star) + C) + \mathcal{O}\left(\frac{1}{\alpha} \cdot \sum_{s \neq s_L} \sum_{a \neq \pi^\star(s)} \frac{J}{\Delta(s,a)}\right),$$

$$\mathbb{E}\left[\mathbb{G}_2(J)\right] \leq \beta \cdot (\mathrm{Reg}_T(\pi^\star) + C) + \mathcal{O}\left(\frac{1}{\beta} \cdot \frac{L|S|J}{\Delta_{\mathrm{MIN}}}\right),$$

$$\mathbb{E}\left[\mathbb{G}_3(J)\right] \leq (\alpha + \beta) \cdot (\mathrm{Reg}_T(\pi^\star) + C) + \mathcal{O}\left(\frac{1}{\alpha} \cdot \sum_{s \neq s_L} \sum_{a \neq \pi^\star(s)} \frac{L^2|S|J}{\Delta(s,a)}\right) + \mathcal{O}\left(\frac{1}{\beta} \cdot \frac{L^2|S|^2 J}{\Delta_{\mathrm{MIN}}}\right),$$

$$\mathbb{E}\left[\mathbb{G}_4(J)\right] \leq \beta \cdot (\mathrm{Reg}_T(\pi^\star) + C) + \mathcal{O}\left(\frac{1}{\beta} \cdot \frac{J}{\Delta_{\mathrm{MIN}}}\right).$$

We emphasize again that the proposed joint decomposition on $\mathrm{ERR}_1 + \mathrm{ERR}_2$ plays a crucial rule in this analysis and addresses the key challenge on how to bound the transition estimation error by something better than $\sqrt{T}$. We also point out that in this analysis, only EstReg is related to the FTRL procedure, while the other three terms are purely based on our new framework to handle unknown transition. In fact, the reason that we can only derive a $\mathrm{polylog}(T)$ bound on $\mathrm{Reg}_T(\pi^\star)$ but not directly on $\mathrm{Reg}_T(\mathring{\pi})$ is also due to these three terms — they can be related to the right-hand side of Condition (1) only when we use the benchmark $\pi = \pi^\star$ but not when $\pi = \mathring{\pi}$. This is not the case for EstReg, which is the reason why Jin and Luo [2020] are able to derive a bound on $\mathrm{Reg}_T(\mathring{\pi})$ directly when the transition is known. Whether this issue can be addressed is left as a future direction.

# 6 Conclusions

In this work, we propose an algorithm for learning episodic MDPs which achieves favorable regret guarantees simultaneously in the stochastic and adversarial worlds with unknown transition. We start from the known transition setting and propose a loss-shifting trick for FTRL applied to MDPs, which simplifies the method of Jin and Luo [2020] and improves their results. Then, we design a new framework to extend our known transition algorithm to the unknown transition case, which is critical for the application of the loss-shifting trick. Finally, we develop a novel analysis which carefully upper bounds the transition estimation error by (a fraction of) the regret itself plus a gap-dependent poly-logarithmic term in the stochastic setting, resulting in our final best-of-both-worlds result.

Besides the open questions discussed earlier (such as improving our bounds in Theorem 4.1.1 and Theorem 4.1.2), one other key future direction is to remove the assumption that there exists a unique optimal action for each state, which appears to be challenging despite the recent progress for the bandit case [Ito, 2021], since the occupancy measure computed from Eq. (10) has a very complicated structure. Another interesting direction would be to extend the sub-optimality gap function to other fine-grained gap functions, such as that of Dann et al. [2021].

## Acknowledgments and Disclosure of Funding

HL is supported by NSF Award IIS-1943607 and a Google Faculty Research Award. LH is supported in part by the Technology and Innovation Major Project of the Ministry of Science and Technology of China under Grants 2020AAA0108400 and 2020AAA0108403. We thank Max Simchowitz for many helpful discussions, and the anonymous reviewers for their valuable feedback and suggestions.

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
