# Contents

**An important convention**  Note that the value of $m_i(s, a)$ is changing in the algorithm. For the entire analysis, we see $m_i(s, a)$ as its initial value, which is the number of visits to $(s, a)$ from epoch 1 to epoch $i - 1$. In this sense, if we let $N$ be the total number of epochs, then $m_{N+1}(s, a)$ is naturally defined as the total number of visits to $(s, a)$ within $T$ episodes.

## A    Best of Both Worlds for MDPs with Known Transition

In this section, we show how to extend the loss-shifting technique to MDPs with known transition and obtain best-of-both-worlds results.

### A.1    Loss-shifting Technique

First of all, we introduce a general invariant condition with a fixed transition in Lemma A.1.1

**Lemma A.1.1.** *Fix the transition function $P$. For any policy $\pi$ and loss function $\mathring{\ell} : S \times A \to \mathbb{R}$, define invariant function $g \in S \times A \to \mathbb{R}$ as:*

$$g^{P,\pi,\mathring{\ell}}(s, a) \triangleq \left( Q^{P,\pi,\mathring{\ell}}(s, a) - V^{P,\pi,\mathring{\ell}}(s) - \mathring{\ell}(s, a) \right), \tag{15}$$

*where $Q^{P,\pi,\mathring{\ell}}$ and $V^{P,\pi,\mathring{\ell}}$ are state-action value and state value functions associated with $\mathring{\ell}$ and the fixed policy $\pi$. Then, it holds for any policy $\pi'$ that*

$$\left\langle q^{P,\pi'}, g^{P,\pi,\mathring{\ell}} \right\rangle \triangleq \sum_{s \neq s_L} \sum_{a \in A} q^{P,\pi'}(s, a) \cdot g^{P,\pi,\mathring{\ell}}(s, a) = -V^{P,\pi,\mathring{\ell}}(s_0)$$

*where $V^{P,\pi,\mathring{\ell}}(s_0)$ only depends on $\pi$ and $\mathring{\ell}$ (but not $\pi'$).*

*Proof.* For notational convenience, we drop the superscripts for fixed transition $P$ and loss function $\mathring{\ell}$. By the standard performance difference lemma [Kakade, 2003, Theorem 5.2.1], it holds for any policy $\pi'$ that

$$V^{\pi'}(s_0) - V^{\pi}(s_0) = \sum_{s \neq s_L} \sum_{a \in A} q^{\pi'}(s, a) \left( Q^{\pi}(s, a) - V^{\pi}(s) \right). \tag{16}$$

On the other hand, it also holds that

$$V^{\pi'}(s_0) = \sum_{s \neq s_L} \sum_{a \in A} q^{\pi'}(s, a) \mathring{\ell}(s, a). \tag{17}$$

Therefore, subtracting $V^{\pi'}(s_0)$ from Eq. (16) yields that

$$-V^{\pi}(s_0) = \sum_{s \neq s_L} \sum_{a \in A} q^{\pi'}(s, a) \left( Q^{\pi}(s, a) - V^{\pi}(s) - \mathring{\ell}(s, a) \right)$$

which completes the proof after putting back the superscripts for $P$ and $\mathring{\ell}$.    □

As discussed in Section 3, the invariant function $g^{P,\pi,\mathring{\ell}}$ defined in Eq. (15) allows us to treat FTRL as dealing with a hypothesized loss sequence, as restated below.

**Corollary A.1.2.** *Consider the selected occupancy measure $\widehat{q}_t$ via* FTRL *with respect to a regularizer $\phi_t(\cdot)$ and loss sequence $\{\widehat{\ell}_\tau\}_{\tau < t}$ (on the decision set $\Omega(\bar{P})$), then it holds that*

$$\widehat{q}_t = \operatorname*{argmin}_{q \in \Omega(\bar{P})} \left\langle q, \sum_{\tau < t} \widehat{\ell}_\tau \right\rangle + \phi_t(q) = \operatorname*{argmin}_{q \in \Omega(\bar{P})} \left\langle q, \sum_{\tau < t} (\widehat{\ell}_\tau + g_\tau) \right\rangle + \phi_t(q).$$

*for any invariant function sequence $\{g_\tau\}_{\tau < t}$ which are constructed with hypothesized losses $\{\mathring{\ell}_\tau\}_{\tau < t}$ and policies $\{\pi'_\tau\}_{\tau < t}$.*

*Proof.* By Lemma A.1.1, one can verify that

$$\left\langle q, \sum_{\tau < t} g_\tau \right\rangle = - \sum_{\tau < t} V^{\bar{P}, \pi'_\tau, \mathring{\ell}_\tau}(s_0)$$

for any occupancy measure $q \in \Omega(\bar{P})$. Therefore, this term does not affect the optimization. $\square$

Then, we consider the "loss-shifting function" defined in Eq. (6), that is, constructing $g_t$ via the loss estimator $\widehat{\ell}_t$ and the policy $\pi_t$ selected at episode $t$. Importantly, in the known transition setting where $\widehat{q}_t = q_t$, $\widehat{\ell}_t$ is inverse propensity weighted estimator, in other words, $\widehat{\ell}_t(s, a) = \mathbb{I}_t(s,a)\ell_t(s,a)/q_t(s,a)$. More specifically, we have

$$g_t(s,a) = \widehat{Q}_t(s,a) - \widehat{V}_t(s) - \widehat{\ell}_t(s,a),$$

where

$$\widehat{Q}_t(s,a) = \widehat{\ell}_t(s,a) + \sum_{s' \in S_{k(s)+1}} \bar{P}(s'|s,a)\widehat{V}_t(s'), \quad \widehat{V}_t(s) = \sum_{a \in A} \pi_t(a|s)\widehat{Q}_t(s,a)$$

(with $\widehat{V}_t(s_L) = 0$). Below we show several useful properties, which are key to achieve the best-of-both-worlds guarantee in the known transition setting.

**Lemma A.1.3.** *With $\bar{P} = P$ being the true transition function (therefore, $\widehat{q}_t = q_t$), we have*

- $q_t(s,a)\widehat{Q}_t(s,a) \leq L,$

- $q_t(s)\widehat{V}_t(s) \leq L,$

- $\mathbb{E}_t\left[\left(\widehat{Q}_t(s,a) - \widehat{V}_t(s)\right)^2\right] \leq \frac{2L^2(1-\pi_t(a|s))}{q_t(s,a)},$

*for all state-action pairs $(s,a)$ (where $\mathbb{E}_t$ denotes the conditional expectation given everything before episode $t$).*

*Proof.* Denote by $q_t(s', a'|s, a)$ the probability of visiting $(s', a')$ after taking action $a$ at state $s$ and following $\pi_t$ afterwards. Then we have $\widehat{Q}_t(s,a) = \sum_{k=k(s)}^{L-1} \sum_{s' \in S_k} \sum_{a' \in A} q_t(s', a'|s, a)\widehat{\ell}_t(s', a')$. Therefore, plugging in the definition of $\widehat{\ell}_t(s, a)$, we verify the following:

$$q_t(s,a)\widehat{Q}_t(s,a) = \sum_{k=k(s)}^{L-1} \sum_{s' \in S_k} \sum_{a' \in A} \frac{q_t(s,a)q_t(s', a'|s, a)}{q_t(s', a')} \mathbb{I}_t(s', a')\ell_t(s', a')$$

$$\leq \sum_{k=k(s)}^{L-1} \sum_{s' \in S_k} \sum_{a' \in A} \mathbb{I}_t(s', a') \leq L,$$

where the inequality is by $q_t(s,a)q_t(s', a'|s, a) \leq q_t(s', a')$ and $\ell_t(s', a') \in [0, 1]$. This also proves $q_t(s)\widehat{V}_t(s) \leq L$ using the definition of $\widehat{V}_t(s)$.

To prove the last statement, we first note that

$$\mathbb{E}_t\left[\left(\widehat{Q}_t(s,a) - \widehat{V}_t(s)\right)^2\right] \leq 2\mathbb{E}_t\left[(1 - \pi_t(a|s))^2\widehat{Q}_t(s,a)^2 + \left(\sum_{b \neq a} \pi_t(b|s)\widehat{Q}_t(s,b)\right)^2\right] \quad (18)$$

by the fact $(x - y)^2 \leq 2x^2 + 2y^2$ for all $x, y \in \mathbb{R}$.

For the first term in Eq. (18), we have:

$$\mathbb{E}_t\left[\widehat{Q}_t(s,a)^2\right] = \mathbb{E}_t\left[\left(\sum_{k=k(s)}^{L-1} \sum_{s' \in S_k} \sum_{a' \in A} \frac{q_t(s', a'|s, a)}{q_t(s', a')} \mathbb{I}_t(s', a')\ell_t(s', a')\right)^2\right]$$

$$\leq L \cdot \mathbb{E}_t \left[ \sum_{k=k(s)}^{L-1} \left( \sum_{s' \in S_k} \sum_{a' \in A} \frac{q_t(s', a'|s, a)}{q_t(s', a')} \mathbb{I}_t(s', a') \ell_t(s', a') \right)^2 \right]$$

$$\leq L \cdot \mathbb{E}_t \left[ \sum_{k=k(s)}^{L-1} \sum_{s' \in S_k} \sum_{a' \in A} \frac{q_t(s', a'|s, a)^2}{q_t(s', a')^2} \mathbb{I}_t(s', a') \right]$$

$$= L \cdot \sum_{k=k(s)}^{L-1} \sum_{s' \in S_k} \sum_{a' \in A} \frac{q_t(s', a'|s, a)^2}{q_t(s', a')}$$

$$= \frac{L}{q_t(s, a)} \cdot \sum_{k=k(s)}^{L-1} \sum_{s' \in S_k} \sum_{a' \in A} \frac{q_t(s, a) q_t(s', a'|s, a)}{q_t(s', a')} \cdot q_t(s', a'|s, a)$$

$$\leq \frac{L}{q_t(s, a)} \cdot \sum_{k=k(s)}^{L-1} \sum_{s' \in S_k} \sum_{a' \in A} q_t(s', a'|s, a) \leq \frac{L^2}{q_t(s, a)},$$

where the second line uses the Cauchy-Schwartz inequality; the third line follows from the fact $\mathbb{I}_t(s, a) \mathbb{I}_t(s', a') = 0$ for all $(s, a), (s', a') \in S_k \times A$ such that $(s, a) \neq (s', a')$; the fourth line uses $\mathbb{E}_t[\mathbb{I}_t(s', a')] = q_t(s', a')$; and the last line follows from the fact $q_t(s, a) q_t(s', a'|s, a) \leq q_t(s', a')$.

Repeating the similar arguments, we bound the second term as

$$\mathbb{E}_t \left[ \left( \sum_{b \neq a} \pi_t(b|s) \widehat{Q}_t(s, b) \right)^2 \right] = \mathbb{E}_t \left[ \left( \sum_{k=k(s)}^{L-1} \sum_{s' \in S_k} \sum_{a' \in A} \left( \sum_{b \neq a} \pi_t(b|s) q_t(s', a'|s, b) \right) \widehat{\ell}_t(s', a') \right)^2 \right]$$

$$\leq L \cdot \mathbb{E}_t \left[ \sum_{k=k(s)}^{L-1} \left( \sum_{s' \in S_k} \sum_{a' \in A} \left( \sum_{b \neq a} \pi_t(b|s) q_t(s', a'|s, b) \right) \widehat{\ell}_t(s', a') \right)^2 \right]$$

(Cauchy-Schwarz inequality)

$$\leq L \cdot \mathbb{E}_t \left[ \sum_{k=k(s)}^{L-1} \sum_{s' \in S_k} \sum_{a' \in A} \left( \sum_{b \neq a} \pi_t(b|s) q_t(s', a'|s, b) \right)^2 \frac{\mathbb{I}_t(s', a')}{q_t(s', a')^2} \right]$$

$(\mathbb{I}_t(s, a) \mathbb{I}_t(s', a') = 0$ for $(s, a) \neq (s', a'))$

$$= L \cdot \sum_{k=k(s)}^{L-1} \sum_{s' \in S_k} \sum_{a' \in A} \left( \frac{\sum_{b \neq a} \pi_t(b|s) q_t(s', a'|s, b)}{q_t(s', a')} \right) \cdot \left( \sum_{b \neq a} \pi_t(b|s) \cdot q_t(s', a'|s, b) \right)$$

$$= \frac{L}{q_t(s)} \cdot \sum_{k=k(s)}^{L-1} \sum_{s' \in S_k} \sum_{a' \in A} \left( \frac{\sum_{b \neq a} q_t(s, b) q_t(s', a'|s, b)}{q_t(s', a')} \right) \cdot \left( \sum_{b \neq a} \pi_t(b|s) \cdot q_t(s', a'|s, b) \right)$$

$$\leq \frac{L}{q_t(s)} \cdot \sum_{k=k(s)}^{L-1} \sum_{s' \in S_k} \sum_{a' \in A} \left( \sum_{b \neq a} \pi_t(b|s) \cdot q_t(s', a'|s, b) \right)$$

$$= \frac{L}{q_t(s)} \cdot \sum_{b \neq a} \pi_t(b|s) \cdot \left( \sum_{k=k(s)}^{L-1} \left( \sum_{s' \in S_k} \sum_{a' \in A} q_t(s', a'|s, b) \right) \right)$$

$$\leq \frac{L^2}{q_t(s)} \cdot \sum_{b \neq a} \pi_t(b|s) = \frac{L^2 (1 - \pi_t(a|s))}{q_t(s)}.$$

Plugging these bounds into Eq. (18) concludes the proof:

$$\mathbb{E}_t \left[ \left( \widehat{Q}_t(s, a) - \widehat{V}_t(s) \right)^2 \right] \leq 2L^2 \left( \frac{(1 - \pi_t(a|s))^2}{q_t(s, a)} + \frac{1 - \pi_t(a|s)}{q_t(s)} \right)$$

**Algorithm 2** Best-of-both-worlds for MDPs with Known Transition and Full-information Feedback

**for** $t = 1$ **to** $T$ **do**
    Compute $q_t = \operatorname{argmin}_{q \in \Omega(P)} \langle q, \sum_{\tau < t} \ell_\tau \rangle + \phi_t(q)$ where $\phi_t(q)$ is defined in Eq. (20).
    Execute policy $\pi_t$ where $\pi_t(a|s) = q_t(s,a)/q_t(s)$.
    Observe the entire loss function $\ell_t$.

$$= 2L^2 \left(1 - \pi_t(a|s)\right) \left( \frac{1 - \pi_t(a|s)}{q_t(s,a)} + \frac{1}{q_t(s)} \right) = \frac{2L^2 \left(1 - \pi_t(a|s)\right)}{q_t(s,a)}.$$

$\square$

## A.2 Known Transition and Full-information Feedback: FTRL with Shannon Entropy

Although not mentioned in the main text, in this section, we discuss a simple application of the loss-shifting technique: achieving the best-of-both-worlds in the full-information feedback setting with known transition via the FTRL framework with the Shannon entropy regularizer. Some of the lemmas in this section are useful for proving similar results for the unknown transition case in Appendix B.

Therefore, the specific state-action and state value functions defined in Lemma A.1.3 are now constructed based on the received loss vector $\ell_t$, instead of the loss estimator $\widehat{\ell}_t$. In other words, the loss-shifting function $g_t$ is defined as $g_t(s,a) = \widehat{Q}(s,a) - \widehat{V}(s) - \ell_t(s,a)$ where

$$\widehat{Q}_t(s,a) = \ell_t(s,a) + \sum_{s' \in S_{k(s)+1}} P(s'|s,a) \widehat{V}_t(s), \quad \widehat{V}_t(s) = \sum_{a \in A} \pi_t(a|s) \widehat{Q}_t(s,a). \quad (19)$$

Our goal is to show that, using an adaptive time-varying learning rate schedule, FTRL with Shannon entropy is able to attain a self-bounding regret guarantee with full-information feedback. This idea will be further discussed in Appendix B to address the unknown transition setting.

In particular, the algorithm uses following regularizer for episode $t$:

$$\phi_t(q) = \frac{1}{\eta_t} \sum_{s \neq s_L} \sum_{a \in A} q(s,a) \ln q(s,a) = \frac{1}{\eta_t} \phi(q), \quad (20)$$

where the adaptive learning rate $\eta_t$ is defined as $\eta_t = \sqrt{\frac{L \ln(|S||A|)}{M_{t-1} + 64 L^3 \ln(|S||A|)}}$ with

$$M_t = \sum_{\tau=1}^{t} \min \left\{ \sum_{s \neq s_L} \sum_{a \in A} q_\tau(s,a) \left( \widehat{Q}_\tau(s,a) - \widehat{V}_\tau(s) \right)^2, \sum_{s \neq s_L} \sum_{a \in A} q_\tau(s,a) \ell_\tau(s,a)^2 \right\}.$$

The pseudocode of our algorithm is presented in Algorithm 2.

In the known transition setting, we assume the loss functions satisfy a more general condition compared to Condition (1): there exists a deterministic policy $\pi^\star : S \to A$, a gap function $\Delta : S \times A \to \mathbb{R}_+$ and a constant $C > 0$ such that

$$\operatorname{Reg}_T(\mathring{\pi}) \geq \mathbb{E} \left[ \sum_{t=1}^{T} \sum_{s \neq s_L} \sum_{a \neq \pi^\star(s)} q_t(s,a) \Delta(s,a) - C \right]. \quad (21)$$

Note that this is only weaker than Condition (1) since $\operatorname{Reg}_T(\mathring{\pi}) \geq \operatorname{Reg}_T(\pi^\star)$.

Then, we show that Algorithm 2 ensures a worst-case guarantee $\operatorname{Reg}_T(\mathring{\pi}) = \widetilde{\mathcal{O}}(L\sqrt{T})$, and simultaneously an adaptive regret bound which further leads to logarithmic regret under Condition (21) (Corollary A.2.2). Importantly, the worst-case regret bound matches the lower bound of learning MDPs with known transition and full-information feedback [Zimin and Neu, 2013].

**Theorem A.2.1.** *Algorithm 2 ensures that* $\operatorname{Reg}_T(\mathring{\pi})$ *is bounded by*

$$\mathcal{O} \left( \sqrt{\min \left\{ L^2 T, L^3 \mathbb{E} \left[ \sum_{t=1}^{T} \sum_{s \neq s_L} \sum_{a \neq \pi(s)} q_t(s,a) \right] \right\} \ln(|S||A|)} + L^2 \ln(|S||A|) \right) \quad (22)$$

*for any mapping $\pi : S \to A$.*

*Proof.* Due to the invariant property (that $\langle q, g_t \rangle$ is independent of $q \in \Omega(P)$), we can apply Lemma A.2.3 with $\widehat{\ell}_t$ being either $\ell_t$ or $\ell_t + g_t$ for any $t$ — note that the condition $\eta_t \widehat{\ell}_t(s, a) \geq -1$ is always satisfied since $\ell_t(s, a) \in [0, 1]$ and $\widehat{Q}_t(s, a) - \widehat{V}_t(s) \in [-L, L]$. Therefore, we have for any $u \in \Omega(P)$,

$$
\sum_{t=1}^{T} \langle q_t - u, \ell_t \rangle \leq \frac{L \ln(|S||A|)}{\eta_{T+1}}
$$

$$
+ \sum_{t=1}^{T} \eta_t \min \left\{ \sum_{s \neq s_L} \sum_{a \in A} q_t(s, a) \left( \widehat{Q}_t(s, a) - \widehat{V}_t(s) \right)^2, \sum_{s \neq s_L} \sum_{a \in A} q_t(s, a) \ell_t(s, a)^2 \right\}
$$

$$
= \frac{L \ln(|S||A|)}{\eta_{T+1}} + \sum_{t=1}^{T} \eta_t \left( M_t - M_{t-1} \right), \qquad \text{(definition of } M_t\text{)}
$$

$$
= \frac{L \ln(|S||A|)}{\eta_{T+1}} + \sum_{t=1}^{T} \eta_t \left( \sqrt{M_t} + \sqrt{M_{t-1}} \right) \left( \sqrt{M_t} - \sqrt{M_{t-1}} \right),
$$

$$
\leq \frac{L \ln(|S||A|)}{\eta_{T+1}} + 2 \sum_{t=1}^{T} \eta_t \sqrt{M_{t-1} + L} \left( \sqrt{M_t} - \sqrt{M_{t-1}} \right). \quad (M_t \leq M_{t-1} + L)
$$

Further plugging in the definition of $\eta_t$ and taking expectation, we arrive at

$$
\text{Reg}_T(\mathring{\pi}) \leq \mathbb{E} \left[ \frac{L \ln(|S||A|)}{\eta_{T+1}} + 2 \sqrt{L \ln(|S||A|)} \sum_{t=1}^{T} \left( \sqrt{M_t} - \sqrt{M_{t-1}} \right) \right]
$$

$$
= \mathbb{E} \left[ \sqrt{L \ln(|S||A|) \left( M_T + 64 L^3 \ln |S||A| \right)} + 2 \sqrt{L M_T \ln(|S||A|)} \right]
$$

$$
= \mathcal{O} \left( \sqrt{L \mathbb{E}\left[ M_T \right] \ln(|S||A|)} + L^2 \ln(|S||A|) \right).
$$

It remains to bound $M_T$. First, we note that

$$
M_T = \sum_{t=1}^{T} \min \left\{ \sum_{s \neq s_L} \sum_{a \in A} q_t(s, a) \left( \widehat{Q}_t(s, a) - \widehat{V}_t(s) \right)^2, \sum_{s \neq s_L} \sum_{a \in A} q_t(s, a) \ell_t(s, a)^2 \right\}
$$

$$
\leq \min \left\{ \sum_{t=1}^{T} \sum_{s \neq s_L} \sum_{a \in A} q_t(s, a) \left( \widehat{Q}_t(s, a) - \widehat{V}_t(s) \right)^2, \sum_{t=1}^{T} \sum_{s \neq s_L} \sum_{a \in A} q_t(s, a) \ell_t(s, a)^2 \right\}
$$

$$
\leq \min \left\{ \sum_{t=1}^{T} \sum_{s \neq s_L} \sum_{a \in A} q_t(s, a) \left( \widehat{Q}_t(s, a) - \widehat{V}_t(s) \right)^2, LT \right\}.
$$

where the second line follows from the fact $\min \{a, b\} + \min \{c, d\} \leq \min \{a + c, b + d\}$, and the third line uses the property $0 \leq \ell_t(s, a) \leq 1$ for all state-action pairs $(s, a)$.

On the other hand, we have

$$
\left( \widehat{Q}_t(s, a) - \widehat{V}_t(s) \right)^2 \leq 2 \left[ (1 - \pi_t(a|s))^2 \, \widehat{Q}_t(s, a)^2 + \left( \sum_{b \neq a} \pi_t(b|s) \widehat{Q}_t(s, b) \right)^2 \right]
$$

$$
\leq 2L^2 \cdot \left[ (1 - \pi_t(a|s))^2 + (1 - \pi_t(a|s))^2 \right]
$$

$$
\leq 4L^2 \left( 1 - \pi_t(a|s) \right), \tag{23}
$$

where we use the facts $(a - b)^2 \leq 2(a^2 + b^2)$ and $0 \leq \widehat{Q}_t(s, a) \leq L$ for all state-action pairs $(s, a)$. Therefore, we have for any mapping $\pi : S \to A$,

$$\sum_{t=1}^{T} \sum_{s \neq s_L} \sum_{a \in A} q_t(s, a) \left( \widehat{Q}_t(s, a) - \widehat{V}_t(s) \right)^2$$

$$\leq 4L^2 \cdot \sum_{t=1}^{T} \sum_{s \neq s_L} \sum_{a \in A} q_t(s, a) \left( 1 - \pi_t(a|s) \right)$$

$$\leq 4L^2 \cdot \sum_{t=1}^{T} \sum_{s \neq s_L} \left( q_t(s) \cdot (1 - \pi_t(\pi(s)|s)) + \sum_{a \neq \pi(s)} q_t(s, a) \right)$$

$$= 8L^2 \cdot \sum_{t=1}^{T} \sum_{s \neq s_L} \sum_{a \neq \pi(s)} q_t(s, a), \tag{24}$$

which finishes the proof. $\qquad \square$

**Corollary A.2.2.** *Suppose Condition* (21) *holds.* *Algorithm 2 guarantees that:*

$$\mathrm{Reg}_T(\mathring{\pi}) = \mathcal{O}\left( U + \sqrt{CU} \right), \text{ where } U = \frac{L^3 \ln(|S||A|)}{\Delta_{\mathrm{MIN}}}.$$

*Proof.* By Theorem A.2.1, $\mathrm{Reg}_T(\mathring{\pi})$ is bounded by

$$\kappa \cdot \left( \sqrt{L^3 \ln(|S||A|) \cdot \mathbb{E}\left[ \sum_{t=1}^{T} \sum_{s \neq s_L} \sum_{a \neq \pi^\star(s)} q_t(s, a) \right]} + L^2 \ln(|S||A|) \right)$$

where $\kappa \geq 1$ is a universal constant, and $\pi^\star$ is the mapping specified in Condition (21).

For any $z > 1$, $\mathrm{Reg}_T(\mathring{\pi})$ is bounded by

$$\kappa \sqrt{L^3 \ln(|S||A|) \cdot \mathbb{E}\left[ \sum_{t=1}^{T} \sum_{s \neq s_L} \sum_{a \neq \pi^\star(s)} q_t(s, a) \right]} + \kappa L^2 \ln(|S||A|)$$

$$= \sqrt{\frac{z \kappa^2 L^3 \ln(|S||A|)}{2 \Delta_{\mathrm{MIN}}} \cdot \left( \frac{2}{z} \cdot \mathbb{E}\left[ \sum_{t=1}^{T} \sum_{s \neq s_L} \sum_{a \neq \pi^\star(s)} q_t(s, a) \Delta_{\mathrm{MIN}} \right] \right)} + \kappa L^2 \ln(|S||A|)$$

$$\leq \frac{\mathrm{Reg}_T(\mathring{\pi}) + C}{z} + \frac{z \kappa^2 L^3 \ln(|S||A|)}{4 \Delta_{\mathrm{MIN}}} + \kappa L^2 \ln(|S||A|)$$

$$\leq \frac{\mathrm{Reg}_T(\mathring{\pi}) + C}{z} + z \cdot 2\kappa^2 U,$$

where the third line uses the AM-GM inequality and Eq. (21), and the last line uses the shorthand $U$ and the facts $\kappa, z > 1$ and $\Delta_{\mathrm{MIN}} \leq 1$.

Therefore, by defining $x = z - 1 > 0$, we can rearrange and arrive at

$$\mathrm{Reg}_T(\mathring{\pi}) \leq \frac{C}{z - 1} + \frac{z^2}{z - 1} \cdot 2\kappa^2 U$$

$$= \frac{C}{x} + \frac{(x + 1)^2}{x} \cdot \left( 2\kappa^2 U \right)$$

$$= \frac{1}{x} \cdot \left( C + 2\kappa^2 U \right) + x \cdot \left( 2\kappa^2 U \right) + 4\kappa^2 U,$$

where we replace all $z$'s in the second line. Picking the optimal $x = \sqrt{\frac{C + 2\kappa^2 U}{2\kappa^2 U}}$ gives

$$\mathrm{Reg}_T(\mathring{\pi}) \leq 2\sqrt{\left( C + 2\kappa^2 U \right) \cdot \left( 2\kappa^2 U \right)} + 4\kappa^2 U$$

$$\leq 8\kappa^2 U + 2\sqrt{2}\kappa \cdot \sqrt{CU}$$
$$= \mathcal{O}\left(U + \sqrt{UC}\right),$$

where the second line follows from the fact $\sqrt{x+y} \leq \sqrt{x} + \sqrt{y}$. $\qquad\square$

**Lemma A.2.3.** *Suppose* $q_t = \arg\min_{q \in \Omega(P)} \left\langle q, \sum_{\tau < t} \widehat{\ell}_\tau \right\rangle + \phi_t(q)$, *where* $\phi_t(q) = \frac{1}{\eta_t}\phi(q)$ *for some* $\eta_t > 0$, $\phi(q) = \sum_{s \neq s_L} \sum_{a \in A} q(s,a) \ln q(s,a)$, *and* $\eta_t \widehat{\ell}_t(s,a) \geq -1$ *holds for all* $t$ *and* $(s,a)$. *Then*

$$\sum_{t=1}^{T} \left\langle q_t - u, \widehat{\ell}_t \right\rangle \leq \frac{L\ln(|S||A|)}{\eta_{T+1}} + \sum_{t=1}^{T} \eta_t \cdot \sum_{s \neq s_L} \sum_{a \in A} q_t(s,a)\widehat{\ell}_t(s,a)^2,$$

*holds for any* $u \in \Omega(P)$.

*Proof.* Let $\Phi_t = \min_{q \in \Omega(P)} \left\langle q, \sum_{\tau=1}^{t-1} \widehat{\ell}_\tau \right\rangle + \phi_t(q)$ and $D_F(u,v)$ being the Bregman divergence with convex function $F$, that is, $D_F(u,v) = F(u) - F(v) - \langle u-v, \nabla F(v)\rangle$.

Then, we have

$$\Phi_t = \left\langle q_t, \sum_{\tau=1}^{t-1} \widehat{\ell}_\tau \right\rangle + \phi_t(q_t)$$

$$= \left\langle q_{t+1}, \sum_{\tau=1}^{t-1} \widehat{\ell}_\tau \right\rangle + \phi_t(q_{t+1}) - \left(\left\langle q_{t+1} - q_t, \sum_{\tau=1}^{t-1} \widehat{\ell}_\tau \right\rangle + \phi_t(q_{t+1}) - \phi_t(q_t)\right)$$

$$\leq \left\langle q_{t+1}, \sum_{\tau=1}^{t-1} \widehat{\ell}_\tau \right\rangle + \phi_t(q_{t+1}) - (-\langle q_{t+1} - q_t, \nabla\phi_t(q_t)\rangle + \phi_t(q_{t+1}) - \phi_t(q_t))$$

$$= \left\langle q_{t+1}, \sum_{\tau=1}^{t-1} \widehat{\ell}_\tau \right\rangle + \phi_t(q_{t+1}) - D_{\phi_t}(q_{t+1}, q_t)$$

$$= \Phi_{t+1} - \left\langle q_{t+1}, \widehat{\ell}_t \right\rangle - (\phi_{t+1}(q_{t+1}) - \phi_t(q_{t+1})) - D_{\phi_t}(q_{t+1}, q_t),$$

where the third line follows from the first order optimality condition of $q_t$, that is, $\left\langle q_{t+1} - q_t, \nabla\phi_t(q_t) + \sum_{\tau=1}^{t-1} \widehat{\ell}_\tau \right\rangle \geq 0$.

Taking the summation over all episodes gives

$$\Phi_1 = \Phi_{T+1} - \sum_{t=1}^{T} \left\langle q_{t+1}, \widehat{\ell}_t \right\rangle - \sum_{t=1}^{T} (\phi_{t+1}(q_{t+1}) - \phi_t(q_{t+1})) - \sum_{t=1}^{T} D_{\phi_t}(q_{t+1}, q_t).$$

Therefore, we have

$$\sum_{t=1}^{T} \left\langle q_t - u, \widehat{\ell}_t \right\rangle$$

$$= \sum_{t=1}^{T} \left\langle q_t - u, \widehat{\ell}_t \right\rangle + \Phi_{T+1} - \Phi_1 - \sum_{t=1}^{T} \left\langle q_{t+1}, \widehat{\ell}_t \right\rangle - \sum_{t=1}^{T} (\phi_{t+1}(q_{t+1}) - \phi_t(q_{t+1})) - \sum_{t=1}^{T} D_{\phi_t}(q_{t+1}, q_t)$$

$$= \sum_{t=1}^{T} \left(\left\langle q_t - q_{t+1}, \widehat{\ell}_t \right\rangle - D_{\phi_t}(q_{t+1}, q_t)\right) - \sum_{t=1}^{T} \left\langle u, \widehat{\ell}_t \right\rangle + \Phi_{T+1} - \Phi_1 - \sum_{t=1}^{T} (\phi_{t+1}(q_{t+1}) - \phi_t(q_{t+1}))$$

$$\leq \underbrace{\sum_{t=1}^{T} \left(\left\langle q_t - q_{t+1}, \widehat{\ell}_t \right\rangle - D_{\phi_t}(q_{t+1}, q_t)\right)}_{\text{STABILITY}} + \underbrace{\phi_{T+1}(u) - \phi_1(q_1) - \sum_{t=1}^{T} (\phi_{t+1}(q_{t+1}) - \phi_t(q_{t+1}))}_{\text{PENALTY}}$$

where the last line follows from the optimality condition $\Phi_{T+1} \leq \sum_{t=1}^{T} \left\langle u, \widehat{\ell}_t \right\rangle + \phi_{T+1}(u)$.

To bound the stability term, we first consider relaxing the constraint and taking the maximum as:

$$\left\langle q_t - q_{t+1}, \widehat{\ell}_t \right\rangle - D_{\phi_t}(q_{t+1}, q_t) \leq \max_{q \in \mathbb{R}_+^{S \times A}} \left\langle q_t - q, \widehat{\ell}_t \right\rangle - D_{\phi_t}(q, q_t).$$

Denote by $\widetilde{q}_t$ the maximizer of the right hand side. Setting the gradient to zero yields the equality $\nabla\phi_t(q_t) - \nabla\phi_t(\widetilde{q}_t) = \widehat{\ell}_t$. By direction calculation, one can verify that $\widetilde{q}_t(s, a) = q_t(s, a) \cdot \exp\left(-\eta_t \cdot \widehat{\ell}_t(s, a)\right)$ for all state-action pairs, and the following inequality that

$$\left\langle q_t - q_{t+1}, \widehat{\ell}_t \right\rangle - D_{\phi_t}(q_{t+1}, q_t) \leq \left\langle q_t - \widetilde{q}_t, \widehat{\ell}_t \right\rangle - D_{\phi_t}(\widetilde{q}_t, q_t)$$

$$= \left\langle q_t - \widetilde{q}_t, \widehat{\ell}_t \right\rangle - \phi_t(\widetilde{q}_t) + \phi_t(q_t) - \langle \widetilde{q}_t - q_t, \nabla\phi_t(q_t) \rangle$$

$$= D_{\phi_t}(q_t, \widetilde{q}_t)$$

where the second equality uses the equality $\nabla\phi_t(q_t) - \nabla\phi_t(\widetilde{q}_t) = \widehat{\ell}_t$.

Moreover, the term $D_{\phi_t}(q_t, \widetilde{q}_t)$ can be bounded as:

$$D_{\phi_t}(q_t, \widetilde{q}_t) = \frac{1}{\eta_t} \sum_{s \neq s_L} \sum_{a \in A} \left( q_t(s, a) \ln\left(\frac{q_t(s, a)}{\widetilde{q}_t(s, a)}\right) - q_t(s, a) + \widetilde{q}_t(s, a) \right)$$

$$= \frac{1}{\eta_t} \sum_{s \neq s_L} \sum_{a \in A} q_t(s, a) \cdot \left( \eta_t \widehat{\ell}_t(s, a) - 1 + \exp\left(-\eta_t \cdot \widehat{\ell}_t(s, a)\right) \right)$$

$$\leq \eta_t \sum_{s \neq s_L} \sum_{a \in A} q_t(s, a) \widehat{\ell}_t(s, a)^2$$

where the last inequality follows from the facts $y - 1 + e^{-y} \leq y^2$ for $y > -1$ and $\eta_t \cdot \widehat{\ell}_t(s, a) \geq -1$ for all sate-action pairs.

On the other hand, the penalty term is at most

$$\phi_{T+1}(u) - \phi_1(q_1) - \sum_{t=1}^{T} (\phi_{t+1}(q_{t+1}) - \phi_t(q_{t+1})) \leq -\frac{\phi(q_1)}{\eta_1} - \sum_{t=1}^{T} \left(\frac{1}{\eta_{t+1}} - \frac{1}{\eta_t}\right) \phi(q_t),$$

since $\phi(u) \leq 0$. Moreover, note that for any valid occupancy measure $q$, it holds that

$$\phi(q) = \sum_{k=0}^{L-1} \sum_{s \in S_k} \sum_{a \in A} q(s, a) \geq -\sum_{k=0}^{L-1} \ln(|S_k||A|) \geq -L \ln(|S||A|).$$

Therefore, the penalty term is bounded by

$$-\frac{\phi(q_1)}{\eta_1} - \sum_{t=1}^{T} \left(\frac{1}{\eta_{t+1}} - \frac{1}{\eta_t}\right) \phi(q_t)$$

$$\leq L \ln(|S||A|) \cdot \left(\frac{1}{\eta_1} + \sum_{t=1}^{T} \left(\frac{1}{\eta_{t+1}} - \frac{1}{\eta_t}\right)\right) = \frac{L \ln(|S||A|)}{\eta_{T+1}}.$$

Finally, combining the bounds for the stability and penalty terms finishes the proof. $\qquad\square$

### A.3  Known Transition and Bandit Feedback: FTRL with Tsallis Entropy

In this section, we consider the bandit feedback setting with known transition. We use the following hybrid regularizer with learning rate $\eta_t = \gamma/\sqrt{t}$ for episode $t$:

$$\phi_t(q) = \frac{\phi_H(q)}{\eta_t} + \beta \underbrace{\sum_{s \neq s_L} \sum_{a \in A} \log \frac{1}{q(s, a)}}_{=\phi_L(q)}, \tag{25}$$

**Algorithm 3** Best-of-both-worlds for MDPs with Known Transition and Bandit Feedback

> **for** $t = 1$ **to** $T$ **do**
>> compute $q_t = \operatorname{argmin}_{q \in \Omega} \left\langle q, \sum_{\tau < t} \widehat{\ell}_\tau \right\rangle + \phi_t(q)$ where $\phi_t(q)$ is defined in Eq. (25).
>> execute policy $\pi_t$ where $\pi_t(a|s) = q_t(s,a)/q_t(s)$.
>> observe $(s_0, a_0, \ell_t(s_0, a_0)), \ldots, (s_{L-1}, a_{L-1}, \ell_t(s_{L-1}, a_{L-1}))$.
>> construct estimator $\widehat{\ell}_t$ such that: $\forall (s,a), \widehat{\ell}_t(s,a) = \frac{\ell_t(s,a)}{q_t(s,a)} \mathbb{I}\{s_{k(s)} = s, a_{k(s)} = a\}$.

where $\phi_L$ is a fixed log-barrier regularizer, and $\phi_H(q)$ is the $1/2$-Tsallis entorpy:

$$\phi_H(q) = - \sum_{s \neq s_L} \sum_{a \in A} \sqrt{q(s,a)}.$$

We present the pseudocode of our algorithm in Algorithm 3, and show the ensured guarantees in Theorem A.3.1, which is a more detailed version of Theorem 3.1. In particular, the adaptive regret bound Eq. (26) is a strict improvement of [Jin and Luo, 2020, Theorem 1] and leads to the best-of-both-worlds guarantee automatically. We emphasize that the key to achieve such a guarantees is the loss-shifting function defined in Eq. (6).

**Theorem A.3.1.** *With $\beta = 64L$ and $\gamma = 1$, Algorithm 3 ensures that $\operatorname{Reg}_T(\mathring{\pi})$ is bounded by*

$$\sum_{t=1}^{T} \widetilde{\mathcal{O}} \left( \min \left\{ \mathbb{E} \left[ B \sum_{s \neq s_L} \sum_{a \neq \pi(s)} \sqrt{\frac{q_t(s,a)}{t}} + D \sqrt{\sum_{s \neq s_L} \sum_{a \neq \pi(s)} \frac{q_t(s,a) + \mathring{q}(s,a)}{t}} \right], \sqrt{\frac{L|S||A|}{t}} \right\} \right) \tag{26}$$

*for any mapping $\pi : S \to A$, where $B = L^2$ and $D = \sqrt{L|S|}$. Therefore, the regret of Algorithm 3 is always bounded as $\operatorname{Reg}_T(\mathring{\pi}) = \widetilde{\mathcal{O}} \left( \sqrt{L|S||A|T} \right)$. Moreover, under Condition (21), $\operatorname{Reg}_T(\mathring{\pi})$ is bounded by $\mathcal{O} \left( U + \sqrt{UC} \right)$ where $U = \frac{L|S| \log T}{\Delta_{\text{MIN}}} + \sum_{s \neq s_L} \sum_{a \neq \pi^\star(a)} \frac{L^4 \log T}{\Delta(s,a)} + L|S||A| \log T$.*

*Proof.* By [Jin and Luo, 2020, Lemma 5], with a sufficiently large log-barrier component (in particular, $\beta = 64L$ suffices), the regret can be decomposed and bounded as:

$$\mathbb{E} \left[ \sum_{t=1}^{T} \left\langle q_t - \mathring{q}, \widehat{\ell}_t \right\rangle \right] \leq \underbrace{\sum_{t=1}^{T} \left( \frac{1}{\eta_t} - \frac{1}{\eta_{t-1}} \right) \mathbb{E} [\phi_H(\mathring{q}) - \phi_H(q_t)]}_{\text{PENALTY}} + \underbrace{8 \sum_{t=1}^{T} \eta_t \mathbb{E} \left[ \left\| \widehat{\ell}_t \right\|_{\nabla^{-2} \phi(q_t)}^2 \right]}_{\text{STABILITY}}$$
$$+ \mathcal{O} \left( L|S||A| \log T \right).$$

where $\mathring{q}$ is the occupancy measure of an deterministic optimal policy $\mathring{\pi} : S \to A$. Moreover, with the help of Corollary A.1.2, we can in fact bound $\operatorname{Reg}_T(\mathring{\pi})$ as

$$\operatorname{Reg}_T(\mathring{\pi}) \leq \underbrace{\sum_{t=1}^{T} \left( \frac{1}{\eta_t} - \frac{1}{\eta_{t-1}} \right) \mathbb{E} [\phi_H(\mathring{q}) - \phi_H(q_t)]}_{\text{PENALTY}} + \mathcal{O} \left( L|S||A| \log T \right)$$

$$+ \underbrace{8 \sum_{t=1}^{T} \eta_t \mathbb{E} \left[ \min \left\{ \mathbb{E}_t \left[ \left\| \widehat{\ell}_t \right\|_{\nabla^{-2} \phi(q_t)}^2 \right], \mathbb{E}_t \left[ \left\| \widehat{\ell}_t + g_t \right\|_{\nabla^{-2} \phi(q_t)}^2 \right] \right\} \right]}_{\text{STABILITY}}. \tag{27}$$

where $g_t$ is the specific loss-shifting function defined in Eq. (6). This is again because adding the loss-shifting function $g_t$ does not influence the outcomes of FTRL and thus in the analysis, one can decide whether to add $g_t$ or not for episode $t$ in hindsight to establish a tighter adaptive regret bound.

Before analyzing the stability term, we point out that $\phi_H(\mathring{q}) - \phi_H(q_t)$ can be bounded as

$$(\phi_H(\mathring{q}) - \phi_H(q_t)) \leq \sum_{s \neq s_L} \sum_{a \neq \pi(s)} \sqrt{q_t(s,a)} + 2 \sqrt{|S|L \sum_{s \neq s_L} \sum_{a \neq \pi(s)} q_t(s,a) + \mathring{q}(s,a)} \tag{28}$$

for any mapping $\pi : S \to A$ according to [Jin and Luo, 2020, Lemma 6] (take $\alpha$ in their lemma to be 0). On the other hand, we also have $\phi_H(\mathring{q}) - \phi_H(q_t) \leq -\phi_H(q_t) \leq \sqrt{L|S||A|}$ by the Cauchy-Schwarz inequality. Combining these two cases and the fact $\frac{1}{\eta_t} - \frac{1}{\eta_{t-1}} = \frac{1}{\gamma} \cdot (\sqrt{t} - \sqrt{t-1}) \leq \frac{1}{\gamma} \cdot \frac{1}{\sqrt{t}}$, the penalty term is bounded by

$$
\frac{1}{\gamma} \sum_{t=1}^{T} \mathbb{E} \left[ \min \left\{ \sqrt{\frac{L|S||A|}{t}}, \left( \sum_{s \neq s_L} \sum_{a \neq \pi(s)} \sqrt{\frac{q_t(s,a)}{t}} \right) + 2 \sqrt{|S|L \sum_{s \neq s_L} \sum_{a \neq \pi(s)} \frac{q_t(s,a) + \mathring{q}(s,a)}{t}} \right\} \right]
$$

We now bound the stability term. By direct calculation, we have

$$
\mathbb{E}_t \left[ \left\| \widehat{\ell}_t + g_t \right\|_{\nabla^{-2}\phi(q_t)}^2 \right] = \sum_{s \neq s_L} \sum_{a \in A} q_t(s,a)^{3/2} \mathbb{E}_t \left[ \left( \widehat{\ell}_t(s,a) + g_t(s,a) \right)^2 \right]
$$

$$
\leq 2L^2 \sum_{s \neq s_L} \sum_{a \in A} \sqrt{q_t(s,a)} \cdot (1 - \pi_t(a|s)), \tag{29}
$$

where the second line applies the properties of the loss-shifting function in Lemma A.1.3.

For any mapping $\pi : S \to A$, we can further bound Eq. (29) as

$$
2L^2 \sum_{s \neq s_L} \sum_{a \in A} \sqrt{q_t(s,a)} \cdot (1 - \pi_t(a|s))
$$

$$
\leq 2L^2 \sum_{s \neq s_L} \sum_{a \neq \pi(s)} \sqrt{q_t(s,a)} + 2L^2 \sum_{s \neq s_L} \sqrt{q_t(s)} \cdot \left( \sum_{a \neq \pi(s)} \pi_t(a|s) \right)
$$

$$
\leq 4L^2 \sum_{s \neq s_L} \sum_{a \neq \pi(s)} \sqrt{q_t(s,a)},
$$

where the third line follows from the fact $x \leq \sqrt{x}$ for $x \in [0,1]$.

Therefore, for any mapping $\pi : S \to A$, the stability term is bounded by

$$
\sum_{t=1}^{T} 8\eta_t \mathbb{E} \left[ \left\| \widehat{\ell}_t + g_t \right\|_{\nabla^{-2}\phi(q_t)}^2 \right] \leq 32L^2 \cdot \sum_{t=1}^{T} \eta_t \mathbb{E} \left[ \sum_{s \neq s_L} \sum_{a \neq \pi(s)} \sqrt{q_t(s,a)} \right]. \tag{30}
$$

On the other hand, without the loss-shifting function, the stability term is simultaneously bounded as

$$
\sum_{t=1}^{T} 8\eta_t \mathbb{E} \left[ \left\| \widehat{\ell}_t \right\|_{\nabla^{-2}\phi(q_t)}^2 \right] = \sum_{t=1}^{T} 8\eta_t \mathbb{E} \left[ \sum_{s \neq s_L} \sum_{a \in A} q_t(s,a)^{3/2} \cdot \widehat{\ell}_t(s,a)^2 \right]
$$

$$
\leq \sum_{t=1}^{T} 8\eta_t \mathbb{E} \left[ \sum_{s \neq s_L} \sum_{a \in A} \sqrt{q_t(s,a)} \right] \leq \sum_{t=1}^{T} 8\eta_t \sqrt{L|S||A|}. \qquad \text{(Cauchy-Schwarz inequality)}
$$

Plugging Eq. (28) and Eq. (30) into the Eq. (27) shows that Algorithm 3 ensures the following self-bounding regret bound for $\text{Reg}_T(\mathring{\pi})$:

$$
\frac{1}{\gamma} \sum_{t=1}^{T} \mathbb{E} \left[ \min \left\{ \sqrt{\frac{L|S||A|}{t}}, \left( \sum_{s \neq s_L} \sum_{a \neq \pi(s)} \sqrt{\frac{q_t(s,a)}{t}} \right) + 2 \sqrt{|S|L \sum_{s \neq s_L} \sum_{a \neq \pi(s)} \frac{q_t(s,a) + \mathring{q}(s,a)}{t}} \right\} \right]
$$

$$
32\gamma \cdot \sum_{t=1}^{T} \mathbb{E} \left[ \min \left\{ \sqrt{\frac{L|S||A|}{t}}, L^2 \sum_{s \neq s_L} \sum_{a \neq \pi(s)} \sqrt{\frac{q_t(s,a)}{t}} \right\} \right] + \mathcal{O} \left( L|S||A| \log T \right)
$$

$$
\tag{31}
$$

for any mapping $\pi : S \to A$. Picking $\gamma = 1$ and using $\min\{a,b\} + \min\{c,d\} \leq \min\{a+c, b+d\}$ proves Eq. (26).

The (optimal) worst-case bound $\text{Reg}_T(\mathring{\pi}) = \widetilde{\mathcal{O}}(\sqrt{L|S||A|T})$ can be obtained by using the second argument of the min operator in Eq. (26), while the logarithmic regret bound under Condition (21) is obtained by using the first argument of the min operator and the exact same reasoning as in [Jin and Luo, 2020, Appendix A.1]. □

We point out that with a different choice $\gamma = 1/L$, Algorithm 3 achieves a regret bound of $\text{Reg}_T(\mathring{\pi}) = \mathcal{O}\left(V + \sqrt{VC}\right)$ under Condition (21), where

$$V = \frac{L^3|S|\log T}{\Delta_{\text{MIN}}} + \sum_{s \neq s_L} \sum_{a \neq \pi^\star(a)} \frac{L^2 \log T}{\Delta(s,a)} + L|S||A|\log T$$

which matches the best existing regret bound in Simchowitz and Jamieson [2019]. This choice of $\gamma$ worsens the worst-case bound though.

# B   Best of Both Worlds for MDPs with Unknown Transition and Full Information

In this part, we will prove the best of both worlds results for the full-information setting. We present the bound for the adversarial world in Proposition B.1, and that for the stochastic world in Proposition B.2 (part of which is a restatement of Lemma 5.1). Together, they prove Theorem 4.1.1.

**Proposition B.1.** *Consider the decomposition* $\text{Reg}_T(\mathring{\pi}) = \mathbb{E}[\text{ERR}_1 + \text{ESTREG} + \text{ERR}_2]$ *stated in Eq.* (14). *Then, with* $\delta = \frac{1}{T^2}$ *Algorithm 1 ensures:*

- $\mathbb{E}[\text{ERR}_1] = \widetilde{\mathcal{O}}\left(L|S|\sqrt{|A|T} + L^3|S|^3|A|\right),$

- $\mathbb{E}[\text{ERR}_2] = \widetilde{\mathcal{O}}(1),$

- $\mathbb{E}[\text{ESTREG}] = \widetilde{\mathcal{O}}\left(L\sqrt{|S||A|T} + L^2|S|^2|A|^{\frac{3}{2}} + L^3|S||A|\right).$

**Proposition B.2.** *With* $\delta = \frac{1}{T^2}$, *Algorithm 1 ensures that* $\text{Reg}_T(\pi^\star)$ *is bounded as*

$$
\mathcal{O}\left(\mathbb{E}\left[\underbrace{\mathbb{G}_1\left(L^4|S|\ln T\right)}_{\text{ERRSUB}} + \underbrace{\mathbb{G}_2\left(L^4|S|\ln T\right)}_{\text{ERROPT}} + \underbrace{\mathbb{G}_3\left(L^4\ln T\right)}_{\text{OCCDIFF}} + \underbrace{\mathbb{G}_4\left(L^5|S||A|\ln T\ln(|S||A|)\right)}_{}\right]\right)
$$
$$
+ \mathcal{O}\left(L^4|S|^3|A|^2\ln^2 T\right),
$$

*where* $\mathbb{G}_1$-$\mathbb{G}_4$ *are defined in Definition D.2.1. Under Condition* (1), *this bound implies* $\text{Reg}_T(\pi^\star) = \mathcal{O}\left(U + \sqrt{UC} + V\right)$ *where*

$$
U = \frac{\left(L^6|S|^2 + L^5|S||A|\log(|S||A|)\right)\log T}{\Delta_{\text{MIN}}} + \sum_{s \neq s_L}\sum_{a \neq \pi^\star(s)}\frac{L^6|S|\log T}{\Delta(s,a)}, \quad V = L^4|S|^3|A|^2\ln^2 T.
$$

Before diving into the proof details, we first give formal definitions of several notations mentioned in Section 4 and Section 5 for the full-information setting. Through out this paper, we denote by $\mathcal{A}$ the event that $P \in \mathcal{P}_i$ for all $i$, which happens with probability at least $1 - 4\delta$ based on [Jin et al., 2020, Lemma 2]. We denote by $N$ the total number of epochs, and set $t_{N+1} = T + 1$ for convenience (recall that $t_i$ is the first episode for epoch $i$).

Then, recall the $\widehat{Q}_t^\pi$ and $\widehat{V}_t^\pi$ defined in Section 5, that is, the state-action and state value functions associated with the empirical transition $\bar{P}_{i(t)}$ and the adjusted loss $\widehat{\ell}_t$, formally defined as:

$$
\widehat{Q}_t^\pi(s,a) = \widehat{\ell}_t(s,a) + \sum_{s' \in S_{k(s)+1}}\bar{P}_{i(t)}(s'|s,a)\widehat{V}_t(s'), \quad \widehat{V}_t^\pi(s) = \sum_{a \in A}\pi(a|s)\widehat{Q}_t^\pi(s,a), \tag{32}
$$

and $\widehat{Q}_t^\pi(s_L,a) = 0$ for all $a$. Also recall that the notation $\widehat{Q}_t$ and $\widehat{V}_t$ used in the loss-shifting function are shorthands for $\widehat{Q}_t^{\pi_t}$ and $\widehat{V}_t^{\pi_t}$. Similarly, the true state-action and state value functions of episode $t$ are defined as:

$$
Q_t^\pi(s,a) = \ell_t(s,a) + \sum_{s' \in S_{k(s)+1}}P(s'|s,a)V_t(s'), \quad V_t^\pi(s) = \sum_{a \in A}\pi(a|s)Q_t^\pi(s,a), \tag{33}
$$

with $Q_t^\pi(s_L,a) = 0$ for all $a$. For notational convenience, we let $\iota = \frac{T|S||A|}{\delta}$ and assume that $\delta \in (0,1)$, and denote by $T_k$ the set of transition tuples at layer $k$, that is, $T_k = \{(s,a,s') \in S_k \times A \times S_{k+1}\}$.

## B.1   Optimism of Adjusted losses and Other Lemmas

First, we show that the adjusted loss $\widehat{\ell}_t$ defined in Eq. (11) ensures the optimism of the estimated state-action and state value functions as stated in Lemma B.1.1. As discussed in Section 5, this certain kind of optimism ensures that $\mathbb{E}[\text{ERR}_2]$ is bounded by a constant with a sufficiently small confidence parameter $\delta$.

**Lemma B.1.1.** *Using the notations in Eq. (32) and Eq. (33) and conditioning on the event $\mathcal{A}$, we have*
$$\widehat{Q}_t^\pi(s,a) \leq Q_t^\pi(s,a), \forall (s,a) \in S \times A, t \in [T].$$

*Proof.* We prove this result via a backward induction from layer $L$ to layer $0$.

**Base case:** for $s_L$, $\widehat{Q}_t^\pi(s,a) = Q_t^\pi(s,a) = 0$ holds always.

**Induction step:** Suppose $\widehat{Q}_t^\pi(s,a) \leq Q_t^\pi(s,a)$ holds for all the states $s$ with $k(s) > h$. Then, for any state $s$ in layer $h$, we have

$$\widehat{Q}_t^\pi(s,a) = \ell_t(s,a) + \sum_{s' \in S_{k(s)+1}} \bar{P}_{i(t)}(s'|s,a)\widehat{V}_t^\pi(s') - L \cdot B_t(s,a)$$

$$\leq \ell_t(s,a) + \sum_{s' \in S_{k(s)+1}} \bar{P}_{i(t)}(s'|s,a)V_t^\pi(s') - L \cdot B_t(s,a) \qquad \text{(Induction hypothesis)}$$

$$\leq \ell_t(s,a) + \sum_{s' \in S_{k(s)+1}} P(s'|s,a)V_t^\pi(s')$$

$$+ \sum_{s' \in S_{k(s)+1}} \left( \bar{P}_{i(t)}(s'|s,a) - P(s'|s,a) \right) V_t^\pi(s') - L \cdot B_{i(t)}(s,a)$$

$$= Q_t^\pi(s,a) + \sum_{s' \in S_{k(s)+1}} \left( \bar{P}_{i(t)}(s'|s,a) - P(s'|s,a) \right) V_t^\pi(s') - L \cdot B_{i(t)}(s,a)$$

where the first line follows from the definition of $\widehat{\ell}_t$.

Clearly, when $B_{i(t)}(s,a) = 1$, we have

$$\sum_{s' \in S_{k(s)+1}} \left( \bar{P}_{i(t)}(s'|s,a) - P(s'|s,a) \right) V_t^\pi(s') - L \cdot B_{i(t)}(s,a) \leq \sum_{s' \in S_{k(s)+1}} \bar{P}_{i(t)}(s'|s,a) \cdot L - L = 0$$

where the inequality follows from the fact $0 \leq V_t^\pi(s') \leq L$.

On the other hand, when $\sum_{s' \in S_{k(s)+1}} B_{i(t)}(s,a,s') = B_{i(t)}(s,a)$, we have

$$\sum_{s' \in S_{k(s)+1}} \left( \bar{P}_{i(t)}(s'|s,a) - P(s'|s,a) \right) V_t^\pi(s') - L \cdot B_{i(t)}(s,a)$$

$$\leq \sum_{s' \in S_{k(s)+1}} B_{i(t)}(s,a,s') \cdot L - L \cdot B_{i(t)}(s,a) = 0$$

where the second line uses the definition of event $\mathcal{A}$.

Combining these two cases shows that $\widehat{Q}_t^\pi(s,a) \leq Q_t^\pi(s,a)$ holds for all state-action pairs $(s,a)$ at layer $h$, finishing the induction. $\qquad\square$

Next, we analyze the estimated regret suffered within one epoch. With slightly abuse of notation, we denote by $\text{EstReg}_i(\pi)$ the difference between the total loss suffered within epoch $i$ and that of the fixed policy $\pi$ with respect to the empirical transition $\bar{P}_i$ and the adjusted losses within epoch $i$, that is,

$$\text{EstReg}_i(\pi) = \mathbb{E}\left[ \sum_{t=t_i}^{t_{i+1}-1} \left\langle q^{\bar{P}_i,\pi_t} - q^{\bar{P}_i,\pi}, \widehat{\ell}_t \right\rangle \right] = \mathbb{E}\left[ \sum_{t=t_i}^{t_{i+1}-1} \left\langle \widehat{q}_t - q^{\bar{P}_i,\pi}, \widehat{\ell}_t \right\rangle \right]. \qquad (34)$$

In addition, we let $\text{EstReg}_i = \max_\pi \text{EstReg}_i(\pi)$ be the maximum regret suffered within epoch $i$.

**Lemma B.1.2.** *For full-information feedback, Algorithm 1 ensures that $\text{EstReg}_i$ is bounded by $\mathcal{O}\left( L^3 \ln(|S||A|) \right)$ plus:*

$$\mathcal{O}\left( \mathbb{E}\left[ \sqrt{L \ln(|S||A|) \cdot \min\left\{ L^4 \sum_{t=t_i}^{t_{i+1}-1} \sum_{s \in S} \sum_{a \neq \pi(s)} \widehat{q}_t(s,a), \sum_{t=t_i}^{t_{i+1}-1} \sum_{s \neq s_L} \sum_{a \in A} \widehat{q}_t(s,a)\widehat{\ell}_t(s,a)^2 \right\}} \right] \right).$$
$$(35)$$

*Proof.* The proof follows the same steps as in that of Theorem A.2.1. Due to the invariant property, the loss of episode $t$ fed to FTRL can be seen as either $\widehat{\ell}_t(s,a)$ or $\widehat{Q}_t(s,a) - \widehat{V}_t(s,a)$. By the definition of $\eta_t$, we have both $\eta_t \widehat{\ell}_t(s,a) \geq -1$ and $\eta_t(\widehat{Q}_t(s,a) - \widehat{V}_t(s,a)) \geq -1$. Therefore, we can apply Lemma A.2.3 and bound $\text{EstReg}_i$ by

$$\mathbb{E}\left[\frac{L\ln(|S||A|)}{\eta_{t_{i+1}}} + \sum_{t=t_i}^{t_{i+1}-1} \eta_t \min\left\{\sum_{s\neq s_L}\sum_{a\in A} \widehat{q}_t(s,a)\left(\widehat{Q}_t(s,a) - \widehat{V}_t(s)\right)^2, \sum_{s\neq s_L}\sum_{a\in A}\widehat{q}_t(s,a)\ell_t(s,a)^2\right\}\right].$$

The tuning of $\eta_t$ makes sure that the above is further bounded by $\mathcal{O}\left(L^3\ln(|S||A|)\right)$ plus $\sqrt{L\ln(|S||A|)}$ multiplied with

$$\mathcal{O}\left(\mathbb{E}\left[\sqrt{\min\left\{\sum_{t=t_i}^{t_{i+1}-1}\sum_{s\neq s_L}\sum_{a\in A}\widehat{q}_t(s,a)\left(\widehat{Q}_t(s,a)-\widehat{V}_t(s)\right)^2, \sum_{t=t_i}^{t_{i+1}-1}\sum_{s\neq s_L}\sum_{a\in A}\widehat{q}_t(s,a)\ell_t(s,a)^2\right\}}\right]\right);$$

see the beginning of the proof of Theorem A.2.1 for the same reasoning. Finally, it remains to bound $\sum_{s\neq s_L}\sum_{a\in A}\widehat{q}_t(s,a)\left(\widehat{Q}_t(s,a) - \widehat{V}_t(s)\right)^2$ by $8L^4\sum_{s\neq s_L}\sum_{a\neq\pi(s)}\widehat{q}_t(s,a)$. This is again by the same reasoning as Eq. (23) and Eq. (24), except that $\widehat{Q}_t(s,a)$ now has a range in $[-L^2, L^2]$ which explains the extra $L^2$ factor. □

## B.2 Proof for the Adversarial World (Proposition B.1)

We analyze the regret based on the decomposition in Eq. (14) and consider bounding the terms $\mathbb{E}[\text{ERR}_1]$, $\mathbb{E}[\text{ERR}_2]$ and $\mathbb{E}[\text{EstReg}]$ separately.

**ERR$_1$** Following the similar idea of Jin et al. [2020], we decompose this term as:

$$\text{ERR}_1 = \sum_{t=1}^{T} V_t^{\pi_t}(s_0) - \widehat{V}_t^{\pi_t}(s_0) = \sum_{t=1}^{T}\langle q_t, \ell_t\rangle - \left\langle\widehat{q}_t, \widehat{\ell}_t\right\rangle$$

$$= \sum_{t=1}^{T}\langle q_t, \ell_t\rangle - \langle\widehat{q}_t, \ell_t\rangle + L\cdot\sum_{t=1}^{T}\left\langle\widehat{q}_t, B_{i(t)}\right\rangle$$

$$= \sum_{t=1}^{T}\langle q_t, \ell_t\rangle - \langle\widehat{q}_t, \ell_t\rangle + L\cdot\sum_{t=1}^{T}\left\langle q_t, B_{i(t)}\right\rangle + L\cdot\sum_{t=1}^{T}\left\langle\widehat{q}_t - q_t, B_{i(t)}\right\rangle$$

$$\leq \sum_{t=1}^{T}\sum_{s\neq s_L}\sum_{a\in A}|q_t(s,a) - \widehat{q}_t(s,a)| + L\cdot\sum_{t=1}^{T}\left\langle q_t, B_{i(t)}\right\rangle + L\cdot\sum_{t=1}^{T}\left\langle\widehat{q}_t - q_t, B_{i(t)}\right\rangle$$

where the last line follows from the fact $0 \leq \ell_t(s,a) \leq 1$. According to this decomposition, we next consider bounding the expectation of these three terms separately.

First, we focus on the second term:

$$\mathbb{E}\left[L\cdot\sum_{t=1}^{T}\left\langle q_t, B_{i(t)}\right\rangle\right]$$

$$\leq L\cdot\mathbb{E}\left[\sum_{k=0}^{L-1}\sum_{s\in S_k}\sum_{a\in A}\sum_{t=1}^{T}q_t(s,a)\left(2\sqrt{\frac{|S_{k(s)+1}|\ln\iota}{\max\{m_i(s,a),1\}}} + \frac{14|S_{k(s)+1}|\ln\iota}{3\max\{m_i(s,a),1\}}\right)\right]$$

$$= \mathcal{O}\left(L\cdot\sum_{k=0}^{L-1}\left(\sqrt{|S_k||S_{k+1}||A|T\ln\iota} + |S_{k(s)+1}||S_k||A|(2+\ln T)\ln\iota\right)\right)$$

$$\leq \mathcal{O}\left(L\sqrt{|A|T\ln\iota}\cdot\sum_{k=0}^{L-1}(|S_k|+|S_{k+1}|) + L|S|^2|A|\ln^2\iota\right)$$

$$= \mathcal{O}\left(L|S|\sqrt{|A|T\ln\iota} + L|S|^2|A|\ln^2\iota\right)$$

$$= \widetilde{\mathcal{O}}\left(L|S|\sqrt{|A|T} + L|S|^2|A|\right) \tag{36}$$

where the second line follows Lemma D.3.2, the third line follows from Lemma D.3.8 and the fourth line applies AM-GM inequality.

Then, for the first term, with the help from residual term $r_t$ defined in Definition D.3.9, we have

$$\mathbb{E}\left[\sum_{t=1}^{T}\sum_{s\neq s_L}\sum_{a\in A}|q_t(s,a) - \widehat{q}_t(s,a)|\right]$$

$$\leq \mathbb{E}\left[4\sum_{t=1}^{T}\sum_{s\neq s_L}\sum_{a\in A}\sum_{k=0}^{k(s)-1}\sum_{(u,v,w)\in T_k} q_t(u,v)\sqrt{\frac{P(w|u,v)\ln\iota}{\max\{m_{i(t)}(u,v),1\}}}q_t(s,a|w) + \sum_{t=1}^{T}\sum_{s\neq s_L}\sum_{a\in A}r_t(s,a)\right]$$

$$\leq \mathbb{E}\left[4L\cdot\sum_{t=1}^{T}\sum_{u\neq s_L}\sum_{v\in A}\sum_{w\in S_{k(u)+1}} q_t(u,v)\sqrt{\frac{P(w|u,v)\ln\iota}{\max\{m_{i(t)}(u,v),1\}}} + \sum_{t=1}^{T}\sum_{s\neq s_L}\sum_{a\in A}r_t(s,a)\right]$$

$$\leq \mathbb{E}\left[4L\cdot\sum_{t=1}^{T}\sum_{u\neq s_L}\sum_{v\in A} q_t(u,v)\sqrt{\frac{|S_{k(u)+1}|\ln\iota}{\max\{m_{i(t)}(u,v),1\}}} + \sum_{t=1}^{T}\sum_{s\neq s_L}\sum_{a\in A}r_t(s,a)\right]$$

$$= \mathcal{O}\left(L|S|\sqrt{|A|T\ln\iota} + L^2|S|^3|A|^2\ln^2\iota + \delta|S||A|T\right)$$

$$= \widetilde{\mathcal{O}}\left(L|S|\sqrt{|A|T} + L^2|S|^3|A|^2\right) \tag{37}$$

where the second line uses the bound of $|q_t(s,a) - \widehat{q}_t(s,a)|$ in Lemma D.3.10; the third line follows from the fact $\sum_{s\neq s_L}\sum_{a\in A}q_t(s,a|w)\leq L$; the forth line uses the Cauchy-Schwarz inequality; the fifth line follows the same argument in Eq. (36) and applies the expectation bound of residual terms in Lemma D.3.10; and the last line plugs in the value of $\delta = 1/T^2$.

For the last term, using the bound of $|\widehat{q}_t(s,a) - q_t(s,a)|$ in Lemma D.3.10, we arrive at

$$\mathbb{E}\left[L\cdot\sum_{t=1}^{T}\left\langle\widehat{q}_t - q_t, B_{i(t)}\right\rangle\right]$$

$$\leq \mathbb{E}\left[L\cdot\sum_{t=1}^{T}\sum_{s\neq s_L}\sum_{a\in A}B_{i(t)}(s,a)\cdot\left(4\sum_{k=0}^{k(s)-1}\sum_{(u,v,w)\in T_k} q_t(u,v)\sqrt{\frac{P(w|u,v)\ln\iota}{\max\{m_{i(t)}(u,v),1\}}}q_t(s,a|w) + r_t(s,a)\right)\right]$$

$$\leq \mathbb{E}\left[4L\cdot\sum_{t=1}^{T}\sum_{s\neq s_L}\sum_{a\in A}\sum_{s'\in S_{k(s)+1}} B_{i(t)}(s,a,s')\left(\sum_{k=0}^{k(s)-1}\sum_{(u,v,w)\in T_k} q_t(u,v)\sqrt{\frac{P(w|u,v)\ln\iota}{\max\{m_{i(t)}(u,v),1\}}}q_t(s,a|w)\right)\right]$$

$$+ \mathbb{E}\left[L\cdot\sum_{t=1}^{T}\sum_{s\neq s_L}\sum_{a\in A}r_t(s,a)\right],$$

where the last line follows from the fact $B_{i(t)}(s,a)\leq 1$.

According to the definition of the residual term in Definition D.3.9, we have

$$r_t(s,a)\geq \sum_{s'\in S_{k(s)+1}} B_{i(t)}(s,a,s')\cdot\left(\sum_{k=0}^{k(s)-1}\sum_{(u,v,w)\in T_k} q_t(u,v)\sqrt{\frac{P(w|u,v)\ln\iota}{\max\{m_{i(t)}(u,v),1\}}}\right)q_t(s,a|w)$$

(in particular, the second summand in the definition of $r_t(s,a)$ is an upper bound of the right-hand side above). Therefore, we have $\mathbb{E}\left[L\cdot\sum_{t=1}^{T}\left\langle\widehat{q}_t - q_t, B_{i(t)}\right\rangle\right]$ further bounded by

$$\mathbb{E}\left[(4L+L)\cdot\sum_{t=1}^{T}\sum_{s\neq s_L}\sum_{a\in A}r_t(s,a)\right] \leq \mathcal{O}\left(L^3|S|^3|A|^2\ln^2\iota + \delta\cdot L|S||A|T\right) = \widetilde{\mathcal{O}}\left(L^3|S|^3|A|^2\right)$$

$$\tag{38}$$

where the last inequality uses the expectation bound of residual terms in Lemma D.3.10.

Combining all bounds yields

$$\mathbb{E}\left[\mathrm{ERR}_1\right] = \widetilde{\mathcal{O}}\left(L|S|\sqrt{|A|T} + L^3|S|^3|A|\right).$$

$\mathrm{ERR}_2$ According to Lemma B.1.1, Lemma D.3.5, and the fact $\left|\widehat{V}_t^\pi(s)\right| \le L^2$, we have

$$\mathbb{E}\left[\mathrm{ERR}_2\right] = \mathbb{E}\left[\sum_{t=1}^T \widehat{V}_t^\pi(s_0) - V_t^\pi(s_0)\right] \le L^2 T \Pr[\mathcal{A}^c] \le 4L^2 T\delta = \widetilde{\mathcal{O}}(1).$$

$\mathrm{ESTREG}$ By Lemma B.1.2, we have $\mathbb{E}\left[\mathrm{ESTREG}\right]$ bounded as

$$\mathbb{E}\left[\sum_{i=1}^N \sum_{t=t_i}^{t_{i+1}-1} \left\langle \widehat{q}_t - q^{\bar{P}_i, \hat{\pi}}, \widehat{\ell}_t \right\rangle\right] \le \mathbb{E}\left[\sum_{i=1}^N \mathrm{EstReg}_i\right]$$

$$\le \mathbb{E}\left[\widetilde{\mathcal{O}}\left(\sum_{i=1}^N \sqrt{L \sum_{t=t_i}^{t_{i+1}-1} \sum_{s\in S}\sum_{a\in A} \widehat{q}_t(s,a)\widehat{\ell}_t(s,a)^2} + L^3\right)\right]$$

$$\le \widetilde{\mathcal{O}}\left(\sqrt{\mathbb{E}\left[L|S||A|\sum_{t=1}^T \sum_{s\in S}\sum_{a\in A} \widehat{q}_t(s,a)\widehat{\ell}_t(s,a)^2\right]} + L^3|S||A|\right)$$

where the last line follows from the fact $N \le 4|S||A|(\log T + 1)$ according to Lemma D.3.12 and uses Cauchy-Schwarz inequality.

Next, we continue to bound the following key term:

$$\mathbb{E}\left[\sum_{t=1}^T \sum_{s\in S}\sum_{a\in A} \widehat{q}_t(s,a)\widehat{\ell}_t(s,a)^2\right]$$

$$= \mathbb{E}\left[\sum_{t=1}^T \sum_{s\in S}\sum_{a\in A} \widehat{q}_t(s,a)\left(\ell_t(s,a) - L \cdot B_{i(t)}(s,a)\right)^2\right]$$

$$\le 2 \cdot \mathbb{E}\left[\sum_{t=1}^T \sum_{s\in S}\sum_{a\in A} \widehat{q}_t(s,a)\left(\ell_t(s,a)^2 + L^2 \cdot B_{i(t)}(s,a)^2\right)\right]$$

$$\le 2LT + 2L^2 \cdot \mathbb{E}\left[\sum_{t=1}^T \left\langle \widehat{q}_t, B_{i(t)}\right\rangle\right]$$

$$= 2LT + 2L \cdot \left(L \cdot \mathbb{E}\left[\sum_{t=1}^T \left\langle \widehat{q}_t - q_t, B_{i(t)}\right\rangle\right] + L \cdot \mathbb{E}\left[\sum_{t=1}^T \left\langle q_t, B_{i(t)}\right\rangle\right]\right),$$

where the third line uses $(x+y)^2 \le 2\left(x^2 + y^2\right)$ and the fourth line uses $B_{i(t)}(s,a) \le 1$. Moreover, in the previous analysis of the term $\mathrm{ERR}_1$, we bound the terms in the bracket with

$$\mathbb{E}\left[L \cdot \sum_{t=1}^T \left\langle q_t, B_{i(t)}\right\rangle\right] \le \widetilde{\mathcal{O}}\left(L|S|\sqrt{|A|T} + L|S|^2|A|\right), \qquad \text{(from Eq. (36))}$$

$$\mathbb{E}\left[L \cdot \sum_{t=1}^T \left\langle \widehat{q}_t - q_t, B_{i(t)}\right\rangle\right] \le \widetilde{\mathcal{O}}\left(L^3|S|^3|A|^2\right). \qquad \text{(from Eq. (38))}$$

Therefore, we have

$$\mathbb{E}\left[\sum_{t=1}^T \sum_{s\in S}\sum_{a\in A} \widehat{q}_t(s,a)\widehat{\ell}_t(s,a)^2\right] = \widetilde{\mathcal{O}}\left(LT + L|S|\sqrt{|A|T} + L^3|S|^3|A|^2\right) = \widetilde{\mathcal{O}}\left(LT + L^3|S|^3|A|^2\right),$$

which further proves

$$\mathbb{E}[\mathrm{ESTREG}] = \widetilde{\mathcal{O}}\left(L\sqrt{|S||A|T} + L^2|S|^2|A|^{\frac{3}{2}} + L^3|S||A|\right).$$

## B.3 Proof for the Stochastic World (Proposition B.2)

As discussed in Section 5, we decompose $\text{ERR}_1 + \text{ERR}_2$ as (see Corollary D.1.2):

$$\text{ERR}_1 + \text{ERR}_2 = \sum_{t=1}^{T} \sum_{s \neq s_L} \sum_{a \neq \pi^\star(s)} q_t(s,a) \widehat{E}_t^{\pi^\star}(s,a) \qquad \text{(ERRSUB)}$$

$$+ \sum_{t=1}^{T} \sum_{s \neq s_L} \sum_{a = \pi^\star(s)} \left( q_t(s,a) - q_t^\star(s,a) \right) \widehat{E}_t^{\pi^\star}(s,a) \qquad \text{(ERROPT)}$$

$$+ \sum_{t=1}^{T} \sum_{s \neq s_L} \sum_{a \in A} \left( q_t(s,a) - \widehat{q}_t(s,a) \right) \left( \widehat{Q}_t^{\pi^\star}(s,a) - \widehat{V}_t^{\pi^\star}(s) \right) \qquad \text{(OCCDIFF)}$$

$$+ \sum_{t=1}^{T} \sum_{s \neq s_L} \sum_{a \neq \pi^\star(s)} q_t^\star(s,a) \left( \widehat{V}_t^{\pi^\star}(s) - V_t^{\pi^\star}(s) \right) \qquad \text{(BIAS)}$$

where $\widehat{E}_t^{\pi}(s,a)$ is defined as:

$$\widehat{E}_t^{\pi}(s,a) = \ell_t(s,a) + \sum_{s' \in S_{k(s)+1}} P(s'|s,a) \widehat{V}_t^{\pi}(s') - \widehat{Q}_t^{\pi}(s,a).$$

Then, we proceed to bound each of the five terms: ERRSUB, ERROPT, OCCDIFF, BIAS, and ESTREG.

**ERRSUB** Conditioning on $\mathcal{A}$, we know that

$$\widehat{E}_t^{\pi^\star}(s,a) = LB_{i(t)}(s,a) + \sum_{s' \in S_{k(s)+1}} \left( P(s'|s,a) - \bar{P}_{i(t)}(s'|s,a) \right) \widehat{V}_t^{\pi^\star}(s')$$

$$\leq LB_{i(t)}(s,a) + L^2 \cdot \sum_{s' \in S_{k(s)+1}} B_{i(t)}(s,a,s')$$

$$\leq 4L^2 \cdot \sum_{s' \in S_{k(s)+1}} \left( \sqrt{\frac{\bar{P}_{i(t)}(s'|s,a) \ln \iota}{\max\left\{ m_{i(t)}(s,a), 1 \right\}}} + \frac{7 \ln \iota}{3 \max\left\{ m_{i(t)}(s,a), 1 \right\}} \right)$$

$$\leq 4L^2 \left( \sqrt{\frac{|S| \ln \iota}{\max\left\{ m_{i(t)}(s,a), 1 \right\}}} + \frac{7|S| \ln \iota}{3 \max\left\{ m_{i(t)}(s,a), 1 \right\}} \right),$$

where the second line follows from the event $\mathcal{A}$ and the fact $\left| \widehat{V}_t^{\pi}(s) \right| \leq L^2$, and the last line applies the Cauchy-Schwarz inequality.

Therefore, under event $\mathcal{A}$, ERRSUB can be bounded as:

$$\text{ERRSUB} \leq \sum_{t=1}^{T} \sum_{s \neq s_L} \sum_{a \neq \pi^\star(s)} q_t(s,a) \cdot 4L^2 \left( \sqrt{\frac{|S| \ln \iota}{\max\left\{ m_{i(t)}(s,a), 1 \right\}}} + \frac{7|S| \ln \iota}{3 \max\left\{ m_{i(t)}(s,a), 1 \right\}} \right)$$

$$\leq 4\mathbb{G}_1 \left( L^4 |S| \ln \iota \right) + \frac{28|S|L^2 \ln \iota}{3} \sum_{t=1}^{T} \sum_{s \neq s_L} \sum_{a \in A} \frac{q_t(s,a)}{3 \max\left\{ m_{i(t)}(s,a), 1 \right\}},$$

where the second line follows from the definition of $\mathbb{G}_1(\cdot)$ in Definition D.2.1.

With the help of Lemma D.3.5 and the fact $|\text{ERRSUB}| \leq L^3 T$, we have

$$\mathbb{E}\left[\text{ERRSUB}\right] \leq \mathcal{O}\left( L^3 T \delta + \mathbb{E}\left[ \mathbb{G}_1 \left( L^4 |S| \ln \iota \right) \right] \right) + \mathbb{E}\left[ \frac{28|S|L^2 \ln \iota}{3} \sum_{t=1}^{T} \sum_{s \neq s_L} \sum_{a \in A} \frac{q_t(s,a)}{3 \max\left\{ m_{i(t)}(s,a), 1 \right\}} \right]$$

$$= \mathcal{O}\left( \mathbb{E}\left[ \mathbb{G}_1 \left( L^4 |S| \ln \iota \right) \right] + L^2 |S|^2 |A| \ln^2 \iota \right), \qquad (39)$$

where the last line uses Lemma D.3.8.

**ERROPT** By the similar arguments above, we have ERROPT bounded by the following given event $\mathcal{A}$:

$$\text{ERROPT} \leq \sum_{t=1}^{T} \sum_{s \neq s_L} \sum_{a = \pi^\star(s)} (q_t(s,a) - q_t^\star(s,a)) \cdot 4L^2 \left( \sqrt{\frac{|S| \ln \iota}{\max\{m_{i(t)}(s,a), 1\}}} + \frac{7|S| \ln \iota}{3 \max\{m_{i(t)}(s,a), 1\}} \right).$$

Using the definition of $\mathbb{G}_2(\cdot)$ in Definition D.2.1 and Lemma D.3.5, we have

$$\mathbb{E}\left[\text{ERROPT}\right] \leq \mathcal{O}\left(L^3 T \delta + \mathbb{E}\left[\mathbb{G}_2\left(L^4 |S| \ln \iota\right)\right]\right) + \mathbb{E}\left[\frac{28|S| L^2 \ln \iota}{3} \sum_{t=1}^{T} \sum_{s \neq s_L} \sum_{a \in A} \frac{q_t(s,a)}{3 \max\{m_{i(t)}(s,a), 1\}}\right]$$

$$= \mathcal{O}\left(\mathbb{E}\left[\mathbb{G}_2\left(L^4 |S| \ln \iota\right)\right] + L^2 |S|^2 |A| \ln^2 \iota\right). \tag{40}$$

**OCCDIFF** First, we have

$$\text{OCCDIFF} = \sum_{t=1}^{T} \sum_{s \neq s_L} \sum_{a \in A} (q_t(s,a) - \widehat{q}_t(s,a)) \left(\widehat{Q}_t^{\pi^\star}(s,a) - \widehat{V}_t^{\pi^\star}(s)\right)$$

$$= \sum_{t=1}^{T} \sum_{s \neq s_L} \sum_{a \neq \pi^\star(s)} (q_t(s,a) - \widehat{q}_t(s,a)) \left(\widehat{Q}_t^{\pi^\star}(s,a) - \widehat{V}_t^{\pi^\star}(s)\right)$$

$$\leq 2L^2 \sum_{t=1}^{T} \sum_{s \neq s_L} \sum_{a \neq \pi^\star(s)} |q_t(s,a) - \widehat{q}_t(s,a)|,$$

where the second line follows from the fact $\widehat{V}_t^{\pi^\star}(s) = \widehat{Q}_t^{\pi^\star}(s,a)$ for all state-action pairs $(s,a)$ satisfying $a = \pi^\star(s)$, and the last line uses the fact $\widehat{Q}_t^{\pi^\star}(s,a) - \widehat{V}_t^{\pi^\star}(s) \leq 2L^2$ for all state-action pairs. With the help of the residual terms in Definition D.3.9 and Lemma D.3.10, we further bound OCCDIFF as

$$2L^2 \sum_{t=1}^{T} \sum_{s \neq s_L} \sum_{a \neq \pi^\star(s)} |q_t(s,a) - \widehat{q}_t(s,a)|$$

$$\leq 2L^2 \sum_{t=1}^{T} \sum_{s \neq s_L} \sum_{a \neq \pi^\star(s)} r_t(s,a) \tag{41}$$

$$+ 8L^2 \sum_{t=1}^{T} \sum_{s \neq s_L} \sum_{a \neq \pi^\star(s)} \sum_{k=0}^{k(s)-1} \sum_{(u,v,w) \in T_k} q_t(u,v) \sqrt{\frac{P(w|u,v) \ln \iota}{\max\{m_{i(t)}(u,v), 1\}}} q_t(s,a|w)$$

$$= \mathcal{O}\left(L^4 |S|^3 |A|^2 \ln^2 \iota + L^2 |S| |A| T \cdot \delta + \mathbb{G}_3(L^4 \ln \iota)\right)$$

where the last line is by the definition of $\mathbb{G}_3(\cdot)$ in Definition D.2.1. Therefore, we conclude

$$\mathbb{E}\left[\text{OCCDIFF}\right] \leq \mathcal{O}\left(L^4 |S|^3 |A|^2 \ln^2 \iota + \mathbb{E}\left[\mathbb{G}_3(L^4 \ln \iota)\right]\right). \tag{42}$$

**BIAS** Conditioning on the event $\mathcal{A}$, BIAS is nonpositive due to Lemma B.1.1. Then, by Lemma D.3.5, we bound the expectation of BIAS by

$$\mathbb{E}\left[\text{BIAS}\right] \leq 0 + \mathbb{E}\left[\mathbb{I}\{\mathcal{A}^c\}\right] \cdot L^3 T = \mathcal{O}(1). \tag{43}$$

**ESTREG** By the analysis of estimated regret in Lemma B.1.2, we have $\mathbb{E}\left[\text{ESTREG}\right]$ bounded by (with $C_{\text{ESTREG}} = L^5 |S| |A| \ln T \ln(|S||A|)$)

$$\mathcal{O}\left(\mathbb{E}\left[\sum_{i=1}^{N} \sqrt{L^5 \ln(|S||A|) \cdot \sum_{t=t_i}^{t_{i+1}-1} \sum_{s \in S} \sum_{a \neq \pi^\star(s)} \widehat{q}_t(s,a)} + L^3 \ln(|S||A|)\right]\right)$$

$$\leq \mathcal{O}\left(\mathbb{E}\left[\sqrt{C_{\text{ESTREG}} \cdot \sum_{t=1}^{T} \sum_{s \in S} \sum_{a \neq \pi^\star(s)} \widehat{q}_t(s,a)}\right] + L^3 |S| |A| \ln T \ln(|S||A|)\right)$$

$$\leq \mathcal{O}\left(\mathbb{E}\left[\sqrt{C_{\text{ESTREG}} \cdot \sum_{t=1}^{T}\sum_{s\in S}\sum_{a\neq\pi^\star(s)} q_t(s,a)}\right] + L^3|S||A|\ln T \ln(|S||A|)\right)$$

$$+ \mathcal{O}\left(\mathbb{E}\left[\sqrt{C_{\text{ESTREG}} \cdot \sum_{t=1}^{T}\sum_{s\in S}\sum_{a\neq\pi^\star(s)} |\widehat{q}_t(s,a) - q_t(s,a)|}\right]\right)$$

$$\leq \mathcal{O}\left(\mathbb{E}\left[\mathbb{G}_4(L^5|S||A|\ln T \ln(|S||A|))\right] + L^5|S||A|\ln T \ln(|S||A|)\right)$$

$$+ \mathcal{O}\left(\mathbb{E}\left[\sum_{t=1}^{T}\sum_{s\in S}\sum_{a\neq\pi^\star(s)} |\widehat{q}_t(s,a) - q_t(s,a)|\right]\right)$$

where the second line uses the Cauchy-Schwarz inequality and the fact $N \leq 4|S||A|(\log T + 1)$ according to Lemma D.3.12; the third line uses the fact that $\sqrt{x} \leq \sqrt{y} + \sqrt{|x-y|}$ for $x, y > 0$; the last line uses the definition of $\mathbb{G}_4(\cdot)$ in Definition D.2.1 and the AM-GM inequality.

Note that in the analysis of OCCDIFF (see Eq. (41)), we have already shown that

$$\sum_{t=1}^{T}\sum_{s\neq s_L}\sum_{a\neq\pi^\star(s)} |q_t(s,a) - \widehat{q}_t(s,a)| = \mathcal{O}\left(L^2|S|^3|A|^2\ln^2\iota + \mathbb{E}[\mathbb{G}_3(\ln\iota)]\right). \tag{44}$$

Combining everything, we have $\mathbb{E}\left[\text{ESTREG}\right]$ bounded by:

$$\mathcal{O}\left(\mathbb{E}\left[\mathbb{G}_4(L^5|S||A|\ln T \ln(|S||A|)) + \mathbb{G}_3(\ln\iota)\right] + L^3|S|^3|A|^2\ln^2\iota\right). \tag{45}$$

Finally, combining everything we have shown that Algorithm 1 ensures the following regret bound for $\text{Reg}_T(\pi^\star)$:

$$\mathcal{O}\left(\mathbb{E}\left[\mathbb{G}_1\left(L^4|S|\ln\iota\right)\right]\right) \qquad\qquad \text{(from Eq. (39) for ERRSUB)}$$
$$+ \mathcal{O}\left(\mathbb{E}\left[\mathbb{G}_2\left(L^4|S|\ln\iota\right)\right]\right) \qquad\qquad \text{(from Eq. (40) for ERROPT)}$$
$$+ \mathcal{O}\left(\mathbb{E}\left[\mathbb{G}_3\left(L^4\ln\iota\right)\right]\right) \qquad\qquad \text{(from Eq. (42) for OCCDIFF)}$$
$$+ \mathcal{O}\left(\mathbb{E}\left[\mathbb{G}_4\left(L^5|S||A|\ln(|S||A|)\ln T\right)\right]\right) \qquad \text{(from Eq. (45) for ESTREG)}$$
$$+ \mathcal{O}\left(L^4|S|^3|A|^2\ln^2\iota\right).$$

Now suppose that Condition (1) holds. For some universal constant $\kappa > 0$, $\text{Reg}_T(\pi^\star)$ is bounded as

$$\text{Reg}_T(\pi^\star) \leq \kappa \cdot \left(\mathbb{E}\left[\mathbb{G}_1\left(L^4|S|\ln\iota\right)\right] + \mathbb{E}\left[\mathbb{G}_2\left(L^4|S|\ln\iota\right)\right] + \mathbb{E}\left[\mathbb{G}_3\left(L^4\ln\iota\right)\right]\right)$$
$$+ \kappa \cdot \left(\mathbb{E}\left[\mathbb{G}_4\left(L^5|S||A|\ln(|S||A|)\ln T\right)\right]\right) + \kappa \cdot \left(L^4|S|^3|A|^2\ln^2\iota\right).$$

For any $z > 0$, by Lemma D.2.2, Lemma D.2.3, Lemma D.2.4 and Lemma D.2.5 with $\alpha = \beta = \frac{1}{12z\kappa}$ we have

$$\text{Reg}_T(\pi^\star) \leq \frac{\text{Reg}_T(\pi^\star) + C}{z}$$

$$+ 12z \cdot \left(\sum_{s\neq s_L}\sum_{a\neq\pi^\star(s)} \frac{8\kappa^2}{\Delta(s,a)}\right) \cdot \left(L^4|S|\ln\iota + L^6|S|\ln\iota\right)$$

$$+ 12z \cdot \left(\frac{\kappa^2}{\Delta_{\text{MIN}}}\right) \cdot \left(8L^5|S|\ln\iota + 8L^6|S|^2\ln\iota + \frac{L^5|S||A|\ln(|S||A|)\ln T}{4}\right)$$

$$+ \kappa \cdot \left(L^4|S|^3|A|^2\ln^2\iota\right)$$

$$\leq \frac{\text{Reg}_T(\pi^\star) + C}{z} + 288z\kappa^2 \cdot U + 2\kappa \cdot V,$$

where the last line uses the shorthands $U$ and $V$ defined in Proposition B.2.

Rearranging the terms arrive at:

$$
\begin{aligned}
\mathrm{Reg}_T(\pi^\star) &\le \frac{C}{z-1} + \frac{z^2}{z-1} \cdot 288\kappa^2 U + \frac{z}{z-1} \cdot 2\kappa \cdot V \\
&= \frac{C}{x} + \frac{(x+1)^2}{x} \cdot 288\kappa^2 U + \frac{x+1}{x} \cdot 2\kappa \cdot V \\
&= \frac{1}{x} \cdot \left(C + 288\kappa^2 U + 2\kappa \cdot V\right) + x \cdot 288\kappa^2 U + 2\kappa \cdot V + 576\kappa^2 U
\end{aligned}
$$

where we replace all $z$'s by $x = z - 1 > 0$ in the second line. Finally, by selecting the optimal $x$ to balance the first two terms, we have

$$
\begin{aligned}
\mathrm{Reg}_T(\pi^\star) &\le 2\sqrt{(C + 288\kappa^2 U + 2\kappa \cdot V) \cdot 288\kappa^2 U} + 2\kappa V + 576\kappa^2 U \\
&= \mathcal{O}\left(U + \sqrt{UC} + V\right),
\end{aligned}
$$

finishing the entire proof for Proposition B.2.

## C  Best of Both Worlds for MDPs with Unknown Transition and Bandit Feedback

In this section, we prove the best of both worlds results for the bandit setting with unknown transition. We present the bound for the adversarial world in Proposition C.1, and that for the stochastic world in Proposition C.2. Together, they prove Theorem 4.1.2.

**Proposition C.1.** *With $\delta = \frac{1}{T^3}$, Algorithm 1 ensures*

$$\text{Reg}_T(\mathring{\pi}) = \widetilde{\mathcal{O}}\left(\left(L + \sqrt{A}\right)|S|\sqrt{|A|T}\right).$$

**Proposition C.2.** *Suppose Condition (1) holds. With $\delta = \frac{1}{T^3}$, Algorithm 1 ensures that $\text{Reg}_T(\pi^\star)$ is bounded by $\mathcal{O}\left(U + \sqrt{CU} + V\right)$ where $V = L^6|S|^3|A|^3 \ln^2 T$ and $U$ is defined as*

$$U = \sum_{s \neq s_L} \sum_{a \neq \pi^\star(s)} \left[\frac{L^6|S|\ln T + L^4|S||A|\ln^2 T}{\Delta(s,a)}\right] + \left[\frac{L^6|S|^2 \ln T + L^3|S|^2|A|\ln^2 T}{\Delta_{\text{MIN}}}\right].$$

The analysis is similar to that for the full-information setting, except that we need to handle some bias terms caused by the new loss estimators. To this end, we denote by $\widetilde{\ell}_t$ the conditional expectation of $\widehat{\ell}_t$, that is

$$\widetilde{\ell}_t(s,a) = \mathbb{E}_t\left[\widehat{\ell}_t(s,a)\right] = \frac{q_t(s,a)}{u_t(s,a)} \cdot \ell_t(s,a) - L \cdot B_{i(t)}(s,a). \tag{46}$$

Then we define the following:

**Definition C.3.** *For any policy $\pi$, the estimated state-action and state value functions associated with $\bar{P}_{i(t)}$ and loss function $\widetilde{\ell}_t$ are defined as:*

$$\widetilde{Q}_t^\pi(s,a) = \widetilde{\ell}_t(s,a) + \sum_{s' \in S_{k(s)+1}} \bar{P}_{i(t)}(s'|s,a)\widetilde{V}_t^\pi(s'), \quad \forall(s,a) \in (S - \{s_L\}) \times A,$$

$$\widetilde{V}_t^\pi(s) = \sum_{a \in A} \pi(a|s)\widetilde{Q}_t^\pi(s,a), \quad \forall s \in S, \tag{47}$$

$$\widetilde{Q}_t^\pi(s_L,a) = 0, \quad \forall a \in A.$$

On the other hand, the true state-action and value functions are again defined as:

$$Q_t^\pi(s,a) = \ell_t(s,a) + \sum_{s' \in S_{k(s)+1}} P(s'|s,a)V_t^\pi(s'), \quad \forall(s,a) \in (S - \{s_L\}) \times A,$$

$$V_t^\pi(s) = \sum_{a \in A} \pi(a|s)Q_t^\pi(s,a), \quad \forall s \in S, \tag{48}$$

$$Q_t^\pi(s_L,a) = 0, \quad \forall a \in A.$$

where $P$ denotes the true transition function.

Besides the definition of event $\mathcal{A}$, we also define $\mathcal{A}_i$ to be the event $P \in \mathcal{P}_i$. Importantly, the value of $\mathbb{I}\{\mathcal{A}_i\}$ is only based on observations prior to epoch $i$. For notational convenience, we again let $\iota = \frac{T|S||A|}{\delta}$ and assume $\delta \in (0,1)$.

Similarly to the full-information setting, we decompose the regret against policy $\pi$, $\text{Reg}(\pi) = \mathbb{E}\left[\sum_{t=1}^T V_t^{\pi_t}(s_0) - V_t^\pi(s_0)\right]$, as

$$\underbrace{\mathbb{E}\left[\sum_{t=1}^T V_t^{\pi_t}(s_0) - \widetilde{V}_t^{\pi_t}(s_0)\right]}_{\text{ERR}_1} + \underbrace{\mathbb{E}\left[\sum_{t=1}^T \widetilde{V}_t^{\pi_t}(s_0) - \widetilde{V}_t^\pi(s_0)\right]}_{\text{ESTREG}} + \underbrace{\mathbb{E}\left[\sum_{t=1}^T \widetilde{V}_t^\pi(s_0) - V_t^\pi(s_0)\right]}_{\text{ERR}_2}. \tag{49}$$

Note that, the second term is exactly

$$\mathbb{E}[\text{ESTREG}] = \mathbb{E}\left[\sum_{t=1}^T \left\langle q^{\bar{P}_{i(t)},\pi_t} - q^{\bar{P}_{i(t)},\pi}, \widetilde{\ell}_t\right\rangle\right] = \mathbb{E}\left[\sum_{t=1}^T \left\langle q^{\bar{P}_{i(t)},\pi_t} - q^{\bar{P}_{i(t)},\pi}, \widehat{\ell}_t\right\rangle\right],$$

which is controlled by the FTRL process.

## C.1 Auxiliary Lemmas

First, we show the following optimism lemma.

**Lemma C.1.1.** *With the notations defined in Eq. (47) and Eq. (48), the following holds conditioning on event $\mathcal{A}$:*

$$\widetilde{Q}_t^\pi(s,a) \leq Q_t^\pi(s,a), \forall (s,a) \in S \times A, t \in [T].$$

*Specifically, we have*

$$\left\langle q^{\bar{P}_{i(t)},\pi}, \widetilde{\ell}_t \right\rangle = \widetilde{V}_t^\pi(s_0) \leq V_t^\pi(s_0) = \left\langle q^{P,\pi}, \ell_t \right\rangle.$$

*Proof.* We prove this result via a backward induction from layer $L$ to layer 0.

**Base case:** for $s_L$, $\widetilde{Q}_t^\pi(s,a) = Q_t^\pi(s,a) = 0$ holds always.

**Induction step:** Suppose $\widetilde{Q}_t^\pi(s,a) \leq Q_t^\pi(s,a)$ holds for all states $s$ with $k(s) > h$. Then, for any state $s$ with $k(s) = h$, we have

$$\widetilde{Q}_t^\pi(s,a) = \frac{q_t(s,a)}{u_t(s,a)} \cdot \ell_t(s,a) + \sum_{s' \in S_{k(s)+1}} \bar{P}_{i(t)}(s'|s,a)\widetilde{V}_t^\pi(s') - L \cdot B_{i(t)}(s,a) \quad \text{(Eq. (46))}$$

$$\leq \frac{q_t(s,a)}{u_t(s,a)} \cdot \ell_t(s,a) + \sum_{s' \in S_{k(s)+1}} \bar{P}_{i(t)}(s'|s,a)V_t^\pi(s') - L \cdot B_{i(t)}(s,a) \quad \text{(induction hypothesis)}$$

$$\leq \frac{q_t(s,a)}{u_t(s,a)} \cdot \ell_t(s,a) + \sum_{s' \in S_{k(s)+1}} P(s'|s,a)V_t^\pi(s')$$

$$+ \sum_{s' \in S_{k(s)+1}} \left( \bar{P}_{i(t)}(s'|s,a) - P(s'|s,a) \right) V_t^\pi(s') - L \cdot B_{i(t)}(s,a)$$

$$\leq \frac{q_t(s,a)}{u_t(s,a)} \cdot \ell_t(s,a) + \sum_{s' \in S_{k(s)+1}} P(s'|s,a)V_t^\pi(s')$$

$$\leq \ell_t(s,a) + \sum_{s' \in S_{k(s)+1}} P(s'|s,a)V^\pi(s') = Q_t^\pi(s,a),$$

where the forth step follows from the same arguments in Lemma B.1.1, and the last step holds since under event $\mathcal{A}$, we have $q_t(s,a) \leq u_t(s,a)$ by the definition of $u_t$. This finishes the induction.

$\square$

Next, we provide a sequence of boundedness results, useful for regret analysis.

**Lemma C.1.2** (Lower Bound of Upper Occupancy Bound)**.** *Algorithm 1 ensures $u_t(s) \geq \frac{1}{|S|t}$ for all $t$ and $s$.*

*Proof.* We prove by constructing a special transition function $\widehat{P}_{i(t)}$ within the confidence set $\mathcal{P}_{i(t)}$, which ensures $q^{\bar{P}_{i(t)},\pi_t}(s) \geq \frac{1}{|S|t}$ for all state-action pairs. Specifically, let $\widehat{P}_{i(t)}$ be such that

$$\widehat{P}_{i(t)}(s'|s,a) = \frac{1}{t} \cdot \frac{1}{|S_{k(s)+1}|} + \frac{t-1}{t} \cdot \bar{P}_{i(t)}(s'|s,a), \quad \forall (s,a,s') \in T_k, k < L.$$

Clearly, $\widehat{P}_{i(t)}(\cdot|s,a)$ is a valid transition distribution over $S_{k(s)+1}$ for all state-action pairs. Then, we prove that $\widehat{P}_{i(t)} \in \mathcal{P}_i$ by

$$\left| \widehat{P}_{i(t)}(s'|s,a) - \bar{P}_{i(t)}(s'|s,a) \right| = \frac{1}{t} \cdot \left| \bar{P}_{i(t)}(s'|s,a) - \frac{1}{|S_{k(s)+1}|} \right| \leq \frac{1}{t} \leq \frac{14 \ln\left( \frac{T|S||A|}{\delta} \right)}{3 \max\left\{ m_{i(t)}(s,a), 1 \right\}}$$

where the last inequality follows from the fact that $m_{i(t)}(s,a) \leq t$.

Then, for any state $s \neq s_0$, we have by the definition of occupancy measures

$$q^{\widehat{P}_{i(t)},\pi_t}(s) = \sum_{s' \in S_{k(s)-1}} \sum_{a' \in A} q^{\widehat{P}_{i(t)},\pi_t}(s',a') \cdot \widehat{P}_{i(t)}(s|s',a')$$

$$\geq \sum_{s' \in S_{k(s)-1}} \sum_{a' \in A} q^{\widehat{P}_{i(t)},\pi_t}(s',a') \cdot \frac{1}{|S_{k(s)}| \, t}$$

$$= \frac{1}{|S_{k(s)}| \, t} \geq \frac{1}{|S| t}$$

Clearly, for $s_0$ it holds that $q^{\widehat{P}_{i(t)},\pi_t}(s_0) = 1 \geq 1/|S|t$, which finishes the proof. $\qquad \square$

**Corollary C.1.3.** *Algorithm 1 ensures that, the adjusted loss $\widehat{\ell}_t$ defined in Eq. (11) for bandit-feedback is bounded as:*

$$\left| \widehat{\ell}_t(s,a) \right| \leq L + \frac{\mathbb{I}_t(s,a)}{q_t(s,a)} \cdot |S|t.$$

*Also, we have*

$$\mathbb{E}\left[ \frac{\mathbb{I}_t(s,a)}{q_t(s,a)} \middle| \mathcal{A}_{i(t)} \right] = \mathbb{E}\left[ \frac{\mathbb{I}_t(s,a)}{q_t(s,a)} \middle| \mathcal{A}_{i(t)}^c \right] = 1.$$

*Proof.* By Lemma C.1.2, we have

$$\left| \widehat{\ell}_t(s,a) \right| \leq \frac{\mathbb{I}_t(s,a)}{u_t(s) \cdot \pi_t(a|s)} + L \leq \frac{\mathbb{I}_t(s,a)}{q_t(s) \cdot \pi_t(a|s)} \cdot |S|t + L = L + \frac{\mathbb{I}_t(s,a)}{q_t(s,a)} \cdot |S|t,$$

where the first inequality follows from $B_i(s,a) \leq 1$ and $\ell_t(s,a) \leq 1$, and the second inequality uses Lemma C.1.2 and the fact $q_t(s) \leq 1$.

For the second statement, we have

$$\mathbb{E}\left[ \frac{\mathbb{I}_t(s,a)}{q_t(s,a)} \middle| \mathcal{A}_{i(t)} \right] = \mathbb{E}\left[ \mathbb{E}_t \left[ \frac{\mathbb{I}_t(s,a)}{q_t(s,a)} \right] \middle| \mathcal{A}_{i(t)} \right] = \mathbb{E}\left[ 1 \middle| \mathcal{A}_{i(t)} \right] = 1,$$

By the same arguments we can prove $\mathbb{E}\left[ \frac{\mathbb{I}_t(s,a)}{q_t(s,a)} \middle| \mathcal{A}_{i(t)}^c \right] = 1$ as well. $\qquad \square$

**Lemma C.1.4.** *Algorithm 1 ensures that, the expected adjusted loss $\widetilde{\ell}_t$ defined in Eq. (46) is bounded as:*

$$\left| \widetilde{\ell}_t(s,a) \right| \leq L + |S| \cdot t \leq 2|S| \cdot t, \quad \forall (s,a) \in S \times A, t \in [T].$$

*Proof.* By Eq. (46), we know that

$$\left| \widetilde{\ell}_t(s,a) \right| = \left| \frac{q_t(s,a)}{u_t(s,a)} \cdot \ell_t(s,a) - L \cdot B_{i(t)}(s,a) \right| \leq \frac{q_t(s)}{u_t(s)} + L \leq L + |S| \cdot t$$

where the last inequality follows from Lemma C.1.2. Combining with the fact $|S| \geq L$ finishes the proof. $\qquad \square$

**Corollary C.1.5.** *Algorithm 1 ensures that, the estimated state-action value functions defined in Eq. (47) are bounded as:*

$$\left| \widetilde{Q}_t^\pi(s,a) \right| \leq 2L|S|t, \quad \forall (s,a) \in S \times A, t \in [T].$$

*Proof.* This is directly by Lemma C.1.4 and the definition of $\widetilde{Q}_t^\pi(s,a)$. $\qquad \square$

Next, we analyze the estimated regret in each epoch. Reloading the notation from the full-information setting, we define

$$\text{EstReg}_i(\pi) = \mathbb{E}\left[ \sum_{t=t_i}^{t_{i+1}-1} \left\langle q^{\bar{P}_i,\pi_t} - q^{\bar{P}_i,\pi}, \widehat{\ell}_t \right\rangle \right] = \mathbb{E}\left[ \sum_{t=t_i}^{t_{i+1}-1} \left\langle \widehat{q}_t - q^{\bar{P}_i,\pi}, \widehat{\ell}_t \right\rangle \right].$$

**Lemma C.1.6.** *With $\beta = 128L^4$, for any epoch $i$, Algorithm 1 ensures*

$$\text{EstReg}_i(\pi) \leq \mathcal{O}\left(\mathbb{E}\left[\sum_{t=t_i}^{t_{i+1}-1} \eta_t \cdot \left(\sqrt{L|S||A|} + L^2 \sum_{s\neq s_L} \sum_{a\in A} \widehat{q}_t(s,a) \cdot B_{i(t)}(s,a)^2\right)\right]\right) \tag{50}$$
$$+ \mathcal{O}\left(L^4|S||A|\log T + \delta \cdot \mathbb{E}\left[L|S|T\left(t_{i+1}-t_i\right)\right]\right),$$

*for any policy $\pi$, and simultaneously*

$$\text{EstReg}_i(\pi) \leq \mathcal{O}\left(\mathbb{E}\left[\sqrt{L|S|} \sum_{t=t_i}^{t_{i+1}-1} \eta_t \cdot \sqrt{\sum_{s\neq s_L} \sum_{a\neq\pi(s)} \widehat{q}_t(s,a)}\right]\right)$$
$$+ \mathcal{O}\left(L^2 \cdot \mathbb{E}\left[\sum_{t=t_i}^{t_{i+1}-1} \eta_t \cdot \sum_{s\neq s_L} \sum_{a\neq\pi(s)} \sqrt{\widehat{q}_t(s,a)}\right]\right) \tag{51}$$
$$+ \mathcal{O}\left(L^4|A| \cdot \mathbb{E}\left[\sum_{t=t_i}^{t_{i+1}-1} \eta_t \cdot \sum_{s\neq s_L} \sum_{a\in A} \widehat{q}_t(s,a) \cdot B_{i(t)}(s,a)^2\right]\right)$$
$$+ \mathcal{O}\left(L^4|S||A|\log T + \delta \cdot \mathbb{E}\left[L|S|T\left(t_{i+1}-t_i\right)\right]\right),$$

*for any deterministic policy $\pi : S \to A$.*

*Proof.* The proof is largely based on that of Theorem A.3.1, but with some careful treatments based one whether $\mathcal{A}_i$ holds or not. Let $q = q^{\bar{P}_i,\pi}$ be the occupancy measure we want to compete against. When $\mathcal{A}_i$ does not hold, we first derive the following naive bound on $\sum_{t=t_i}^{t_{i+1}-1}\left\langle \widehat{q}_t - q, \widehat{\ell}_t \right\rangle$:

$$\sum_{t=t_i}^{t_{i+1}-1}\left\langle \widehat{q}_t - q, \widehat{\ell}_t \right\rangle \leq \sum_{t=t_i}^{t_{i+1}-1} \sum_{s\neq s_L} \sum_{a\in A} (\widehat{q}_t(s,a) + q(s,a)) \cdot \left|\widehat{\ell}_t(s,a)\right|$$
$$\leq \sum_{t=t_i}^{t_{i+1}-1} \sum_{s\neq s_L} \sum_{a\in A} (\widehat{q}_t(s,a) + q(s,a)) \cdot \left(L + \frac{\mathbb{I}_t(s,a)}{u_t(s,a)} \cdot |S|t\right) \qquad \text{(Corollary C.1.3)}$$
$$\leq 2L^2 \cdot (t_{i+1}-t_i) + |S|T \cdot \sum_{t=t_i}^{t_{i+1}-1} \sum_{s\neq s_L} \sum_{a\in A} (\widehat{q}_t(s,a) + q(s,a)) \cdot \frac{\mathbb{I}_t(s,a)}{q_t(s,a)}.$$

Therefore, we have the conditional expectation $\mathbb{E}\left[\sum_{t=t_i}^{t_{i+1}-1}\left\langle \widehat{q}_t - q, \widehat{\ell}_t \right\rangle \Big| \mathcal{A}_i^c\right]$ bounded by

$$\mathbb{E}\left[2L^2 \cdot (t_{i+1}-t_i) + |S|t \cdot \sum_{t=t_i}^{t_{i+1}-1} \sum_{s\neq s_L} \sum_{a\in A} (\widehat{q}_t(s,a) + q(s,a)) \cdot \frac{\mathbb{I}_t(s,a)}{q_t(s,a)} \Bigg| \mathcal{A}_i^c\right]$$
$$\leq \mathbb{E}\left[\left(2L^2 + 2L|S|T\right) \cdot (t_{i+1}-t_i) \big| \mathcal{A}_i^c\right] \qquad \text{(Corollary C.1.3)}$$
$$\leq \mathcal{O}\left(\mathbb{E}\left[L|S|T \cdot (t_{i+1}-t_i)| \mathcal{A}_i^c\right]\right).$$

Next, we condition on event $\mathcal{A}_i$. In this case, by the same argument as [Jin and Luo, 2020, Lemma 5] and also our loss-shifting technique, Algorithm 1 with $\beta = 128L^4$ ensures that $\sum_{t=t_i}^{t_{i+1}-1}\left\langle \widehat{q}_t - q, \widehat{\ell}_t \right\rangle$ is bounded by

$$\mathcal{O}\left(L^4|S||A|\log T\right) + \sum_{t=t_i+1}^{t_{i+1}-1} \left(\frac{1}{\eta_t} - \frac{1}{\eta_{t-1}}\right)(\phi_H(q) - \phi_H(\widehat{q}_t))$$
$$+ 8 \sum_{t=t_i}^{t_{i+1}-1} \eta_t \min\left\{ \sum_{s\neq s_L} \sum_{a\in A} \widehat{q}_t(s,a)^{3/2}\left(\widehat{Q}_t(s,a) - \widehat{V}_t(s)\right)^2, \sum_{s\neq s_L} \sum_{a\in A} \widehat{q}_t(s,a)^{3/2}\widehat{\ell}_t(s,a)^2 \right\} \tag{52}$$

where $\phi_H(q) = -\sum_{s\neq s_L}\sum_{a\in A}\sqrt{q(s,a)}$, and $\widehat{Q}_t$ and $\widehat{V}_t$ are state-action and state value functions associated with the loss estimator $\widehat{\ell}_t$ and the empirical transition $\bar{P}_{i(t)}$:

$$\widehat{Q}_t(s,a) = \widehat{\ell}_t(s,a) + \sum_{s'\in S_{k(s)+1}} \bar{P}_{i(t)}(s'|s,a)\widehat{V}_t(s'), \quad \widehat{V}_t(s) = \sum_{a\in A}\pi_t(a|s)\widehat{Q}_t(s,a).$$

Below, we discuss how to proceed from here to prove Eq. (50) and Eq. (51) respectively.

**Proving Eq. (50)** In this case, we take the second argument of the min operator from Eq. (52) and bound $\phi_H(q) - \phi_H(\widehat{q}_t) \leq \sum_{s\neq s_L}\sum_{a\in A}\sqrt{\widehat{q}_t(s,a)}$ trivially by $\sqrt{L|S||A|}$ using Cauchy-Schwarz inequality, leading to

$$\sum_{t=t_i}^{t_{i+1}-1} \left\langle \widehat{q}_t - q, \widehat{\ell}_t \right\rangle$$

$$\leq \mathcal{O}\left(L|S||A|\log T\right) + \sqrt{L|S||A|}\cdot \sum_{t=t_i}^{t_{i+1}-1}\eta_t + 8\sum_{t=t_i}^{t_{i+1}-1}\eta_t\cdot\sum_{s\neq s_L}\sum_{a\in A}\widehat{q}_t(s,a)^{3/2}\widehat{\ell}_t(s,a)^2$$

$$\left(\frac{1}{\eta_t} - \frac{1}{\eta_{t-1}} \leq \eta_t \text{ since } \frac{1}{\eta_t} = \sqrt{t-t_i+1}\right)$$

$$\leq \mathcal{O}\left(L|S||A|\log T\right) + 2\sqrt{L|S||A|}\cdot\sum_{t=t_i}^{t_{i+1}-1}\eta_t + 16\sum_{t=t_i}^{t_{i+1}-1}\eta_t\cdot\sum_{s\neq s_L}\sum_{a\in A}\frac{\widehat{q}_t(s,a)^{3/2}\cdot\mathbb{I}_t(s,a)}{u_t(s,a)^2}$$

$$+ 16L^2\sum_{t=t_i}^{t_{i+1}-1}\eta_t\cdot\sum_{s\neq s_L}\sum_{a\in A}\widehat{q}_t(s,a)^{3/2}\cdot B_{i(t)}(s,a)^2$$

$$\leq \mathcal{O}\left(L|S||A|\log T\right) + 2\sqrt{L|S||A|}\cdot\sum_{t=t_i}^{t_{i+1}-1}\eta_t + 16\sum_{t=t_i}^{t_{i+1}-1}\eta_t\cdot\sum_{s\neq s_L}\sum_{a\in A}\frac{\sqrt{\widehat{q}_t(s,a)}\cdot\mathbb{I}_t(s,a)}{q_t(s,a)}$$

$$+ 16L^2\sum_{t=t_i}^{t_{i+1}-1}\eta_t\cdot\sum_{s\neq s_L}\sum_{a\in A}\widehat{q}_t(s,a)\cdot B_{i(t)}(s,a)^2$$

where the second step follows from the definition of $\widehat{\ell}_t$ in Eq. (11) and the last step follows from the fact $\widehat{q}_t(s,a) \leq u_t(s,a)$ and $q_t(s,a) \leq u_t(s,a)$ since $\bar{P}_i, P \in \mathcal{P}_i$ according to event $\mathcal{A}_i$.

Therefore, by Lemma D.3.6 we have for any policy $\pi$ that,

$$\mathbb{E}\left[\text{EstReg}_i(\pi)\right] \leq \mathbb{E}\left[2\sqrt{L|S||A|}\cdot\sum_{t=t_i}^{t_{i+1}-1}\eta_t + 16\sum_{t=t_i}^{t_{i+1}-1}\eta_t\cdot\sum_{s\neq s_L}\sum_{a\in A}\frac{\sqrt{\widehat{q}_t(s,a)}\cdot\mathbb{I}_t(s,a)}{q_t(s,a)}\right]$$

$$+ \mathbb{E}\left[16L^2\sum_{t=t_i}^{t_{i+1}-1}\eta_t\cdot\sum_{s\neq s_L}\sum_{a\in A}\widehat{q}_t(s,a)\cdot B_{i(t)}(s,a)^2\right]$$

$$+ \mathcal{O}\left(L^4|S||A|\log T + \delta\cdot\mathbb{E}\left[L|S|T\left(t_{i+1}-t_i\right)\right]\right)$$

$$\leq \mathcal{O}\left(\mathbb{E}\left[\sqrt{L|S||A|}\cdot\sum_{t=t_i}^{t_{i+1}-1}\eta_t\right] + \mathbb{E}\left[L^2\sum_{t=t_i}^{t_{i+1}-1}\eta_t\cdot\sum_{s\neq s_L}\sum_{a\in A}\widehat{q}_t(s,a)\cdot B_{i(t)}(s,a)^2\right]\right)$$

$$+ \mathcal{O}\left(L^4|S||A|\log T + \delta\cdot\mathbb{E}\left[L|S|T\left(t_{i+1}-t_i\right)\right]\right)$$

where the second step takes the conditional expectation of $\mathbb{I}_t(s,a)$ and applies the Cauchy-Schwarz inequality to get $\sum_{s\neq s_L}\sum_{a\in A}\sqrt{\widehat{q}_t(s,a)} \leq \sqrt{L|S||A|}$. This finishes the proof of Eq. (50).

**Proving Eq. (51)** In this case, recall that $\pi$ is a deterministic policy, so that

$$\phi_H(q) - \phi_H(\widehat{q}_t) = \sum_{s\neq s_L}\sqrt{\widehat{q}_t(s)}\left(\sum_{a\in A}\sqrt{\pi_t(a|s)} - 1\right) + \sum_{s\neq s_L}\left(\sqrt{\widehat{q}_t(s)} - \sqrt{q(s)}\right).$$

Using [Jin and Luo, 2020, Lemma 16] to bound the first term (take $\alpha$ in their lemma to be 0), and [Jin and Luo, 2020, Lemma 19] to bound the second, we obtain

$$\phi_H(q) - \phi_H(\widehat{q}_t) = \sum_{s \neq s_L} \sum_{a \neq \pi(s)} \sqrt{\widehat{q}_t(s,a)} + \sqrt{L|S| \sum_{s \neq s_L} \sum_{a \neq \pi(s)} \widehat{q}_t(s,a)}.$$

Therefore, taking the first argument of the min operator from Eq. (52) and using $\frac{1}{\eta_t} - \frac{1}{\eta_{t-1}} \leq \eta_t$ again, we arrive at

$$
\begin{aligned}
\sum_{t=t_i}^{t_{i+1}-1} \left\langle \widehat{q}_t - q, \widehat{\ell}_t \right\rangle \leq {}& \sqrt{L|S|} \sum_{t=t_i}^{t_{i+1}-1} \eta_t \cdot \sqrt{\sum_{s \neq s_L} \sum_{a \neq \pi(s)} \widehat{q}_t(s,a)} \\
& + \sum_{t=t_i}^{t_{i+1}-1} \eta_t \cdot \sum_{s \neq s_L} \sum_{a \neq \pi(s)} \sqrt{\widehat{q}_t(s,a)} \\
& + 8 \sum_{t=t_i}^{t_{i+1}-1} \eta_t \cdot \sum_{s \neq s_L} \sum_{a \in A} \widehat{q}_t(s,a)^{3/2} \left( \widehat{Q}_t(s,a) - \widehat{V}_t(s) \right)^2 \\
& + \mathcal{O}\left( L^4 |S||A| \log T \right).
\end{aligned}
\tag{53}
$$

Finally, we apply Lemma C.1.7 to bound the term $\sum_{s \neq s_L} \sum_{a \in A} \widehat{q}_t(s,a)^{3/2} \left( \widehat{Q}_t(s,a) - \widehat{V}_t(s) \right)^2$, and use Lemma D.3.6 again to take expectation and arrive at Eq. (51) (with the help of Eq. (54)). $\qquad\square$

**Lemma C.1.7.** *Under event $\mathcal{A}$, we have for any $t$,*

$$
\begin{aligned}
& \sum_{s \neq s_L} \sum_{a \in A} \widehat{q}_t(s,a)^{3/2} \left( \widehat{Q}_t(s,a) - \widehat{V}_t(s) \right)^2 \\
& \leq 4L^4 |A| \sum_{s' \neq s_L} \sum_{a' \in A} \widehat{q}_t(s',a') \cdot B_{i(t)}(s',a')^2 + \sum_{s \neq s_L} \sum_{a \in A} \sqrt{\widehat{q}_t(s,a)} \cdot (O_t(s,a) + W_t(s,a))
\end{aligned}
$$

*where*

$$O_t(s,a) = 4L \cdot (1 - \pi_t(a|s)) \sum_{k=k(s)}^{L-1} \sum_{s' \in S_k} \sum_{a' \in A} \widehat{q}_t(s',a'|s,a) \frac{\mathbb{I}_t(s',a')}{q_t(s',a')},$$

$$W_t(s,a) = 4L \cdot \sum_{b \neq a} \pi_t(b|s) \sum_{k=k(s)}^{L-1} \sum_{s' \in S_k} \sum_{a' \in A} \widehat{q}_t(s',a'|s,b) \frac{\mathbb{I}_t(s',a')}{q_t(s',a')},$$

*and $\widehat{q}_t(s',a'|s,a)$ is the probability of visiting $(s',a')$ starting from $(s,a)$ under $\pi_t$ and $\bar{P}_{i(t)}$. Moreover, we have*

$$\mathbb{E}_t \left[ \sum_{s \neq s_L} \sum_{a \in A} \sqrt{\widehat{q}_t(s,a)} \cdot (O_t(s,a) + W_t(s,a)) \right] \leq 16L^2 \sum_{s \neq s_L} \sum_{a \neq \pi(s)} \sqrt{\widehat{q}_t(s,a)}, \tag{54}$$

*for any mapping $\pi : S \to A$.*

*Proof.* First, $\left( \widehat{Q}_t(s,a) - \widehat{V}_t(s) \right)^2$ is bounded by

$$
\begin{aligned}
\left( \widehat{Q}_t(s,a) - \widehat{V}_t(s) \right)^2 &= \left( (1 - \pi_t(a|s)) \widehat{Q}_t(s,a) - \left( \sum_{b \neq a} \pi_t(b|s) \widehat{Q}_t(s,b) \right) \right)^2 \\
&\leq 2 (1 - \pi_t(a|s))^2 \widehat{Q}_t(s,a)^2 + 2 \left( \sum_{b \neq a} \pi_t(b|s) \widehat{Q}_t(s,b) \right)^2.
\end{aligned}
$$

Following the same idea of Lemma A.1.3, the first term can be bounded as

$$(1 - \pi_t(a|s))^2 \, \widehat{Q}_t(s,a)^2$$

$$= (1 - \pi_t(a|s))^2 \left( \sum_{k=k(s)}^{L-1} \sum_{s' \in S_k} \sum_{a' \in A} \widehat{q}_t(s',a'|s,a) \widehat{\ell}_t(s',a') \right)^2$$

$$\leq 2 \left(1 - \pi_t(a|s)\right)^2 \left( \sum_{k=k(s)}^{L-1} \sum_{s' \in S_k} \sum_{a' \in A} \widehat{q}_t(s',a'|s,a) \frac{\mathbb{I}_t(s',a')}{u_t(s',a')} \cdot \ell_t(s',a') \right)^2$$

$$+ 2 \left(1 - \pi_t(a|s)\right)^2 \left( \sum_{k=k(s)}^{L-1} \sum_{s' \in S_k} \sum_{a' \in A} \widehat{q}_t(s',a'|s,a) \cdot L \cdot B_{i(t)}(s',a') \right)^2$$

$$\leq 2L \cdot (1 - \pi_t(a|s))^2 \sum_{k=k(s)}^{L-1} \sum_{s' \in S_k} \sum_{a' \in A} \widehat{q}_t(s',a'|s,a)^2 \cdot \frac{\mathbb{I}_t(s',a')}{u_t(s',a')^2}$$

$$+ 2L^3 \left(1 - \pi_t(a|s)\right)^2 \sum_{k=k(s)}^{L-1} \sum_{s' \in S_k} \sum_{a' \in A} \widehat{q}_t(s',a'|s,a) \cdot B_{i(t)}(s',a')^2 \tag{55}$$

where the equality follows from the definition of $\widehat{Q}_t$; the first inequality uses the fact $(x+y)^2 \leq 2(x^2 + y^2)$; the second inequality applies the Cauchy-Schwarz inequality with the facts $\mathbb{I}_t(s,a)\mathbb{I}_t(s',a') = 0$ for $(s,a) \neq (s',a')$ and $\sum_{k=k(s)}^{L-1} \sum_{s' \in S_k} \sum_{a' \in A} \widehat{q}_t(s',a'|s,a) \leq L$.

By the same arguments, the second term is bounded as

$$\left( \sum_{b \neq a} \pi_t(b|s) \widehat{Q}_t(s,b) \right)^2$$

$$= \left( \sum_{k=k(s)}^{L-1} \sum_{s' \in S_k} \sum_{a' \in A} \left( \sum_{b \neq a} \pi_t(b|s) \widehat{q}_t(s',a'|s,b) \right) \widehat{\ell}_t(s,a) \right)^2$$

$$\leq 2L \cdot \sum_{k=k(s)}^{L-1} \sum_{s' \in S_k} \sum_{a' \in A} \left( \sum_{b \neq a} \pi_t(b|s) \cdot \widehat{q}_t(s',a'|s,b) \right)^2 \cdot \frac{\mathbb{I}_t(s',a')}{u_t(s',a')^2}$$

$$+ 2L^3 \sum_{k=k(s)}^{L-1} \sum_{s' \in S_k} \sum_{a' \in A} \left( \sum_{b \neq a} \pi_t(b|s) \cdot \widehat{q}_t(s',a'|s,b) \right) \cdot B_{i(t)}(s',a')^2, \tag{56}$$

where in the last step we use $\sum_{k=k(s)}^{L-1} \sum_{s' \in S_k} \sum_{a' \in A} \left( \sum_{b \neq a} \pi_t(b|s) \cdot \widehat{q}_t(s',a'|s,b) \right) \leq L$ (after applying Cauchy-Schwarz).

Combining Eq. (55) and Eq. (56), we show that $\widehat{q}_t(s,a) \left( \widehat{Q}_t(s,a) - \widehat{V}_t(s) \right)^2$ can be bounded as

$$\widehat{q}_t(s,a) \left( \widehat{Q}_t(s,a) - \widehat{V}_t(s) \right)^2$$

$$\leq 4L \cdot \widehat{q}_t(s,a) \left(1 - \pi_t(a|s)\right)^2 \sum_{k=k(s)}^{L-1} \sum_{s' \in S_k} \sum_{a' \in A} \widehat{q}_t(s',a'|s,a)^2 \cdot \frac{\mathbb{I}_t(s',a')}{u_t(s',a')^2}$$

$$+ 4L \cdot \widehat{q}_t(s,a) \sum_{k=k(s)}^{L-1} \sum_{s' \in S_k} \sum_{a' \in A} \left( \sum_{b \neq a} \pi_t(b|s) \cdot \widehat{q}_t(s',a'|s,b) \right)^2 \cdot \frac{\mathbb{I}_t(s',a')}{u_t(s',a')^2}$$

$$+ 4L^3 \widehat{q}_t(s,a) \left(1 - \pi_t(a|s)\right)^2 \sum_{k=k(s)}^{L-1} \sum_{s' \in S_k} \sum_{a' \in A} \widehat{q}_t(s',a'|s,a) \cdot B_{i(t)}(s',a')^2$$

$$+ 4L^3 \widehat{q}_t(s,a) \sum_{k=k(s)}^{L-1} \sum_{s' \in S_k} \sum_{a' \in A} \left( \sum_{b \neq a} \pi_t(b|s) \cdot \widehat{q}_t(s', a'|s, b) \right) \cdot B_{i(t)}(s', a')^2.$$

Moreover, we have the summation of the first two terms bounded as

$$4L \cdot \widehat{q}_t(s,a) \left( 1 - \pi_t(a|s) \right)^2 \sum_{k=k(s)}^{L-1} \sum_{s' \in S_k} \sum_{a' \in A} \widehat{q}_t(s', a'|s, a)^2 \cdot \frac{\mathbb{I}_t(s', a')}{u_t(s', a')^2}$$

$$+ 4L \cdot \widehat{q}_t(s,a) \sum_{k=k(s)}^{L-1} \sum_{s' \in S_k} \sum_{a' \in A} \left( \sum_{b \neq a} \pi_t(b|s) \cdot \widehat{q}_t(s', a'|s, b) \right)^2 \cdot \frac{\mathbb{I}_t(s', a')}{u_t(s', a')^2}$$

$$\leq 4L \cdot \left( 1 - \pi_t(a|s) \right)^2 \sum_{k=k(s)}^{L-1} \sum_{s' \in S_k} \sum_{a' \in A} \frac{\widehat{q}_t(s,a) \widehat{q}_t(s', a'|s, a)}{u_t(s', a')} \cdot \widehat{q}_t(s', a'|s, a) \frac{\mathbb{I}_t(s', a')}{q_t(s', a')}$$

$$+ 4L \cdot \sum_{k=k(s)}^{L-1} \sum_{s' \in S_k} \sum_{a' \in A} \frac{\sum_{b \neq a} \widehat{q}_t(s,b) \cdot \widehat{q}_t(s', a'|s, b)}{u_t(s', a')} \cdot \left( \sum_{b \neq a} \pi_t(b|s) \cdot \widehat{q}_t(s', a'|s, b) \frac{\mathbb{I}_t(s', a')}{q_t(s', a')} \right)$$

$$\leq O_t(s,a) + W_t(s,a)$$

where we use $q_t(s', a') \leq u_t(s', a')$ due to event $\mathcal{A}_i$ in the first step and $\sum_{a \in A} \widehat{q}_t(s,a) \widehat{q}_t(s', a'|s, a) \leq \widehat{q}_t(s', a') \leq u_t(s', a')$ in the second step to bound the fractions by 1.

On the other hand, the summation of the other two terms is bounded as

$$4L^3 \widehat{q}_t(s,a) \left( 1 - \pi_t(a|s) \right)^2 \sum_{k=k(s)}^{L-1} \sum_{s' \in S_k} \sum_{a' \in A} \widehat{q}_t(s', a'|s, a) \cdot B_{i(t)}(s', a')^2$$

$$+ 4L^3 \widehat{q}_t(s,a) \sum_{k=k(s)}^{L-1} \sum_{s' \in S_k} \sum_{a' \in A} \left( \sum_{b \neq a} \pi_t(b|s) \cdot \widehat{q}_t(s', a'|s, b) \right) \cdot B_{i(t)}(s', a')^2$$

$$\leq 4L^3 \widehat{q}_t(s) \sum_{k=k(s)}^{L-1} \sum_{s' \in S_k} \sum_{a' \in A} \left( \widehat{q}_t(s', a'|s, a) \pi_t(a|s) + \sum_{b \neq a} \pi_t(b|s) \cdot \widehat{q}_t(s', a'|s, b) \right) \cdot B_{i(t)}(s', a')^2$$

$$= 4L^3 \sum_{k=k(s)}^{L-1} \sum_{s' \in S_k} \sum_{a' \in A} \widehat{q}_t(s', a'|s) \widehat{q}_t(s) \cdot B_{i(t)}(s', a')^2.$$

Note that, taking the summation of the last bound over all state-action pairs yields

$$4L^3 \sum_{s \neq s_L} \sum_{a \in A} \sum_{k=k(s)}^{L-1} \sum_{s' \in S_k} \sum_{a' \in A} \widehat{q}_t(s', a'|s) \widehat{q}_t(s) \cdot B_{i(t)}(s', a')^2$$

$$= 4L^3 |A| \sum_{s' \neq s_L} \sum_{a' \in A} \left( \sum_{k=0}^{k(s')-1} \sum_{s \in S_k} \widehat{q}_t(s', a'|s) \widehat{q}_t(s) \right) \cdot B_{i(t)}(s', a')^2$$

$$\leq 4L^4 |A| \sum_{s' \neq s_L} \sum_{a' \in A} \widehat{q}_t(s', a') \cdot B_{i(t)}(s', a')^2.$$

Therefore, combining everything, we have shown:

$$\sum_{s \neq s_L} \sum_{a \neq \pi(s)} \widehat{q}_t(s,a)^{3/2} \left( \widehat{Q}_t(s,a) - \widehat{V}_t(s) \right)^2$$

$$\leq 4L^4 |A| \sum_{s' \neq s_L} \sum_{a' \in A} \widehat{q}_t(s', a') \cdot B_{i(t)}(s', a')^2 + \sum_{s \neq s_L} \sum_{a \neq \pi(s)} \sqrt{\widehat{q}_t(s,a)} \cdot \left( O_t(s,a) + W_t(s,a) \right),$$

proving the first statement of the lemma.

To prove the second statement, we first show

$$\mathbb{E}_t\left[O_t(s,a) + W_t(s,a)\right] = 4L\left(1 - \pi_t(a|s)\right) \cdot \sum_{k=k(s)}^{L-1} \sum_{s' \in S_k} \sum_{a' \in A} \widehat{q}_t(s',a'|s,a)$$

$$+ 4L \cdot \sum_{k=k(s)}^{L-1} \sum_{s' \in S_k} \sum_{a' \in A} \left(\sum_{b \neq a} \pi_t(b|s) \cdot \widehat{q}_t(s',a'|s,b)\right)$$

$$= 4L\left(1 - \pi_t(a|s)\right) \sum_{k=k(s)}^{L-1} 1 + 4L \cdot \sum_{k=k(s)}^{L-1}\left(1 - \pi_t(a|s)\right)$$

$$\leq 8L^2\left(1 - \pi_t(a|s)\right),$$

and therefore

$$\mathbb{E}_t\left[\sum_{s \neq s_L} \sum_{a \in A} \sqrt{\widehat{q}_t(s,a)} \cdot \left(O_t(s,a) + W_t(s,a)\right)\right]$$

$$\leq 8L^2 \sum_{s \neq s_L} \sum_{a \in A} \sqrt{\widehat{q}_t(s,a)}\left(1 - \pi_t(a|s)\right)$$

$$\leq 8L^2 \sum_{s \neq s_L} \sum_{a \neq \pi(s)} \sqrt{\widehat{q}_t(s,a)} + 8L^2 \sum_{s \neq s_L} \sqrt{\widehat{q}_t(s)}\left(1 - \pi_t(\pi(s)|s)\right)$$

$$\leq 16L^2 \sum_{s \neq s_L} \sum_{a \neq \pi(s)} \sqrt{\widehat{q}_t(s,a)},$$

which proves Eq. (54). □

Note that both Eq. (50) and Eq. (51) contain a term related to $\sum_{s \neq s_L} \sum_{a \in A} \widehat{q}_t(s,a) \cdot B_{i(t)}(s,a)^2$. Below, we show that when summed over $t$, this is only logarithmic in $T$.

**Lemma C.1.8.** *Algorithm 1 ensures the following:*

$$\mathbb{E}\left[\sum_{t=1}^{T} \sum_{s \neq s_L} \sum_{a \in A} \widehat{q}_t(s,a) \cdot B_{i(t)}(s,a)^2\right] = \mathcal{O}\left(L^2|S|^3|A|^2 \ln^2 \iota + |S||A|T \cdot \delta\right). \qquad (57)$$

*Proof.* By Lemma D.3.2, we know that

$$B_i(s,a)^2 \leq \left(2\sqrt{\frac{|S_{k(s)+1}| \ln \iota}{\max\{m_i(s,a),1\}}} + \frac{14|S_{k(s)+1}| \ln \iota}{3 \max\{m_i(s,a),1\}}\right)^2$$

$$\leq \mathcal{O}\left(\frac{|S_{k(s)+1}| \ln \iota}{\max\{m_i(s,a),1\}} + \frac{|S_{k(s)+1}|^2 \ln^2 \iota}{\max\{m_i(s,a),1\}^2}\right).$$

Then, we have

$$\mathbb{E}\left[\sum_{t=1}^{T} \sum_{s \neq s_L} \sum_{a \in A} \widehat{q}_t(s,a) \cdot B_{i(t)}(s,a)^2\right]$$

$$= \mathbb{E}\left[\sum_{t=1}^{T} \sum_{s \neq s_L} \sum_{a \in A} \left(\widehat{q}_t(s,a) - q_t(s,a)\right) \cdot B_{i(t)}(s,a)^2\right] + \mathbb{E}\left[\sum_{t=1}^{T} \sum_{s \neq s_L} \sum_{a \in A} q_t(s,a) \cdot B_{i(t)}(s,a)^2\right]$$

$$\leq \mathbb{E}\left[\sum_{t=1}^{T} \sum_{s \neq s_L} \sum_{a \in A} r_t(s,a)\right]$$

$$+ \mathbb{E}\left[4\sum_{t=1}^{T}\sum_{s\neq s_L}\sum_{a\in A}\sum_{k=0}^{k(s)-1}\sum_{(u,v,w)\in T_k} q_t(u,v)\sqrt{\frac{P(w|u,v)\ln\left(\frac{T|S||A|}{\delta}\right)}{\max\left\{m_{i(t)}(u,v),1\right\}}}q_t(s,a|w)\cdot B_{i(t)}(s,a)\right]$$

$$+ \mathcal{O}\left(\mathbb{E}\left[\sum_{t=1}^{T}\sum_{s\neq s_L}\sum_{a\in A}q_t(s,a)\cdot\left(\frac{|S_{k(s)+1}|\ln\iota}{\max\left\{m_i(s,a),1\right\}}+\frac{|S_{k(s)+1}|^2\ln^2\iota}{\max\left\{m_i(s,a),1\right\}^2}\right)\right]\right)$$

$$\leq \mathcal{O}\left(\mathbb{E}\left[\sum_{t=1}^{T}\sum_{s\neq s_L}\sum_{a\in A}r_t(s,a)\right]\right)$$

$$+ \mathcal{O}\left(\mathbb{E}\left[\sum_{t=1}^{T}\sum_{s\neq s_L}\sum_{a\in A}q_t(s,a)\cdot\left(\frac{|S_{k(s)+1}|\ln\iota}{\max\left\{m_i(s,a),1\right\}}+\frac{|S_{k(s)+1}|^2\ln^2\iota}{\max\left\{m_i(s,a),1\right\}^2}\right)\right]\right)$$

where the first inequality uses Lemma D.3.10 and $B_i(s,a)\in[0,1]$, and the last inequality follows from the observation that, the second term in the previous line is bounded by $\sum_{t=1}^{T}\sum_{s\neq s_L}\sum_{a\in A}r_t(s,a)$ according to the definition of residual terms in Definition D.3.9.

Finally, applying Lemma D.3.10 and Lemma D.3.8, we have

$$\mathbb{E}\left[\sum_{t=1}^{T}\sum_{s\neq s_L}\sum_{a\in A}\widehat{q}_t(s,a)\cdot B_{i(t)}(s,a)^2\right]$$

$$= \mathcal{O}\left(L^2|S|^3|A|^2\ln^2\iota+|S||A|T\cdot\delta\right)+\mathcal{O}\left(\sum_{k=0}^{L-1}\left(|S_{k+1}||S_k||A|\ln T\ln\iota+|S_{k(s)+1}|^2|S_k||A|\ln^2\iota\right)\right)$$

$$= \mathcal{O}\left(L^2|S|^3|A|^2\ln^2\iota+|S||A|T\cdot\delta\right),$$

which completes the proof. $\qquad\square$

Finally, we provide a lemma regarding the learning rates.

**Lemma C.1.9** (Learning Rates). *According to the design of the learning rate $\eta_t=\frac{1}{\sqrt{t-t_{i(t)}+1}}$, the following inequalities hold:*

$$\sum_{t=1}^{T}\eta_t^2 \leq \mathcal{O}\left(|S||A|\log^2 T\right), \tag{58}$$

$$\sum_{t=1}^{T}\eta_t \leq \mathcal{O}\left(\sqrt{|S||A|T\log T}\right). \tag{59}$$

*Proof.* By direct calculation, we have

$$\sum_{t=t_i}^{t_{i+1}-1}\eta_t^2 = \sum_{n=1}^{t_{i+1}-t_i}\frac{1}{n} \leq 2\int_1^{t_{i+1}-t_i+1}\frac{1}{x}dx = 2\ln\left(t_{i+1}-t_i+1\right) \leq \mathcal{O}\left(\log T\right).$$

Combining the inequality with the fact that the total number of epochs $N$ is at most $4|S||A|\left(\log T+1\right)$ (Lemma D.3.12) finishes the proof of Eq. (58). Following the similar idea, we have

$$\sum_{t=t_i}^{t_{i+1}-1}\eta_t = \sum_{n=1}^{t_{i+1}-t_i}\frac{1}{\sqrt{n}} \leq \int_0^{t_{i+1}-t_i}\frac{1}{\sqrt{x}}dx \leq 2\sqrt{t_{i+1}-t_i}.$$

Taking the summation over $N$ epochs and applying the Cauchy-Schwarz inequality yields Eq. (59).

$\qquad\square$

## C.2 Proof for the Adversarial World (Proposition C.1)

Recall the regret decomposition in Eq. (49):

$$\mathbb{E}\left[\underbrace{\sum_{t=1}^{T} V_t^{\pi_t}(s_0) - \widetilde{V}_t^{\pi_t}(s_0)}_{\text{ERR}_1}\right] + \mathbb{E}\left[\underbrace{\sum_{t=1}^{T} \widetilde{V}_t^{\pi_t}(s_0) - \widetilde{V}_t^{\pi}(s_0)}_{\text{ESTREG}}\right] + \mathbb{E}\left[\underbrace{\sum_{t=1}^{T} \widetilde{V}_t^{\pi}(s_0) - V_t^{\pi}(s_0)}_{\text{ERR}_2}\right].$$

We bound each of them separately below.

**ERR₁**  Similarly to the proof for the full-information feedback setting, we have

$$\text{ERR}_1 = \sum_{t=1}^{T} \langle q_t, \ell_t \rangle - \left\langle \widehat{q}_t, \widetilde{\ell}_t \right\rangle$$

$$= \sum_{t=1}^{T} \sum_{s \neq s_L} \sum_{a \in A} \frac{\ell_t(s,a)\widehat{q}_t(s,a)}{u_t(s,a)} \cdot (u_t(s,a) - q_t(s,a)) + \sum_{t=1}^{T} \langle q_t - \widehat{q}_t, \ell_t \rangle + L \cdot \sum_{t=1}^{T} \langle \widehat{q}_t, B_{i(t)} \rangle$$

where the last two terms have been shown to be at most $\widetilde{\mathcal{O}}\left(L|S|\sqrt{|A|T} + L^3|S|^3|A|\right)$ according to the analysis of ERR₁ in Appendix B.2 (see Eq. (36), Eq. (37) and Eq. (38)).

Then, we bound the first term as

$$\mathbb{E}\left[\sum_{t=1}^{T} \sum_{s \neq s_L} \sum_{a \in A} \frac{\ell_t(s,a)\widehat{q}_t(s,a)}{u_t(s,a)} \cdot (u_t(s,a) - q_t(s,a))\right]$$

$$\leq \mathbb{E}\left[\sum_{t=1}^{T} \sum_{s \neq s_L} \sum_{a \in A} |u_t(s,a) - q_t(s,a)|\right] \qquad\qquad (\widehat{q}_t(s,a) \leq u_t(s,a))$$

$$\leq \mathbb{E}\left[4\sum_{t=1}^{T} \sum_{s \neq s_L} \sum_{a \in A} r_t(s,a) + 16\sum_{t=1}^{T} \sum_{s \neq s_L} \sum_{a \in A} \sum_{k=0}^{k(s)-1} \sum_{(u,v,w) \in T_k} q_t(u,v)\sqrt{\frac{P(w|u,v)\ln\iota}{\max\{m_{i(t)}(u,v), 1\}}} q_t(s,a|w)\right]$$
$$\text{(Corollary D.3.11)}$$

$$\leq \mathcal{O}\left(L^2|S|^3|A|^2\ln^2\iota + |S||A|T \cdot \delta\right) + 4L \cdot \mathbb{E}\left[\sum_{t=1}^{T} \sum_{u \neq s_L} \sum_{v \in A} q_t(u,v)\sqrt{\frac{|S_{k(u)+1}|\ln\iota}{\max\{m_{i(t)}(u,v), 1\}}}\right]$$
$$\text{(Lemma D.3.10 and Cauchy-Schwarz)}$$

$$\leq \mathcal{O}\left(L^2|S|^3|A|^2\ln^2\iota + |S||A|T \cdot \delta + L \cdot \sum_{k=0}^{L-1} \sqrt{|S_k| \cdot |S_{k+1}| |A|T\ln\iota}\right) \qquad \text{(Lemma D.3.8)}$$

$$= \mathcal{O}\left(L|S|\sqrt{|A|T\ln\iota} + L^2|S|^3|A|^2\ln^2\iota + |S||A|T \cdot \delta\right).$$

Combining the bounds together, we have $\mathbb{E}[\text{ERR}_1]$ bounded by:

$$\mathbb{E}[\text{ERR}_1] = \widetilde{\mathcal{O}}\left(L|S|\sqrt{|A|T} + L^3|S|^3|A|^2\right).$$

**ERR₂**  Following the same idea of bounding ERR₂, by Lemma C.1.1 and Lemma D.3.5, we have the expectation of ERR₂ bounded as

$$\mathbb{E}[\text{ERR}_2] \leq \delta \cdot 3L|S|T^2 + 0 = \mathcal{O}\left(L|S|T^2 \cdot \delta\right) = \mathcal{O}(1).$$

**ESTREG**  According to Eq. (50) of Lemma C.1.6, we have

$$\text{EstReg}(\mathring{\pi}) = \mathbb{E}\left[\sum_{t=1}^{T} \left\langle \widehat{q}_t - q^{\bar{P}_{i(t)}, \mathring{\pi}}, \widehat{\ell}_t \right\rangle\right] = \mathbb{E}\left[\sum_{i=1}^{N} \text{EstReg}_i(\mathring{\pi})\right]$$

$$\leq \mathcal{O}\left(\mathbb{E}\left[\sum_{i=1}^{N}\sum_{t=t_i}^{t_{i+1}-1}\eta_t\sqrt{L|S||A|}\right] + \mathbb{E}\left[L^2 \cdot \sum_{t=1}^{T}\sum_{s\neq s_L}\sum_{a\in A}\widehat{q}_t(s,a)\cdot B_{i(t)}(s,a)^2\right]\right)$$

$$+ \mathcal{O}\left(L^4|S|^2|A|^2\ln^2 T + \delta L|S|T^2\right)$$

$$\leq \widetilde{\mathcal{O}}\left(\mathbb{E}\left[\sum_{t=1}^{T}\eta_t \cdot \sqrt{L|S||A|}\right] + L^4|S|^3|A|^2\ln^2\iota\right) \qquad \text{(Lemma C.1.8)}$$

$$\leq \widetilde{\mathcal{O}}\left(|S||A|\sqrt{LT} + L^4|S|^3|A|^2\right). \qquad \text{(Eq. (59))}$$

Finally, we combine the bounds of $\text{ERR}_1$, $\text{ERR}_2$ and $\text{EstReg}$ as:

$$\text{Reg}_T(\mathring{\pi}) = \widetilde{\mathcal{O}}\left(L|S|\sqrt{|A|T} + |S||A|\sqrt{LT} + L^4|S|^3|A|^2\right),$$

finishing the proof.

### C.3 Proof for the Stochastic World (Proposition C.2)

Similarly to the proof of Proposition B.2, we decompose $\text{ERR}_1$ and $\text{ERR}_2$ jointly into four terms $\text{ErrSub}$, $\text{ErrOpt}$, $\text{OccDiff}$ and $\text{Bias}$:

$$\text{ERR}_1 + \text{ERR}_2 = \sum_{t=1}^{T}\sum_{s\neq s_L}\sum_{a\neq\pi^\star(s)}q_t(s,a)\widehat{E}_t^{\pi^\star}(s,a) \qquad \text{(ErrSub)}$$

$$+ \sum_{t=1}^{T}\sum_{s\neq s_L}\sum_{a=\pi^\star(s)}\left(q_t(s,a) - q_t^\star(s,a)\right)\widehat{E}_t^{\pi^\star}(s,a) \qquad \text{(ErrOpt)}$$

$$+ \sum_{t=1}^{T}\sum_{s\neq s_L}\sum_{a\in A}\left(q_t(s,a) - \widehat{q}_t(s,a)\right)\left(\widetilde{Q}_t^{\pi^\star}(s,a) - \widetilde{V}_t^{\pi^\star}(s)\right) \qquad \text{(OccDiff)}$$

$$+ \sum_{t=1}^{T}\sum_{s\neq s_L}\sum_{a\neq\pi^\star(s)}q_t^\star(s,a)\left(\widetilde{V}_t^{\pi^\star}(s) - V_t^{\pi^\star}(s)\right) \qquad \text{(Bias)}$$

where $\widehat{E}_t^{\pi}$ is defined as

$$\widehat{E}_t^{\pi}(s,a) = \ell_t(s,a) + \sum_{s'\in S_{k(s)+1}}P(s'|s,a)\widetilde{V}_t^{\pi}(s') - \widetilde{Q}_t^{\pi}(s,a).$$

By the exact same reasoning as in the full-information setting (Appendix B.3), we have $\mathbb{E}\left[\text{OccDiff}\right] = \mathcal{O}\left(L^4|S|^3|A|^2\ln^2\iota + \mathbb{E}\left[\mathbb{G}_3(L^4\ln\iota)\right]\right)$ and $\mathbb{E}\left[\text{Bias}\right] = \mathcal{O}(1)$, but the first two terms $\text{ErrSub}$ and $\text{ErrOpt}$ are slightly different. To see this, note that under event $\mathcal{A}$, we have

$$\widehat{E}_t^{\pi^\star}(s,a) = \ell_t(s,a) - \widetilde{\ell}_t(s,a) + \sum_{s'\in S_{k(s)+1}}\left(P(s'|s,a) - \bar{P}_{i(t)}(s'|s,a)\right)\widetilde{V}_t^{\pi^\star}(s')$$

$$= \ell_t(s,a)\left(1 - \frac{q_t(s,a)}{u_t(s,a)}\right) + L\cdot B_{i(t)}(s,a) + \sum_{s'\in S_{k(s)+1}}\left(P(s'|s,a) - \bar{P}_{i(t)}(s'|s,a)\right)\widetilde{V}_t^{\pi^\star}(s')$$

$$\leq \frac{u_t(s,a) - q_t(s,a)}{q_t(s,a)} + 2L^2\cdot B_{i(t)}(s,a)$$

where the last line applies the definition of event $\mathcal{A}$ and the fact $q_t(s,a) \leq u_t(s,a)$ given this event. Importantly, the second term has been studied and bounded in the proof of Proposition B.2 already, so we only need to focus on the first term. Before doing so, note that the range of $\widehat{E}_t^{\pi}$ is $\mathcal{O}(L|S|t)$ based on Corollary C.1.5, and thus the range of $\text{ErrSub}$ and $\text{ErrOpt}$ is $\mathcal{O}(L^2|S|T^2)$. Therefore, we only need to add a term $\mathcal{O}\left(\delta\cdot L^2|S|T^2\right)$ to address the event $\mathcal{A}^c$.

**Extra term in ERRSUB** According to previous analysis, the extra term in ERRSUB is

$$\sum_{t=1}^{T}\sum_{s\neq s_L}\sum_{a\neq\pi^\star(s)} q_t(s,a)\cdot\frac{u_t(s,a)-q_t(s,a)}{q_t(s,a)} \leq \sum_{t=1}^{T}\sum_{s\neq s_L}\sum_{a\neq\pi^\star(s)}|u_t(s,a)-q_t(s,a)|$$

$$\leq 4\sum_{t=1}^{T}\sum_{s\neq s_L}\sum_{a\neq\pi^\star(s)} r_t(s,a) + 16\sum_{t=1}^{T}\sum_{s\neq s_L}\sum_{a\neq\pi^\star(s)}\sum_{k=0}^{k(s)-1}\sum_{(u,v,w)\in T_k} q_t(u,v)\sqrt{\frac{P(w|u,v)\ln\left(\frac{T|S||A|}{\delta}\right)}{\max\left\{m_{i(t)}(u,v),1\right\}}}q_t(s,a|w)$$

(Corollary D.3.11)

$$= 4\sum_{t=1}^{T}\sum_{s\neq s_L}\sum_{a\neq\pi^\star(s)} r_t(s,a) + 16\mathbb{G}_3(\ln\iota) \qquad\qquad \text{(Definition D.2.1)}$$

$$= 16\mathbb{G}_3(\ln\iota) + \mathcal{O}\left(L^2 S^3 A^2 \ln^2\iota\right). \qquad\qquad \text{(Lemma D.3.10)}$$

Finally, using [Lemma D.3.6](#) and the bound on ERRSUB for the full-information setting, we have

$$\mathbb{E}\left[\text{ERRSUB}\right] = \mathcal{O}\left(\mathbb{G}_3(\ln\iota) + \mathbb{G}_1(L^4|S|\ln\iota) + L^2|S|^3|A|^2\ln^2\iota\right).$$

**Extra term in ERROPT** Similarly, we consider the extra term in ERROPT:

$$\sum_{t=1}^{T}\sum_{s\neq s_L}\sum_{a=\pi^\star(s)}(q_t(s,a)-q_t^\star(s,a))\cdot\frac{u_t(s,a)-q_t(s,a)}{q_t(s,a)}$$

$$\leq 4\sum_{t=1}^{T}\sum_{s\neq s_L}\sum_{a=\pi^\star(s)}\frac{q_t(s,a)-q_t^\star(s,a)}{q_t(s,a)}r_t(s,a)$$

$$+\sum_{t=1}^{T}\sum_{s\neq s_L}\sum_{a=\pi^\star(s)}\frac{q_t(s,a)-q_t^\star(s,a)}{q_t(s,a)}\cdot\left(16\sum_{u,v,w} q_t(u,v)\sqrt{\frac{P(w|u,v)\ln\left(\frac{T|S||A|}{\delta}\right)}{\max\left\{m_{i(t)}(u,v),1\right\}}}q_t(s,a|w)\right)$$

(Corollary D.3.11)

$$\leq 4\sum_{t=1}^{T}\sum_{s\neq s_L}\sum_{a=\pi^\star(s)} r_t(s,a) + 16\mathbb{G}_6(\ln\iota) \qquad\qquad \text{(Definition D.2.1)}$$

$$= 16\mathbb{G}_6(\ln\iota) + \mathcal{O}\left(L^2 S^3 A^2 \ln^2\iota\right). \qquad\qquad \text{(Lemma D.3.10)}$$

Again, considering the term that appears in the full-information setting already, we have

$$\mathbb{E}\left[\text{ERROPT}\right] = \mathcal{O}\left(\mathbb{G}_6(\ln\iota) + \mathbb{G}_2(L^4|S|\ln\iota) + L^2|S|^3|A|^2\ln^2\iota\right).$$

It remains to bound ESTREG with terms that enjoy self-bounding properties.

**Term ESTREG** According to [Eq. (51)](#) in [Lemma C.1.6](#), taking the summation of all the epochs, we have the following bound for $\mathbb{E}\left[\text{ESTREG}\right]$:

$$\mathcal{O}\left(\mathbb{E}\left[\sqrt{|S|L}\sum_{i=1}^{N}\sum_{t=t_i}^{t_{i+1}-1}\eta_t\cdot\sqrt{\sum_{s\neq s_L}\sum_{a\neq\pi^\star(s)}\widehat{q}_t(s,a)}\right] + L^2\cdot\mathbb{E}\left[\sum_{i=1}^{N}\sum_{t=t_i}^{t_{i+1}-1}\eta_t\cdot\sum_{s\neq s_L}\sum_{a\neq\pi^\star(s)}\sqrt{\widehat{q}_t(s,a)}\right]\right)$$

$$+\mathcal{O}\left(\mathbb{E}\left[L^4|A|\sum_{i=1}^{N}\sum_{t=t_i}^{t_{i+1}-1}\sum_{s\neq s_L}\sum_{a\in A}\widehat{q}_t(s,a)\cdot B_{i(t)}(s,a)^2\right]\right)$$

$$+\mathcal{O}\left(\delta\cdot\mathbb{E}\left[L|S|T\sum_{i=1}^{N}(t_{i+1}-t_i)\right] + L^4|S|^2|A|^2\ln^2\iota\right)$$

$$=\mathcal{O}\left(\mathbb{E}\left[\sqrt{|S|L}\sum_{t=1}^{T}\eta_t\cdot\sqrt{\sum_{s\neq s_L}\sum_{a\neq\pi^\star(s)}\widehat{q}_t(s,a)}\right]\right) + \mathcal{O}\left(L^2\cdot\mathbb{E}\left[\sum_{t=1}^{T}\eta_t\cdot\sum_{s\neq s_L}\sum_{a\neq\pi^\star(s)}\sqrt{\widehat{q}_t(s,a)}\right]\right)$$

$$+ \mathcal{O}\left(L^6 |S|^3 |A|^3 \ln^2 \iota\right)$$

where the lase line applies Lemma C.1.8.

Then, for the first term, we have

$$\mathbb{E}\left[\sqrt{|S|L}\sum_{t=1}^{T}\eta_t \cdot \sqrt{\sum_{s\neq s_L}\sum_{a\neq\pi^\star(s)}\widehat{q}_t(s,a)}\right]$$

$$\leq \mathbb{E}\left[\sqrt{|S|L}\cdot\sqrt{\sum_{t=1}^{T}\eta_t^2}\cdot\sqrt{\sum_{t=1}^{T}\sum_{s\neq s_L}\sum_{a\neq\pi^\star(s)}\widehat{q}_t(s,a)}\right]$$

$$\leq \mathbb{E}\left[\sqrt{4L|S|^2|A|\log^2 T}\cdot\sqrt{\sum_{t=1}^{T}\sum_{s\neq s_L}\sum_{a\neq\pi^\star(s)}\widehat{q}_t(s,a)}\right]$$

where the second line follows from the Cauchy-Schwarz inequality, and the third line applies Eq. (58).

Then, we separate the term into two parts:

$$\mathbb{E}\left[\sqrt{4L|S|^2|A|\log^2 T}\cdot\sqrt{\sum_{t=1}^{T}\sum_{s\neq s_L}\sum_{a\neq\pi^\star(s)}q_t(s,a)}\right]$$

$$+ \mathbb{E}\left[\sqrt{4L|S|^2|A|\log^2 T}\cdot\sqrt{\sum_{t=1}^{T}\sum_{s\neq s_L}\sum_{a\neq\pi^\star(s)}|\widehat{q}_t(s,a)-q_t(s,a)|}\right]$$

$$\leq \mathbb{E}\left[2\cdot\mathbb{G}_4(L|S|^2|A|\log^2 T)\right] + \mathbb{E}\left[\sum_{t=1}^{T}\sum_{s\neq s_L}\sum_{a\neq\pi^\star(s)}|\widehat{q}_t(s,a)-q_t(s,a)|\right] + 4|S|^2|A|L\log^2 T$$

where second line follows from the fact $\sqrt{xy} \leq x + y$ for $x, y \geq 0$. Note that, the second term above can be bounded by $\mathcal{O}\left(\mathbb{G}_3\left(\ln \iota\right) + L^2|S|^3|A|^2\ln^2 \iota\right)$ just as in the full-information setting (see Eq. (44)). Therefore, we have finished bounding the first term:

$$\mathbb{E}\left[\sqrt{|S|L}\sum_{t=1}^{T}\eta_t \cdot \sqrt{\sum_{s\neq s_L}\sum_{a\neq\pi^\star(s)}\widehat{q}_t(s,a)}\right]$$

$$= \mathcal{O}\left(\mathbb{E}\left[\mathbb{G}_4(L|S|^2|A|\log^2 T) + \mathbb{G}_3\left(\ln \iota\right)\right] + L^2|S|^3|A|^2\ln^2 \iota\right).$$

On the other hand, the second term can be bounded similarly:

$$L^2 \cdot \mathbb{E}\left[\sum_{t=1}^{T}\eta_t \cdot \sum_{s\neq s_L}\sum_{a\neq\pi^\star(s)}\sqrt{\widehat{q}_t(s,a)}\right]$$

$$\leq L^2 \cdot \mathbb{E}\left[\sum_{s\neq s_L}\sum_{a\neq\pi^\star(s)}\cdot\sqrt{\sum_{t=1}^{T}\eta_t^2}\cdot\sqrt{\sum_{t=1}^{T}\widehat{q}_t(s,a)}\right]$$

$$\leq L^2\sqrt{4|S||A|\log^2 T}\cdot\mathbb{E}\left[\sum_{s\neq s_L}\sum_{a\neq\pi^\star(s)}\cdot\sqrt{\sum_{t=1}^{T}\widehat{q}_t(s,a)}\right]$$

$$\leq \mathbb{E}\left[2\cdot\mathbb{G}_5(L^4|S||A|\log^2 T)\right] + \mathbb{E}\left[\sum_{t=1}^{T}\sum_{s\neq s_L}\sum_{a\neq\pi^\star(s)}|\widehat{q}_t(s,a)-q_t(s,a)|\right] + L^4|S||A|\log^2 T$$

$$= \mathcal{O}\left(\mathbb{E}\left[\mathbb{G}_5(L^4|S||A|\log^2 T) + \mathbb{G}_3\left(\ln \iota\right)\right] + L^2|S|^3|A|^2\ln^2 \iota\right).$$

So we have the final bound on $\mathbb{E}\left[\textsc{EstReg}\right]$:

$$\mathbb{E}\left[\textsc{EstReg}\right] = \mathcal{O}\Big(\mathbb{E}\left[\mathbb{G}_4\left(L|S|^2|A|\log^2 T\right) + \mathbb{G}_5\left(L^4|S||A|\log^2 T\right) + \mathbb{G}_3\left(\ln \iota\right)\right] + L^6|S|^3|A|^3\ln^2 \iota\Big)$$

Finally, by combining the bounds of each term, we finally have

$$\begin{aligned}
\text{Reg}_T(\pi^\star) \leq \mathcal{O}\Big(&\mathbb{E}\left[\mathbb{G}_1\left(L^4|S|\ln T\right) + \mathbb{G}_3\left(\ln T\right)\right] &&\text{(from \textsc{ErrSub})}\\
&+ \mathbb{E}\left[\mathbb{G}_2\left(L^4|S|\ln T\right) + \mathbb{G}_6\left(\ln T\right)\right] &&\text{(from \textsc{ErrOpt})}\\
&+ \mathbb{E}\left[\mathbb{G}_3\left(L^4\ln T\right)\right] &&\text{(from \textsc{OccDiff})}\\
&+ \mathbb{E}\left[\mathbb{G}_4\left(L|S|^2|A|\ln^2 T\right) + \mathbb{G}_5\left(L^4|S||A|\ln^2 T\right) + \mathbb{G}_3\left(\ln T\right)\right] &&\text{(from \textsc{EstReg})}\\
&+ L^6|S|^3|A|^3\ln^2 T\Big).
\end{aligned}$$

When Condition (1) holds, we apply similar self-bounding arguments to obtain a logarithmic regret bound. Specifically, for some universal constant $\kappa > 0$, we have

$$\begin{aligned}
\text{Reg}_T(\pi^\star) \leq\ &\kappa\left(\mathbb{E}\left[\mathbb{G}_1\left(L^4|S|\ln T\right) + \mathbb{G}_2\left(L^4|S|\ln T\right) + \mathbb{G}_3\left(L^4\ln T\right)\right]\right)\\
&+ \kappa\left(\mathbb{E}\left[\mathbb{G}_4\left(L|S|^2|A|\log^2 T\right) + \mathbb{G}_5\left(L^4|S||A|\log^2 T\right) + \mathbb{G}_6\left(\ln T\right)\right]\right)\\
&+ \kappa\left(L^6|S|^3|A|^3\ln^2 \iota\right).
\end{aligned}$$

Then, for any $z > 1$, by applying all the self-bounding lemmas (Lemma D.2.2-Lemma D.2.7) with $\alpha = \beta = \frac{1}{32z\kappa}$, we arrive at

$$\begin{aligned}
\text{Reg}_T(\pi^\star) \leq\ &\frac{1}{z}\cdot\left(\text{Reg}_T(\pi^\star) + C\right)\\
&+ z\cdot\mathcal{O}\left(\left(\sum_{s\neq s_L}\sum_{a\neq\pi^\star(s)}\frac{\kappa^2}{\Delta(s,a)}\right)\cdot\left(L^4|S|\ln T + L^6|S|\ln T + L^4|S||A|\log^2 T\right)\right)\\
&+ z\cdot\mathcal{O}\left(\frac{\kappa^2}{\Delta_{\text{MIN}}}\cdot\left(L^5|S|^2\ln T + L^6|S|^2\ln T + L^3|S|^2|A|\ln T + L|S|^2|A|\log^2 T\right)\right)\\
&+ \kappa\cdot\left(L^6|S|^3|A|^3\ln^2 T\right)\\
\leq\ &\frac{1}{z}\cdot\left(\text{Reg}_T(\pi^\star) + C\right) + \kappa\cdot\left(L^6|S|^3|A|^3\ln^2 T\right)\\
&+ z\cdot\mathcal{O}\left(\sum_{s\neq s_L}\sum_{a\neq\pi^\star(s)}\frac{L^6|S|\ln T + L^4|S||A|\log^2 T}{\Delta(s,a)} + \frac{L^6|S|^2\ln T + L^3|S|^2|A|\log^2 T}{\Delta_{\text{MIN}}}\right)\\
\leq\ &\frac{1}{z}\cdot\left(\text{Reg}_T(\pi^\star) + C\right) + z\cdot\kappa' U + \kappa\cdot V,
\end{aligned}$$

where $\kappa'$ is a universal constant hidden in the $\mathcal{O}(\cdot)$ notation, and $U$ and $V$ are defined in Proposition C.2). The last step is to rearrange and pick the optimal $z$, which is almost identical to that in the proof of Proposition B.2 and finally shows $\text{Reg}_T(\pi^\star) = \mathcal{O}\left(U + \sqrt{UC} + V\right)$. This completes the entire proof.

# D  General Decomposition, Self-bounding Terms, and Supplementary Lemmas

In this section, we provide details of our two key techniques: a general decomposition and self-bounding terms, as well as a set of supplementary Lemmas used throughout the analysis.

## D.1  General Decomposition Lemma

In this section, we consider measuring the performance difference between a policy $\pi$ and a mapping (deterministic policy) $\pi^\star$, that is, $V^\pi(s_0) - V^{\pi^\star}(s_0)$ where $Q$ and $V$ are the state-action and state value functions associated with some transition $P$ and some loss function $\ell$, that is,

$$Q^\pi(s,a) = \ell(s,a) + \sum_{s' \in S_{k(s)+1}} P(s'|s,a)V^\pi(s'), \quad V^\pi(s) = \sum_{a \in A} \pi(a|s)Q^\pi(s,a),$$

for all state-action pairs (with $V^\pi(s_L) = 0$). Moreover, for some estimated transition $\widehat{P}$ and estimated loss function $\widehat{\ell}$, define similarly $\widehat{Q}$ and $\widehat{V}$ as the corresponding state-action and state value functions:

$$\widehat{Q}^\pi(s,a) = \widehat{\ell}(s,a) + \sum_{s' \in S_{k(s)+1}} \widehat{P}(s'|s,a)\widehat{V}^\pi(s'), \quad \widehat{V}^\pi(s) = \sum_{a \in A} \pi(a|s)\widehat{Q}^\pi(s,a),$$

for all state-action pairs (with $\widehat{V}^\pi(s_L) = 0$).

Again, we denote by $q^\star_\pi(s,a)$ the probability of visiting a trajectory of the form $(s_0, \pi^\star(s_0)), (s_1, \pi^\star(s_1)), \ldots, (s_{k(s)-1}, \pi^\star(s_{k(s)-1})), (s,a)$ when executing policy $\pi$. In other words, $q^\star_\pi$ can be formally defined as

$$q^\star_\pi(s,a) = \begin{cases} \pi(a|s), & s = s_0, \\ \pi(a|s) \cdot \left( \sum_{s' \in S_{k(s)-1}} q^\star_\pi(s', \pi^\star(s))P(s|s', \pi^\star(s)) \right), & \text{otherwise.} \end{cases}$$

Note that our earlier notation $q^\star_t$ is thus a shorthand for $q^\star_{\pi_t}$. With slight abuse of notations, we define $q^\star_\pi(s) = \sum_{a \in A} q^\star_\pi(s,a)$.

Now, we present a general decomposition for $V^\pi(s_0) - V^{\pi^\star}(s_0)$.

**Lemma D.1.1.** *(General Performance Decomposition) For any policies $\pi$ and $u$, and a mapping (deterministic policy) $\pi^\star : S \to A$, we have*

$$\begin{aligned}
V^\pi(s_0) - V^{\pi^\star}(s_0) = &\sum_{s \neq s_L} \sum_{a \neq \pi^\star(s)} q(s,a)\widehat{E}^u(s,a) && \textit{(Error of Sub-opt actions)} \\
&+ \sum_{s \neq s_L} \sum_{a = \pi^\star(s)} \left( q(s,a) - q^\star_\pi(s,a) \right) \widehat{E}^u(s,a) && \textit{(Error of Opt actions)} \\
&+ \sum_{s \neq s_L} \sum_{a \in A} q(s,a) \left( \widehat{Q}^u(s,a) - \widehat{V}^u(s) \right) && \textit{(Policy Difference)} \\
&- \sum_{s \neq s_L} \sum_{a = \pi^\star(s)} q^\star_\pi(s,a) \left( \widehat{Q}^u(s,a) - \widehat{V}^u(s) \right) && \textit{(Estimation Bias 1)} \\
&+ \sum_{s \neq s_L} \sum_{a \neq \pi^\star(s)} q^\star_\pi(s,a) \left( \widehat{V}^u(s) - V^{\pi^\star}(s) \right), && \textit{(Estimation Bias 2)}
\end{aligned}$$

*where $q = q^{P,\pi}$ is the occupancy measure associated with transition $P$ and policy $\pi$, and $\widehat{E}^\pi$ is a surplus function with:*

$$\widehat{E}^\pi(s,a) = \ell(s,a) + \sum_{s' \in S_{k(s)+1}} P(s'|s,a)\widehat{V}^\pi(s') - \widehat{Q}^\pi(s,a).$$

Moreover, selecting the surrogate policy $u$ as the mapping $\pi^\star$ yields Corollary D.1.2, which is the key decomposition lemma used in our analysis.

**Corollary D.1.2.** *Consider an arbitrary policy sequence $\{\pi_t\}_{t=1}^T$, an arbitrary estimated transition sequence $\{\widehat{P}_t\}_{t=1}^T$, and an arbitrary estimated loss sequence $\{\widehat{\ell}_t\}_{t=1}^T$. Then, we have*

$$\underbrace{\sum_{t=1}^T \left( V^{\pi_t}(s_0) - \widehat{V}_t^{\pi_t}(s_0) \right)}_{\text{ERR}_1} + \underbrace{\left( \sum_{t=1}^T \widehat{V}_t^{\pi^\star}(s_0) - V^{\pi^\star}(s_0) \right)}_{\text{ERR}_2}$$

$$= \sum_{t=1}^T \sum_{s \neq s_L} \sum_{a \neq \pi^\star(s)} q_t(s,a) \widehat{E}_t^{\pi^\star}(s,a) \qquad \text{(Error of Sub-opt actions)}$$

$$+ \sum_{t=1}^T \sum_{s \neq s_L} \sum_{a = \pi^\star(s)} \left( q_t(s,a) - q_t^\star(s,a) \right) \widehat{E}_t^{\pi^\star}(s,a) \qquad \text{(Error of Opt actions)}$$

$$+ \sum_{t=1}^T \sum_{s \neq s_L} \sum_{a \in A} \left( q_t(s,a) - \widehat{q}_t(s,a) \right) \left( \widehat{Q}_t^{\pi^\star}(s,a) - \widehat{V}_t^{\pi^\star}(s) \right) \qquad \text{(Occupancy Difference)}$$

$$+ \sum_{t=1}^T \sum_{s \neq s_L} \sum_{a \neq \pi^\star(s)} q_t^\star(s,a) \left( \widehat{V}_t^{\pi^\star}(s) - V_t^{\pi^\star}(s) \right), \qquad \text{(Estimation Bias)}$$

*where $\widehat{q}_t = q^{\widehat{P}_t, \pi_t}$, $q_t = q^{P, \pi_t}$, $q_t^\star = q_{\pi_t}^\star$, $\widehat{Q}_t^{\pi_t}$ and $\widehat{V}_t^{\pi_t}$ are the state-action and state value functions associated with $\pi_t$, $\widehat{\ell}_t$, and $\widehat{P}_t$, and $\widehat{E}_t^\pi$ is the surplus function defined as:*

$$\widehat{E}_t^\pi(s,a) = \ell(s,a) + \sum_{s' \in S_{k(s)+1}} P(s'|s,a) \widehat{V}_t^\pi(s') - \widehat{Q}_t^\pi(s,a).$$

*Proof.* (Proof of Lemma D.1.1) By direct calculation, for all states $s$, we have

$$V^\pi(s) - \widehat{V}^u(s) = \sum_{a \in A} \pi(a|s) \left( Q^\pi(s,a) - \widehat{Q}^u(s,a) \right) + \sum_{a \in A} \pi(a|s) \left( \widehat{Q}^u(s,a) - \widehat{V}^u(s) \right)$$

$$= \sum_{a \in A} \pi(a|s) \sum_{s' \in S_{k(s)+1}} P(s'|s,a) \left( V^\pi(s') - \widehat{V}^u(s') \right)$$

$$+ \sum_{a \in A} \pi(a|s) \underbrace{\left( \ell(s,a) + \sum_{s' \in S_{k(s)+1}} P(s'|s,a) \widehat{V}^u(s') - \widehat{Q}^u(s,a) \right)}_{\widehat{E}^u(s,a)}$$

$$+ \sum_{a \in A} \pi(a|s) \left( \widehat{Q}^u(s,a) - \widehat{V}^u(s) \right).$$

By repeatedly expanding $V^\pi(s') - \widehat{V}^u(s')$ in the same way, we conclude

$$V^\pi(s_0) - \widehat{V}^u(s_0) = \sum_{s \neq s_L} \sum_{a \in A} q(s,a) \widehat{E}^u(s,a) + \sum_{s \neq s_L} \sum_{a \in A} q(s,a) \left( \widehat{Q}^u(s,a) - \widehat{V}^u(s) \right). \qquad (60)$$

On the other hand, we also have for all states $s$:

$$V^\pi(s) - \widehat{V}^u(s)$$

$$= \sum_{a = \pi^\star(s)} \pi(a|s) \left( Q^\pi(s,a) - \widehat{V}^u(s) \right) + \sum_{a \neq \pi^\star(s)} \pi(a|s) \left( Q^\pi(s,a) - \widehat{V}^u(s) \right)$$

$$= \sum_{a = \pi^\star(s)} \pi(a|s) \sum_{s' \in S_{k(s)+1}} P(s'|s,a) \left( V^\pi(s') - \widehat{V}^u(s') \right)$$

$$+ \sum_{a=\pi^\star(s)} \pi(a|s) \underbrace{\left( \ell(s,a) + \sum_{s' \in S_{k(s)+1}} P(s'|s,a)\widehat{V}^u(s') - \widehat{Q}^u(s,a) \right)}_{\widehat{E}^u(s,a)}$$

$$+ \sum_{a=\pi^\star(s)} \pi(a|s) \left( \widehat{Q}^u(s,a) - \widehat{V}^u(s) \right)$$

$$+ \sum_{a \neq \pi^\star(s)} \pi(a|s) \left( Q^\pi(s,a) - \widehat{V}^u(s) \right).$$

Using Lemma D.1.3 (which repeatedly expands $V^\pi(s') - \widehat{V}^u(s')$ in the same way) with

$$C(s) = \sum_{a=\pi^\star(s)} \pi(a|s)\widehat{E}^u(s,a) + \sum_{a=\pi^\star(s)} \pi(a|s) \left( \widehat{Q}^u(s,a) - \widehat{V}^u(s) \right)$$

$$+ \sum_{a \neq \pi^\star(s)} \pi(a|s) \left( Q^\pi(s,a) - \widehat{V}^u(s) \right)$$

we obtain

$$
\begin{aligned}
V^\pi(s_0) - \widehat{V}^u(s_0) &= \sum_{s \neq s_L} q_\pi^\star(s)C(s) \\
&= \sum_{s \neq s_L} \sum_{a=\pi^\star(s)} q_\pi^\star(s,a)\widehat{E}^u(s,a) \\
&\quad + \sum_{s \neq s_L} \sum_{a \neq \pi^\star(s)} q_\pi^\star(s,a) \left( Q^\pi(s,a) - \widehat{V}^u(s) \right) \\
&\quad + \sum_{s \neq s_L} \sum_{a=\pi^\star(s)} q_\pi^\star(s,a) \left( \widehat{Q}^u(s,a) - \widehat{V}^u(s) \right).
\end{aligned}
\tag{61}
$$

Combining Eq. (60) and Eq. (61), we have the following equality:

$$
\begin{aligned}
\sum_{s \neq s_L} &\sum_{a \neq \pi^\star(s)} q_\pi^\star(s,a) \left( Q^\pi(s,a) - \widehat{V}^u(s) \right) \\
&= \sum_{s \neq s_L} \sum_{a \in A} q(a,s)\widehat{E}^u(s,a) \\
&\quad + \sum_{s \neq s_L} \sum_{a \in A} q(s,a) \left( \widehat{Q}^u(s,a) - \widehat{V}^u(s) \right) \\
&\quad - \sum_{s \neq s_L} \sum_{a=\pi^\star(s)} q_\pi^\star(s,a)\widehat{E}^u(s,a) \\
&\quad - \sum_{s \neq s_L} \sum_{a=\pi^\star(s)} q_\pi^\star(s,a) \left( \widehat{Q}^u(s,a) - \widehat{V}^u(s) \right) \\
&= \sum_{s \neq s_L} \sum_{a \neq \pi^\star(s)} q(s,a)\widehat{E}^u(s,a) && \text{(Error of Sub-opt actions)} \tag{62} \\
&\quad + \sum_{s \neq s_L} \sum_{a=\pi^\star(s)} \left( q(s,a) - q_\pi^\star(s,a) \right) \widehat{E}^u(s,a) && \text{(Error of Opt actions)} \\
&\quad + \sum_{s \neq s_L} \sum_{a \in A} q(s,a) \left( \widehat{Q}^u(s,a) - \widehat{V}^u(s) \right) && \text{(Policy Difference)} \\
&\quad - \sum_{s \neq s_L} \sum_{a=\pi^\star(s)} q_\pi^\star(s,a) \left( \widehat{Q}^u(s,a) - \widehat{V}^u(s) \right) && \text{(Estimation Bias 1)}, \tag{63}
\end{aligned}
$$

Next, we consider the following:

$$V^\pi(s) - V^{\pi^*}(s)$$

$$= \sum_{a=\pi^\star(s)} \pi(a|s) \left( Q^\pi(s,a) - Q^\star(s,a) \right) + \sum_{a\neq\pi^\star(s)} \pi(a|s) \left( Q^\pi(s,a) - V^{\pi^\star}(s) \right)$$

$$= \sum_{a=\pi^\star(s)} \pi(a|s) \sum_{s'\in S_{k(s)+1}} P(s'|s,a) \left( V^\pi(s') - V^{\pi^\star}(s) \right) + \sum_{a\neq\pi^\star(s)} \pi(a|s) \left( Q^\pi(s,a) - V^{\pi^\star}(s) \right).$$

By Lemma D.1.3 (which again repeatedly expands $V^\pi(s') - V^{\pi^\star}(s)$ in the same way), we obtain

$$V^\pi(s_0) - V^{\pi^\star}(s_0) = \sum_{s\neq s_L} \sum_{a\neq\pi^\star(s)} q_\pi^\star(s,a) \left( Q^\pi(s,a) - V^{\pi^\star}(s) \right). \tag{64}$$

Finally, combining Eq. (62) and Eq. (64), we arrive at

$$V^\pi(s_0) - V^{\pi^\star}(s_0)$$

$$= \sum_{s\neq s_L}\sum_{a\neq\pi^\star(s)} q_\pi^\star(s,a) \left( Q^\pi(s,a) - \widehat{V}^u(s) \right) + \sum_{s\neq s_L}\sum_{a\neq\pi^\star(s)} q_\pi^\star(s,a) \left( \widehat{V}^u(s) - V^{\pi^\star}(s) \right)$$

$$= \sum_{s\neq s_L}\sum_{a\neq\pi^\star(s)} q(s,a)\widehat{E}^u(s,a) \qquad\qquad \text{(Transition Error of Sub-opt actions)}$$

$$+ \sum_{s\neq s_L}\sum_{a=\pi^\star(s)} \left( q(s,a) - q_\pi^\star(s,a) \right) \widehat{E}^u(s,a) \qquad\qquad \text{(Transition Error of Opt actions)}$$

$$+ \sum_{s\neq s_L}\sum_{a\in A} q(s,a) \left( \widehat{Q}^u(s,a) - \widehat{V}^u(s) \right) \qquad\qquad \text{(Policy Difference)}$$

$$- \sum_{s\neq s_L}\sum_{a=\pi^\star(s)} q_\pi^\star(s,a) \left( \widehat{Q}^u(s,a) - \widehat{V}^u(s) \right) \qquad\qquad \text{(Estimation Bias 1)}$$

$$+ \sum_{s\neq s_L}\sum_{a\neq\pi^\star(s)} q_\pi^\star(s,a) \left( \widehat{V}^u(s) - V^{\pi^\star}(s) \right) \qquad\qquad \text{(Estimation Bias 2)}$$

finishing the proof. $\qquad\square$

*Proof.* (Proof of Corollary D.1.2) By applying Lemma D.1.1 with $u = \pi^\star$, we know that $V_t^{\pi_t}(s_0) - V_t^{\pi^\star}(s_0)$ equals to

$$\sum_{s\neq s_L}\sum_{a\neq\pi^\star(s)} q_t(s,a)\widehat{E}_t^{\pi^\star}(s,a)$$

$$+ \sum_{s\neq s_L}\sum_{a=\pi^\star(s)} \left( q_t(s,a) - q_t^\star(s,a) \right) \widehat{E}_t^{\pi^\star}(s,a)$$

$$+ \sum_{s\neq s_L}\sum_{a\in A} \widehat{q}_t(s,a) \left( \widehat{Q}_t^{\pi^\star}(s,a) - \widehat{V}_t^{\pi^\star}(s) \right)$$

$$+ \sum_{s\neq s_L}\sum_{a\in A} \left( q_t(s,a) - \widehat{q}_t(s,a) \right) \left( \widehat{Q}_t^{\pi^\star}(s,a) - \widehat{V}_t^{\pi^\star}(s) \right)$$

$$- \sum_{s\neq s_L}\sum_{a=\pi^\star(s)} q_t^\star(s,a) \left( \widehat{Q}_t^{\pi^\star}(s,a) - \widehat{V}_t^{\pi^\star}(s) \right) \qquad\qquad \text{(Estimation Bias 1)}$$

$$+ \sum_{s\neq s_L}\sum_{a\neq\pi^\star(s)} q_t^\star(s,a) \left( \widehat{V}_t^{\pi^\star}(s) - V_t^{\pi^\star}(s) \right). \qquad\qquad \text{(Estimation Bias 2)}$$

Now observe the following two facts. First, the third term above is in fact equal to $\widehat{V}_t^{\pi_t}(s_0) - \widehat{V}_t^{\pi^\star}(s_0)$ according to the standard performance difference lemma [Kakade, 2003, Theorem 5.2.1]. Second, the first estimation bias term is simply $0$ since $\widehat{Q}_t^{\pi^\star}(s,a) = \widehat{V}_t^{\pi^\star}(s)$ when $a = \pi^\star(s)$.

Therefore, by taking the summation over $t$, we obtain

$$\text{ERR1} + \text{ERR2} = \sum_{t=1}^T \left( V_t^{\pi_t}(s_0) - V_t^{\pi^\star}(s_0) \right) - \left( \widehat{V}_t^{\pi_t}(s_0) - \widehat{V}_t^{\pi^\star}(s_0) \right)$$

$$= \sum_{s \neq s_L} \sum_{a \neq \pi^\star(s)} q_t(s,a) \widehat{E}_t^{\pi^\star}(s,a)$$

$$+ \sum_{s \neq s_L} \sum_{a = \pi^\star(s)} \left( q_t(s,a) - q_t^\star(s,a) \right) \widehat{E}_t^{\pi^\star}(s,a)$$

$$+ \sum_{s \neq s_L} \sum_{a \in A} \left( q_t(s,a) - \widehat{q}_t(s,a) \right) \left( \widehat{Q}_t^{\pi^\star}(s,a) - \widehat{V}_t^{\pi^\star}(s) \right)$$

$$+ \sum_{s \neq s_L} \sum_{a \neq \pi^\star(s)} q_t^\star(s,a) \left( \widehat{V}_t^{\pi^\star}(s) - V_t^{\pi^\star}(s) \right)$$

which finishes the proof. $\qquad\square$

**Lemma D.1.3.** *For any functions* $F : S \to \mathbb{R}$ *and* $C : S \to \mathbb{R}$ *satisfying the following condition:*

$$F(s) = \sum_{a = \pi^\star(s)} \pi(a|s) \sum_{s' \in S_{k(s)+1}} P(s'|s,a)F(s') + C(s)$$

*and* $F(s_L) = 0$, *we have*

$$F(s_0) = \sum_{s \neq s_L} q_\pi^\star(s)C(s).$$

*Proof.* By definition and direct calculation, we have $F(s_0)$ equal to

$$\sum_{a = \pi^\star(s_0)} q(s_0, a) \sum_{s' \in S_1} P(s'|s_0, a)F(s') + C(s) \qquad\qquad (q(s_0) = 1)$$

$$= \sum_{s_1 \in S_1} q_\pi^\star(s_1)F(s_1) + q_\pi^\star(s_0)C(s)$$

$$= \sum_{s_1 \in S_1} q_\pi^\star(s_1) \left( \sum_{a = \pi^\star(s)} \pi(a|s) \sum_{s' \in S_2} P(s'|s,a)F(s') \right) + \sum_{k=0}^{1} \sum_{s \in S_k} q_\pi^\star(s)C(s)$$

$$= \sum_{s_2 \in S_2} q_\pi^\star(s_2)F(s_2) + \sum_{k=0}^{1} \sum_{s \in S_k} q_\pi^\star(s)C(s) \qquad\qquad \text{(definition of } q_\pi^\star(s))$$

$$= \sum_{s_L \in S_L} q_\pi^\star(s_L)F(s_L) + \sum_{k=0}^{L-1} \sum_{s \in S_k} q_\pi^\star(s)C(s) \qquad\qquad \text{(repeatedly expanding)}$$

$$= \sum_{s \neq s_L} q_\pi^\star(s)C(s), \qquad\qquad (F(s_L) = 0)$$

which completes the proof. $\qquad\square$

## D.2   Self-bounding Terms

In this section, we summarize all the self-bounding terms we use in the proofs for the unknown transition settings.

**Definition D.2.1** (Self-bounding Terms). *For some mapping $\pi^\star : S \to A$, define the following:*

$$\mathbb{G}_1(J) = \sum_{t=1}^T \sum_{s\neq s_L} \sum_{a\neq\pi^\star(s)} q_t(s,a)\sqrt{\frac{J}{\max\left\{m_{i(t)}(s,a)\right\}}},$$

$$\mathbb{G}_2(J) = \sum_{t=1}^T \sum_{s\neq s_L} \sum_{a=\pi^\star(s)} (q_t(s,a) - q_t^\star(s,a))\sqrt{\frac{J}{\max\left\{m_{i(t)}(s,a),1\right\}}},$$

$$\mathbb{G}_3(J) = \sum_{t=1}^T \sum_{s\neq s_L} \sum_{a\neq\pi^\star(s)} \sum_{k=0}^{k(s)-1} \sum_{(u,v,w)\in T_k} q_t(u,v)\sqrt{\frac{P(w|u,v)\cdot J}{\max\left\{m_{i(t)}(u,v),1\right\}}} q_t(s,a|w),$$

$$\mathbb{G}_4(J) = \sqrt{J\cdot\sum_{t=1}^T \sum_{s\neq s_L} \sum_{a\neq\pi^\star(s)} q_t(s,a)},$$

$$\mathbb{G}_5(J) = \sum_{s\neq s_L} \sum_{a\neq\pi^\star(s)} \sqrt{J\sum_{t=1}^T q_t(s,a)},$$

$$\mathbb{G}_6(J) = \sum_{t=1}^T \sum_{s\neq s_L} \sum_{a=\pi^\star(s)} \frac{q_t(s,a) - q_t^\star(s,a)}{q_t(s,a)}\left(\sum_{k=0}^{k(s)-1} \sum_{(u,v,w)\in T_k} q_t(u,v)\sqrt{\frac{P(w|u,v)\cdot J}{\max\left\{m_{i(t)}(u,v),1\right\}}} q_t(s,a|w)\right).$$

In the next six lemmas, we show that each of these six functions enjoys a certain self-bounding property under Condition (1) so that they are small whenever the regret of the learner is small. In all these lemmas, the policy $\pi^\star$ used in $\mathbb{G}_1$-$\mathbb{G}_6$ coincides with the $\pi^\star$ in Condition (1). Also note that Lemma 5.2 is simply a collection of the first four lemmas.

**Lemma D.2.2.** *Suppose Condition (1) holds. Then we have for any $\alpha \in \mathbb{R}_+$,*

$$\mathbb{E}\left[\mathbb{G}_1(J)\right] \leq \alpha \cdot \left(\text{Reg}_T(\pi^\star) + C\right) + \frac{1}{\alpha}\sum_{s\neq s_L}\sum_{a\neq\pi^\star(s)}\frac{8J}{\Delta(s,a)}.$$

*Proof.* Under the condition, for any $\alpha \in \mathbb{R}_+$, we have

$$\mathbb{G}_1(J) = \sum_{t=1}^T\sum_{s\neq s_L}\sum_{a\neq\pi^\star(s)} q_t(s,a)\left(\sqrt{\frac{J}{\max\left\{m_{i(t)}(s,a),1\right\}}} - \alpha\Delta(s,a)\right) + \alpha\sum_{t=1}^T\sum_{s\neq s_L}\sum_{a\neq\pi^\star(s)} q_t(s,a)\Delta(s,a)$$

where the expectation of the last term is bounded by $\alpha \cdot (\text{Reg}_T(\pi^\star) + C)$. It thus remains to bound the first term. To this end, for a fixed state-action pair $(s,a)$, we define $N_{s,a}$ as the last epoch where the term in the bracket is still positive, so that:

$$m_{N_{s,a}+1}(s,a) \leq \frac{2J}{\alpha^2\Delta(s,a)^2}$$

due to the doubling epoch schedule. Then we have

$$\mathbb{E}\left[\sum_{t=1}^T q_t(s,a)\left(\sqrt{\frac{J}{\max\left\{m_{i(t)}(s,a),1\right\}}} - \alpha\Delta(s,a)\right)\right]$$

$$= \mathbb{E}\left[\sum_{i=1}^N (m_{i+1}(s,a) - m_i(s,a))\left(\sqrt{\frac{J}{\max\left\{m_i(s,a),1\right\}}} - \alpha\Delta(s,a)\right)\right]$$

$$\leq \mathbb{E}\left[\sum_{i=1}^{N_{s,a}} (m_{i+1}(s,a) - m_i(s,a))\left(\sqrt{\frac{J}{\max\left\{m_i(s,a),1\right\}}} - \alpha\Delta(s,a)\right)\right]$$

$$\leq \mathbb{E}\left[2\int_0^{m_{N_{s,a}+1}(s,a)} \sqrt{\frac{J}{x}}dx\right] \leq \mathbb{E}\left[2\int_0^{\frac{2J}{\alpha^2\Delta(s,a)^2}} \sqrt{\frac{J}{x}}dx\right]$$

$$\leq 4 \cdot \sqrt{J} \cdot \sqrt{\frac{2J}{\alpha^2 \Delta(s,a)^2}} \leq \frac{8J}{\alpha \Delta(s,a)}.$$

Taking the summation over all state-action pairs $(s,a)$ satisfying $a \neq \pi^\star(s)$, we thus have

$$\mathbb{E}\left[\mathbb{G}_2(J)\right] \leq \alpha \cdot \left(\mathrm{Reg}_T(\pi^\star) + C\right) + \sum_{s \neq s_L} \sum_{a \neq \pi^\star(s)} \frac{8J}{\alpha \Delta(s,a)}.$$

$\square$

**Lemma D.2.3.** *Suppose Condition* (1) *holds. Then we have for any* $\beta \in \mathbb{R}_+$,

$$\mathbb{E}\left[\mathbb{G}_2(J)\right] \leq \beta \cdot \left(\mathrm{Reg}_T(\pi^\star) + C\right) + \frac{1}{\beta} \cdot \frac{8|S|LJ}{\Delta_{\mathrm{MIN}}}.$$

*Proof.* Clearly, under the condition, for any $\beta \in \mathbb{R}_+$, we have

$$\mathbb{G}_2(J) = \sum_{t=1}^{T} \sum_{s \neq s_L} \sum_{a = \pi^\star(s)} (q_t(s,a) - q_t^\star(s,a)) \left( \sqrt{\frac{J}{\max\left\{m_{i(t)}(s,a), 1\right\}}} - \beta \cdot \frac{\Delta_{\mathrm{MIN}}}{L} \right)$$

$$+ \beta \sum_{t=1}^{T} \sum_{s \neq s_L} \sum_{a = \pi^\star(s)} (q_t(s,a) - q_t^\star(s,a)) \cdot \frac{\Delta_{\mathrm{MIN}}}{L}$$

where the expectation of the last term is bounded by $\beta \cdot (\mathrm{Reg}_T(\pi^\star) + C)$ according to Lemma D.2.8 (deferred to the end of this subsection). It thus remains to bound the first term. To this end, for a fixed state-action pair $(s,a)$, we similarly define $N_{s,a}$ as the last epoch where the term in the bracket is still positive, so that:

$$m_{N_{s,a}+1}(s,a) \leq \frac{2JL^2}{\beta^2 \Delta_{\mathrm{MIN}}^2}$$

due to the doubling epoch schedule. Then, we have

$$\mathbb{E}\left[ \sum_{t=1}^{T} (q_t(s,a) - q_t^\star(s,a)) \left( \sqrt{\frac{J}{\max\left\{m_{i(t)}(s,a), 1\right\}}} - \beta \cdot \frac{\Delta_{\mathrm{MIN}}}{L} \right) \right]$$

$$\leq \mathbb{E}\left[ \sum_{i=1}^{N_{s,a}} (m_{i+1}(s,a) - m_i(s,a)) \left( \sqrt{\frac{J}{\max\left\{m_i(s,a), 1\right\}}} - \beta \cdot \frac{\Delta_{\mathrm{MIN}}}{L} \right) \right]$$

$$(q_t(s,a) \geq q_t^\star(s,a) \text{ by definition})$$

$$\leq \mathbb{E}\left[ 2 \int_0^{m_{N_{s,a}+1}(s,a)} \sqrt{\frac{J}{x}} dx \right] \leq \mathbb{E}\left[ 2 \int_0^{\frac{2JL^2}{\beta^2 \Delta_{\mathrm{MIN}}^2}} \sqrt{\frac{J}{x}} dx \right]$$

$$\leq 4 \cdot \sqrt{J} \cdot \sqrt{\frac{2JL^2}{\beta^2 \Delta_{\mathrm{MIN}}^2}} \leq \frac{8LJ}{\beta \Delta_{\mathrm{MIN}}}.$$

Taking the summation over all state-action pairs satisfying $a = \pi^\star(s)$, we have

$$\mathbb{E}\left[\mathbb{G}_2(J)\right] \leq \beta \cdot \left(\mathrm{Reg}_T(\pi^\star) + C\right) + \sum_{s \neq s_L} \sum_{a = \pi^\star(s)} \frac{8LJ}{\beta \Delta_{\mathrm{MIN}}}$$

$$= \beta \cdot \left(\mathrm{Reg}_T(\pi^\star) + C\right) + \frac{8|S|LJ}{\beta \Delta_{\mathrm{MIN}}}.$$

$\square$

**Lemma D.2.4.** *Suppose Condition* (1) *holds. Then we have for any* $\alpha, \beta \in \mathbb{R}_+$,

$$\mathbb{E}\left[\mathbb{G}_3(J)\right] \leq (\alpha + \beta) \cdot \left(\mathrm{Reg}_T(\pi^\star) + C\right) + \frac{1}{\alpha} \cdot \sum_{s \neq s_L} \sum_{a \neq \pi^\star(s)} \frac{8L^2|S|J}{\Delta(s,a)} + \frac{1}{\beta} \cdot \frac{8L^2|S|^2 J}{\Delta_{\mathrm{MIN}}}.$$

*Proof.* First we have

$$\mathbb{G}_3(J) = \sum_{t=1}^{T} \sum_{k=0}^{L-1} \sum_{(u,v,w)\in T_k} q_t(u,v) \sqrt{\frac{P(w|u,v)\cdot J}{\max\{m_{i(t)}(s,a)\}}} \left( \sum_{l=k+1}^{L-1} \sum_{s\in S_l} \sum_{a\neq\pi^\star(s)} q_t(s,a|w) \right)$$

$$= \sum_{t=1}^{T} \sum_{k=0}^{L-1} \sum_{u\in S_k} \sum_{v\neq\pi^\star(s)} q_t(u,v) \left( \sum_{w\in S_{k+1}} \sqrt{\frac{P(w|u,v)\cdot J}{\max\{m_{i(t)}(s,a),1\}}} \sum_{l=k+1}^{L-1} \sum_{s\in S_l} \sum_{a\neq\pi^\star(s)} q_t(s,a|w) \right)$$

$$+ \sum_{t=1}^{T} \sum_{k=0}^{L-1} \sum_{u\in S_k} \sum_{v=\pi^\star(s)} q_t(u,v) \left( \sum_{w\in S_{k+1}} \sqrt{\frac{P(w|u,v)\cdot J}{\max\{m_{i(t)}(s,a),1\}}} \sum_{l=k+1}^{L-1} \sum_{s\in S_l} \sum_{a\neq\pi^\star(s)} q_t(s,a|w) \right)$$

$$\leq \sum_{t=1}^{T} \sum_{k=0}^{L-1} \sum_{u\in S_k} \sum_{v\neq\pi^\star(s)} q_t(u,v) \cdot \sqrt{\frac{L^2|S|\cdot J}{\max\{m_{i(t)}(s,a),1\}}}$$

$$+ \sum_{t=1}^{T} \sum_{k=0}^{L-1} \sum_{u\in S_k} \sum_{v=\pi^\star(s)} q_t(u,v) \left( \sum_{w\in S_{k+1}} \sqrt{\frac{P(w|u,v)\cdot J}{\max\{m_{i(t)}(s,a),1\}}} \sum_{l=k+1}^{L-1} \sum_{s\in S_l} \sum_{a\neq\pi^\star(s)} q_t(s,a|w) \right)$$

where the second step separates the optimal and sub-optimal state-action pairs, and the inequality follows from the fact $\sum_{s\neq s_L} \sum_{a\in A} q_t(s,a|w) \leq L$ and the Cauchy-Schwarz inequality. Note that, the first term is simply $\mathbb{G}_1(L^2|S|)$ and can be applied using Lemma D.2.2.

To bound the last term, we first observe the following

$$\sum_{t=1}^{T} \sum_{k=0}^{L-1} \sum_{u\in S_k} \sum_{v=\pi^\star(s)} q_t(u,v) \left( \sum_{w\in S_{k+1}} \left( P(w|u,v)\cdot \frac{\Delta_{\text{MIN}}}{L} \right) \sum_{l=k+1}^{L-1} \sum_{s\in S_l} \sum_{a\neq\pi^\star(s)} q_t(s,a|w) \right)$$

$$= \sum_{t=1}^{T} \sum_{l=0}^{L-1} \sum_{s\in S_l} \sum_{a\neq\pi^\star(s)} \frac{\Delta_{\text{MIN}}}{L} \cdot \left( \sum_{k=0}^{l-1} \sum_{u\in S_k} \sum_{v=\pi^\star(s)} \sum_{w\in S_{k+1}} q_t(u,v) P(w|u,v) q_t(s,a|w) \right)$$

$$\leq \sum_{t=1}^{T} \sum_{l=0}^{L-1} \sum_{s\in S_l} \sum_{a\neq\pi^\star(s)} \frac{\Delta_{\text{MIN}}}{L} \cdot \left( \sum_{k=0}^{l-1} q_t(s,a) \right)$$

$$\leq \sum_{t=1}^{T} \sum_{l=0}^{L-1} \sum_{s\in S_l} \sum_{a\neq\pi^\star(s)} q_t(s,a) \Delta_{\text{MIN}}$$

where the expectation of the last term is bounded by $\text{Reg}_T(\pi^\star) + C$ under Condition (1).

Let $\text{cilp}[x] = \max\{x,0\}$ be the clipping function that removes the negative value. By adding and subtracting $\beta$ times the term above, we have

$$\sum_{t=1}^{T} \sum_{k=0}^{L-1} \sum_{u\in S_k} \sum_{v=\pi^\star(s)} q_t(u,v) \left( \sum_{w\in S_{k+1}} \sqrt{\frac{P(w|u,v)\cdot J}{\max\{m_{i(t)}(s,a),1\}}} \sum_{l=k+1}^{L-1} \sum_{s\in S_l} \sum_{a\neq\pi^\star(s)} q_t(s,a|w) \right)$$

$$= \beta \sum_{t=1}^{T} \sum_{k=0}^{L-1} \sum_{u\in S_k} \sum_{v=\pi^\star(s)} q_t(u,v) \left( \sum_{w\in S_{k+1}} \left( P(w|u,v)\cdot \frac{\Delta_{\text{MIN}}}{L} \right) \sum_{l=k+1}^{L-1} \sum_{s\in S_l} \sum_{a\neq\pi^\star(s)} q_t(s,a|w) \right)$$

$$+ \sum_{t=1}^{T} \sum_{k=0}^{L-1} \sum_{u\in S_k} \sum_{v=\pi^\star(s)} q_t(u,v) \left( \sum_{w\in S_{k+1}} \left( \sqrt{\frac{P(w|u,v)\cdot J}{\max\{m_{i(t)}(s,a),1\}}} - \beta\cdot\frac{\Delta_{\text{MIN}}P(w|u,v)}{L} \right) \sum_{l=k+1}^{L-1} \sum_{s\in S_l} \sum_{a\neq\pi^\star(s)} q_t(s,a|w) \right)$$

$$\leq \beta \sum_{t=1}^{T} \sum_{l=0}^{L-1} \sum_{s\in S_l} \sum_{a\neq\pi^\star(s)} q_t(s,a) \Delta_{\text{MIN}}$$

$$+ L \sum_{t=1}^{T} \sum_{k=0}^{L-1} \sum_{u\in S_k} \sum_{v=\pi^\star(s)} \sum_{w\in S_{k+1}} q_t(u,v) \text{clip}\left[ \sqrt{\frac{P(w|u,v)\cdot J}{\max\{m_{i(t)}(s,a),1\}}} - \beta\cdot\frac{\Delta_{\text{MIN}}P(w|u,v)}{L} \right]$$

where the last line follows from the facts $x \leq \text{clip}[x]$ and $\sum_{s \neq s_L} \sum_{a \in A} q_t(s, a|w) \leq L$.

Fix a tuple $N_{u,v,w}$ where $v = \pi^\star(u)$, we similarly define $N_{u,v,w}$ as the last epoch where the argument of $\text{clip}(\cdot)$ is still positive, so that:

$$m_{N_{u,v,w}+1}(s, a) \leq \frac{2JL^2}{P(w|u,v)\beta^2\Delta_{\text{MIN}}^2}$$

due to the doubling epoch schedule. Then, we have

$$\mathbb{E}\left[\sum_{t=1}^{T} q_t(u, v)\text{clip}\left[\sqrt{\frac{P(w|u,v) \cdot J}{\max\{m_{i(t)}(s, a), 1\}}} - \beta \cdot \frac{\Delta_{\text{MIN}}P(w|u,v)}{L}\right]\right]$$

$$\leq \mathbb{E}\left[\sum_{i=1}^{N_{u,v,w}} (m_{i+1}(u, v) - m_i(u, v))\,\text{clip}\left[\sqrt{\frac{P(w|u,v) \cdot J}{\max\{m_{i(t)}(s, a), 1\}}} - \beta \cdot \frac{\Delta_{\text{MIN}}P(w|u,v)}{L}\right]\right]$$

$$\leq \mathbb{E}\left[2\int_0^{m_{N_{u,v,w}+1}(s,a)} \sqrt{\frac{P(w|u,v) \cdot J}{x}}dx\right] \leq \mathbb{E}\left[2\int_0^{\frac{2JL^2}{P(w|u,v)\beta^2\Delta_{\text{MIN}}^2}} \sqrt{\frac{P(w|u,v)J}{x}}dx\right]$$

$$\leq 4 \cdot \sqrt{P(w|u,v) \cdot J} \cdot \sqrt{\frac{2JL^2}{P(w|u,v)\beta^2\Delta_{\text{MIN}}^2}} \leq \frac{8LJ}{\beta\Delta_{\text{MIN}}}.$$

Taking the summation over all transition tuple $(u, v, w)$ satisfying $v = \pi^\star(s)$ and adding $\mathbb{E}\left[\mathbb{G}_1(L^2|S|J)\right]$, we have

$$\mathbb{E}\left[\mathbb{G}_3(J)\right] \leq \beta \cdot (\text{Reg}_T(\pi^\star) + C) + \mathbb{E}\left[\mathbb{G}_1(L^2|S|J)\right] + L\sum_{k=0}^{L-1}\sum_{u \in S_k}\sum_{v=\pi^\star(u)}\sum_{w \in S_{k+1}}\frac{8LJ}{\beta\Delta_{\text{MIN}}}$$

$$\leq (\alpha + \beta) \cdot (\text{Reg}_T(\pi^\star) + C) + \frac{1}{\alpha} \cdot \sum_{s \neq s_L}\sum_{a \neq \pi^\star(s)}\frac{8L^2|S|J}{\Delta(s,a)} + \frac{1}{\beta} \cdot \frac{8L^2|S|^2J}{\Delta_{\text{MIN}}},$$

where the last line follows from the fact $\sum_{k=0}^{L-1}|S_k||S_k + 1| \leq |S|^2$. $\qquad\square$

**Lemma D.2.5.** *Suppose Condition* (1) *holds. Then we have for any $\beta \in \mathbb{R}_+$,*

$$\mathbb{E}\left[\mathbb{G}_4(J)\right] \leq \beta \cdot (\text{Reg}_T(\pi^\star) + C) + \frac{1}{\beta} \cdot \frac{J}{4\Delta_{\text{MIN}}}.$$

*Proof.* By the fact that $2\sqrt{xy} \leq x + y$ for all $x, y \geq 0$, with Condition (1), we have

$$\mathbb{E}\left[\mathbb{G}_4(J)\right] = \mathbb{E}\left[\sqrt{2\beta\sum_{t=1}^{T}\sum_{s \neq s_L}\sum_{a \neq \pi^\star(s)}q_t(s,a)\Delta_{\text{MIN}} \cdot \frac{J}{2\beta\Delta_{\text{MIN}}}}\right]$$

$$\leq \beta \cdot \mathbb{E}\left[\sum_{t=1}^{T}\sum_{s \neq s_L}\sum_{a \neq \pi^\star(s)}q_t(s,a)\Delta_{\text{MIN}}\right] + \frac{J}{4\beta\Delta_{\text{MIN}}}$$

$$\leq \beta \cdot (\text{Reg}_T(\pi^\star) + C) + \frac{J}{4\beta\Delta_{\text{MIN}}}.$$

$\qquad\square$

**Lemma D.2.6.** *Suppose Condition* (1) *holds. Then we have for any $\alpha \in \mathbb{R}_+$,*

$$\mathbb{E}\left[\mathbb{G}_5(J)\right] \leq \alpha \cdot (\text{Reg}_T(\pi^\star) + C) + \sum_{s \neq s_L}\sum_{a \neq \pi^\star(s)}\frac{J}{4\alpha\Delta(s,a)}.$$

*Proof.* By the fact that $2\sqrt{xy} \leq x + y$ for all $x, y \geq 0$, with Condition (1), we have

$$\mathbb{E}\left[\mathbb{G}_4(J)\right] = \mathbb{E}\left[\sum_{s \neq s_L} \sum_{a \neq \pi^\star(s)} \sqrt{2\alpha \sum_{t=1}^T q_t(s,a)\Delta(s,a) \cdot \frac{J}{2\alpha\Delta(s,a)}}\right]$$

$$\leq \alpha \cdot \mathbb{E}\left[\sum_{t=1}^T \sum_{s \neq s_L} \sum_{a \neq \pi^\star(s)} q_t(s,a)\Delta(s,a)\right] + \sum_{s \neq s_L} \sum_{a \neq \pi^\star(s)} \frac{J}{4\alpha\Delta(s,a)}$$

$$\leq \alpha \cdot \left(\mathrm{Reg}_T(\pi^\star) + C\right) + \sum_{s \neq s_L} \sum_{a \neq \pi^\star(s)} \frac{J}{4\alpha\Delta(s,a)}.$$

$\square$

**Lemma D.2.7.** *Suppose Condition (1) holds. Then we have for any $\beta \in \mathbb{R}_+$,*

$$\mathbb{E}\left[\mathbb{G}_6(J)\right] \leq \beta \cdot \left(\mathrm{Reg}_T(\pi^\star) + C\right) + \frac{1}{\beta} \cdot \frac{8L^3 |S|^2 |A| \cdot J}{\Delta_{\mathrm{MIN}}}.$$

*Proof.* By adding and subtracting terms, we have $\mathbb{G}_6(J)$ equals to

$$\sum_{t=1}^T \sum_{s \neq s_L} \sum_{a = \pi^\star(s)} \frac{q_t(s,a) - q_t^\star(s,a)}{q_t(s,a)} \cdot$$

$$\left(\sum_{k=0}^{k(s)-1} \sum_{(u,v,w) \in T_k} q_t(u,v)\sqrt{\frac{P(w|u,v)\ln\left(\frac{T|S||A|}{\delta}\right)}{\max\left\{m_{i(t)}(u,v), 1\right\}}} q_t(s,a|w) - \beta q_t(s,a) \cdot \frac{\Delta_{\mathrm{MIN}}}{L}\right)$$

$$+ \frac{\beta}{L}\sum_{t=1}^T \sum_{s \neq s_L} \sum_{a = \pi^\star(s)} \left(q_t(s,a) - q_t^\star(s,a)\right)\Delta_{\mathrm{MIN}}$$

where the expectation of the last term is bounded by $\beta \cdot \left(\mathrm{Reg}_T(\pi^\star) + C\right)$ according to Lemma D.2.8.
To bound the first term, we observe that

$$\sum_{k=0}^{k(s)-1} \sum_{(u,v,w) \in T_k} q_t(u,v)\sqrt{\frac{P(w|u,v)\ln\left(\frac{T|S||A|}{\delta}\right)}{\max\left\{m_{i(t)}(u,v), 1\right\}}} q_t(s,a|w) - \beta q_t(s,a) \cdot \frac{\Delta_{\mathrm{MIN}}}{L}$$

$$= \sum_{k=0}^{k(s)-1} \sum_{(u,v,w) \in T_k} q_t(u,v)\sqrt{\frac{P(w|u,v)\ln\left(\frac{T|S||A|}{\delta}\right)}{\max\left\{m_{i(t)}(u,v), 1\right\}}} q_t(s,a|w)$$

$$- \beta \cdot \frac{\Delta_{\mathrm{MIN}}}{L^2} \cdot \left(\sum_{k=0}^{k(s)-1} \sum_{(u,v,w) \in T_k} q_t(u,v)P(w|u,v)q_t(s,a|w)\right)$$

$$= \sum_{k=0}^{k(s)-1} \sum_{(u,v,w) \in T_k} q_t(u,v)\left(\sqrt{\frac{P(w|u,v)\ln\left(\frac{T|S||A|}{\delta}\right)}{\max\left\{m_{i(t)}(u,v), 1\right\}}} - P(w|u,v) \cdot \beta \cdot \frac{\Delta_{\mathrm{MIN}}}{L^2}\right) \cdot q_t(s,a|w)$$

$$\leq \sum_{k=0}^{k(s)-1} \sum_{(u,v,w) \in T_k} q_t(u,v)\underbrace{\mathrm{clip}\left[\sqrt{\frac{P(w|u,v)\ln\left(\frac{T|S||A|}{\delta}\right)}{\max\left\{m_{i(t)}(u,v), 1\right\}}} - P(w|u,v) \cdot \beta \cdot \frac{\Delta_{\mathrm{MIN}}}{L^2}\right]}_{=h_t(u,v,w)} q_t(s,a|w)$$

where the first equality uses $\sum_{(u,v,w) \in T_k} q_t(u,v)P(w|u,v)q_t(s,a|w) = q_t(s,a)$ for all layer $k = 0, \ldots k(s) - 1$. (Recall $\mathrm{clip}[x] = \max\{x, 0\}$.)

Therefore, with Condition (1), we bound the $\mathbb{E}\left[\mathbb{G}_6(J)\right]$ by

$$\mathbb{E}\left[\sum_{t=1}^{T}\sum_{s\neq s_L}\sum_{a=\pi^\star(s)}\frac{q_t(s,a)-q_t^\star(s,a)}{q_t(s,a)}\left(\sum_{u,v,w}q_t(u,v)h_t(u,v,w)q_t(s,a|w)\right)+\beta\cdot\left(\mathrm{Reg}_T(\pi^\star)+C\right)\right]$$

$$\leq\mathbb{E}\left[\sum_{t=1}^{T}\sum_{s\neq s_L}\sum_{a=\pi^\star(s)}\left(\sum_{u,v,w}q_t(u,v)h_t(u,v,w)q_t(s,a|w)\right)\right]+\beta\cdot\left(\mathrm{Reg}_T(\pi^\star)+C\right)$$

$$\leq L\mathbb{E}\left[\cdot\sum_{t=1}^{T}\sum_{u,v,w}q_t(u,v)h_t(u,v,w)\right]+\beta\cdot\left(\mathrm{Reg}_T(\pi^\star)+C\right)$$

where the second line applies the fact $\frac{q_t(s,a)-q_t^\star(s,a)}{q_t(s,a)}\leq 1$, and the third line changes summation order and uses the fact that $\sum_{s\neq s_L}\sum_{a\in A}q_t(s,a|w)\leq L$.

Finally, following the similar idea of handing $\sum_{t=1}q_t(u,v)h_t(u,v,w)$ as in Lemma D.2.4, we have

$$\mathbb{E}\left[\sum_{t=1}q_t(u,v)h_t(u,v,w)\right]\leq\frac{8L^2J}{\beta\Delta_{\mathrm{MIN}}}.$$

By taking the summation over all transition triples, we have

$$\mathbb{E}\left[\mathbb{G}_6(J)\right]\leq\beta\cdot\left(\mathrm{Reg}_T(\pi^\star)+C\right)+L\cdot\sum_{k=0}^{L-1}\sum_{(u,v,w)\in T_k}\frac{1}{\beta}\cdot\frac{8L^2\cdot J}{\Delta_{\mathrm{MIN}}}$$

$$\leq\beta\cdot\left(\mathrm{Reg}_T(\pi^\star)+C\right)+\frac{1}{\beta}\cdot\frac{8L^3|S|^2|A|\cdot J}{\Delta_{\mathrm{MIN}}},$$

where the last line follows from the fact that $\sum_{k=0}^{L}|S_k||S_{k+1}|\leq|S|^2$. $\qquad\square$

**Lemma D.2.8.** *Under Condition* (1)*, we have*

$$\mathbb{E}\left[\sum_{t=1}^{T}\sum_{s\neq s_L}\sum_{a=\pi^\star(s)}\left(q_t(s,a)-q_t^\star(s,a)\right)\Delta_{\mathrm{MIN}}\right]\leq L\cdot\mathbb{E}\left[\mathrm{Reg}_T(\pi^\star)+C\right].$$

*Proof.* For each $k$, we proceed as

$$\sum_{s\in S_k}\sum_{a=\pi^\star(s)}\left(q_t(s,a)-q_t^\star(s,a)\right)$$

$$\leq 1-\sum_{s\in S_k}\sum_{a=\pi^\star(s)}q_t^\star(s,a)\qquad\qquad(\textstyle\sum_{s\in S_k}\sum_{a\in A}q_t(s,a)=1)$$

$$=1-\sum_{s\in S_k}\sum_{a=\pi^\star(s)}\pi_t(a|s)\Pr\left[\{s_k=s\}\bigcap\left(\bigcap_{\tau=0}^{k-1}\{a_\tau=\pi^\star(s_\tau)\}\right)\middle|P,\pi_t\right]\quad(\text{definition of }q_t^\star)$$

$$=1-\Pr\left[\left(\bigcap_{\tau=0}^{k}\{a_\tau=\pi^\star(s_\tau)\}\right)\middle|P,\pi_t\right]$$

$$=\Pr\left[\left(\bigcap_{\tau=0}^{k}\{a_\tau=\pi^\star(s_\tau)\}\right)^c\middle|P,\pi_t\right]$$

$$=\Pr\left[\left(\bigcup_{\tau=0}^{k}\{a_\tau\neq\pi^\star(s_\tau)\}\right)\middle|P,\pi_t\right]\qquad\qquad(\text{De Morgan's laws})$$

$$\leq\sum_{\tau=0}^{k}\Pr\left[a_\tau\neq\pi^\star(s_\tau)|P,\pi_t\right]\qquad\qquad(\text{union bound})$$

$$= \sum_{\tau=0}^{k} \sum_{s \in S_\tau} \sum_{a \neq \pi^\star(s)} q_t(s,a) = \sum_{s \neq s_L} \sum_{a \neq \pi^\star(s)} q_t(s,a).$$

Therefore, we have

$$\sum_{t=1}^{T} \sum_{s \neq s_L} \sum_{a = \pi^\star(s)} (q_t(s,a) - q_\pi^\star(s,a)) \, \Delta_{\text{MIN}}$$

$$\leq L \cdot \sum_{t=1}^{T} \sum_{s \neq s_L} \sum_{a \neq \pi^\star(s)} q_t(s,a) \cdot \Delta(s,a)$$

$$\leq L \cdot \mathbb{E} \left[ \text{Reg}_T(\pi^\star) + C \right]$$

where the last line follows from Condition (1). $\qquad\square$

### D.3 Supplementary Lemmas

**Lemma D.3.1.** *(Occupancy Measure Difference) For any policy $\pi$ and transition functions $P_1$ and $P_2$, with $q_1 = q^{P_1,\pi}$ and $q_2 = q^{P_2,\pi}$ we have for all $s$,*

$$q_1(s) - q_2(s) = \sum_{k=0}^{k(s)-1} \sum_{u \in S_k} \sum_{v \in A} \sum_{w \in S_{k+1}} q_1(u,v) \left[ P_1(w|u,v) - P_2(w|u,v) \right] q_2(s|w) \tag{65}$$

$$= \sum_{k=0}^{k(s)-1} \sum_{u \in S_k} \sum_{v \in A} \sum_{w \in S_{k+1}} q_2(u,v) \left[ P_1(w|u,v) - P_2(w|u,v) \right] q_1(s|w)$$

*where the conditional occupancy measure $q_1(s'|s)$ (similarly for $q_2(s'|s)$) is defined recursively as*

$$q_1(s'|s) = \begin{cases} 0, & k(s') < k(s) \text{ or } (k(s') = k(s) \text{ and } s' \neq s) \\ 1, & k(s') = k(s) \text{ and } s' = s \\ \sum_{u \in S_{k(s')-1}} q_1(u|s) \left( \sum_{v \in A} \pi(v|u) P(s'|u,v) \right), & k(s') > k(s) \end{cases} \tag{66}$$

*which is the conditional probability of visiting state $s'$ from $s$ under $\pi$ and transition $P_1$.*

*Proof.* Fix a state $s$. We proceed as:

$$q_1(s) - q_2(s)$$

$$= \sum_{s' \in S_{k(s)-1}} \sum_{a' \in A} (q_1(s',a') P_1(s|s',a') - q_2(s',a') P_2(s|s',a'))$$

$$= \sum_{s' \in S_{k(s)-1}} \sum_{a' \in A} (q_1(s') - q_2(s')) \, P_1(s|s',a') \pi(a'|s')$$

$$+ \sum_{s' \in S_{k(s)-1}} \sum_{a' \in A} q_2(s',a') \, (P_1(s|s',a') - P_2(s|s',a'))$$

where the second step follows by subtracting and adding $q_2(s',a') P_1(s|s',a')$. Note that, $\sum_{a' \in A} \pi(a'|s') P_1(s|s',a')$ is exactly the conditional probability of transiting to state $s$ from state $s'$ with transition $P_1$. Therefore, we have $\sum_{a' \in A} \pi(a'|s') P_1(s|s',a') = q_1(s|s')$ according to Eq. (66), and further expand $q_1(s) - q_2(s)$ as:

$$\sum_{s' \in S_{k(s)-1}} \sum_{a' \in A} (q_1(s') - q_2(s')) \, P_1(s|s',a') \pi(a'|s')$$

$$+ \sum_{s' \in S_{k(s)-1}} \sum_{a' \in A} q_2(s',a') \, (P_1(s|s',a') - P_2(s|s',a'))$$

$$= \sum_{s' \in S_{k(s)-1}} q_1(s|s') \, (q_1(s') - q_2(s'))$$

$$+ \sum_{s' \in S_{k(s)-1}} \sum_{a' \in A} q_2(s', a') \left[ P_1(s|s', a') - P_2(s|s', a') \right] q_1(s|s)$$

where the second line follows from the fact that $q_1(s|s) = 1$.

Therefore, we can recursively expand $q_1(s) - q_2(s)$ as:

$$q_1(s) - q_2(s)$$

$$= \sum_{s' \in S_{k(s)-1}} (q_1(s') - q_2(s')) q_1(s|s')$$

$$+ \sum_{s' \in S_{k(s)-1}} \sum_{a' \in A} q_2(s', a') \left[ P_1(s|s', a') - P_2(s|s', a') \right] q_1(s|s)$$

$$= \sum_{s' \in S_{k(s)-1}} (q_1(s') - q_2(s')) q_1(s|s')$$

$$+ \sum_{k=k(s)}^{k(s)} \sum_{(u,v,w) \in T_k} q_2(u, v) \left[ P_1(w|u, v) - P_2(w|u, v) \right] q_1(s|w)$$

$$= \sum_{s' \in S_{k(s)-1}} \left( \sum_{s'' \in S_{k(s)-2}} (q_1(s'') - q_2(s'')) q_1(s'|s'') \right) q_1(s|s')$$

$$+ \sum_{k=k(s)-1}^{k(s)} \sum_{(u,v,w) \in T_k} q_2(u, v) \left[ P_1(s|s', a') - P_2(s|s', a') \right] q_1(s|w)$$

$$= \sum_{s'' \in S_{k(s)-2}} (q_1(s'') - q_2(s'')) q_1(s|s'') + \sum_{k=k(s)-1}^{k(s)} \sum_{(u,v,w) \in T_k} q_2(u, v) \left[ P_1(s|s', a') - P_2(s|s', a') \right] q_1(s|w)$$

$$= \sum_{k=0}^{k(s)-1} \sum_{u \in S_k} \sum_{v \in A} \sum_{w \in S_{k+1}} q_2(u, v) \left[ P_1(w|u, v) - P_2(w|u, v) \right] q_1(s|w). \qquad \text{(expand recursively)}$$

where the second step follows from the fact that $q(s'|s) = 0$ for all states $s \neq s'$ with $k(s) = k(s')$, and the third step follows from the fact $\sum_{s' \in S_k} q(s'|s'')q(s|s') = q(s|s'')$ for all state pairs that $k(s) > k > k(s'')$.

By applying the same technique, we also have

$$q_2(s) - q_1(s) = \sum_{k=0}^{k(s)-1} \sum_{u \in S_k} \sum_{v \in A} \sum_{w \in S_{k+1}} q_1(u, v) \left[ P_2(w|u, v) - P_1(w|u, v) \right] q_2(s|w).$$

Flipping this equality finishes the proof for the second statement of the lemma:

$$q_1(s) - q_2(s) = \sum_{k=0}^{k(s)-1} \sum_{u \in S_k} \sum_{v \in A} \sum_{w \in S_{k+1}} q_1(u, v) \left[ P_1(w|u, v) - P_2(w|u, v) \right] q_2(s|w).$$

$\square$

**Lemma D.3.2.** *The following holds:*

$$B_i(s, a) \leq 2\sqrt{\frac{|S_{k(s)+1}| \ln \left( \frac{T|S||A|}{\delta} \right)}{\max \{m_i(s, a), 1\}}} + \frac{14|S_{k(s)+1}| \ln \left( \frac{T|S||A|}{\delta} \right)}{3 \max \{m_i(s, a), 1\}}.$$

*Proof.* By the definition of $B_i(s, a)$, we have

$$B_i(s, a) = \sum_{s' \in S_{k(s)+1}} B_i(s, a, s')$$

$$= \sum_{s' \in S_{k(s)+1}} \left( 2\sqrt{\frac{\bar{P}_i(s'|s,a) \ln\left(\frac{T|S||A|}{\delta}\right)}{\max\{m_i(s,a),1\}}} + \frac{14 \ln\left(\frac{T|S||A|}{\delta}\right)}{3 \max\{m_i(s,a),1\}} \right)$$

$$\leq 2\sqrt{\frac{|S_{k(s)+1}| \ln\left(\frac{T|S||A|}{\delta}\right)}{\max\{m_i(s,a),1\}}} + \frac{14|S_{k(s)+1}| \ln\left(\frac{T|S||A|}{\delta}\right)}{3 \max\{m_i(s,a),1\}}$$

where the last line follows from the Cauchy-Schwarz inequality. $\square$

**Lemma D.3.3.** *Conditioning on event $\mathcal{A}$, we have*

$$B_i(s,a,s') \leq 4\sqrt{\frac{P(s'|s,a) \ln\left(\frac{T|S||A|}{\delta}\right)}{\max\{m_i(s,a),1\}}} + \frac{40 \ln\left(\frac{T|S||A|}{\delta}\right)}{3 \max\{m_i(s,a),1\}}. \tag{67}$$

*Proof.* By direct calculation based on Eq. (8) and the condition of event $\mathcal{A}$, we have

$$B_i(s,a,s') \leq 2\sqrt{\frac{\bar{P}_i(s'|s,a) \ln\left(\frac{T|S||A|}{\delta}\right)}{\max\{m_i(s,a),1\}}} + \frac{14 \ln\left(\frac{T|S||A|}{\delta}\right)}{3 \max\{m_i(s,a),1\}}$$

$$\leq 2\sqrt{\frac{(P(s'|s,a) + B_i(s,a,s')) \ln\left(\frac{T|S||A|}{\delta}\right)}{\max\{m_i(s,a),1\}}} + \frac{14 \ln\left(\frac{T|S||A|}{\delta}\right)}{3 \max\{m_i(s,a),1\}}$$

$$\leq 2\sqrt{\frac{P(s'|s,a) \ln\left(\frac{T|S||A|}{\delta}\right)}{\max\{m_i(s,a),1\}}} + \sqrt{\frac{4B_i(s,a,s') \ln\left(\frac{T|S||A|}{\delta}\right)}{\max\{m_i(s,a),1\}}} + \frac{14 \ln\left(\frac{T|S||A|}{\delta}\right)}{3 \max\{m_i(s,a),1\}}$$

$$\leq 2\sqrt{\frac{P(s'|s,a) \ln\left(\frac{T|S||A|}{\delta}\right)}{\max\{m_i(s,a),1\}}} + \frac{B_i(s,a,s')}{2} + \frac{20 \ln\left(\frac{T|S||A|}{\delta}\right)}{3 \max\{m_i(s,a),1\}},$$

where the third line applies the fact that $\sqrt{x+y} \leq \sqrt{x} + \sqrt{y}$, and the last line follows from the fact $2\sqrt{xy} \leq x + y$ for $x, y > 0$.

Rearranging the terms yields that

$$B_i(s,a,s') \leq 4\sqrt{\frac{P(s'|s,a) \ln\left(\frac{T|S||A|}{\delta}\right)}{\max\{m_i(s,a),1\}}} + \frac{40 \ln\left(\frac{T|S||A|}{\delta}\right)}{3 \max\{m_i(s,a),1\}}.$$

$\square$

Combining with the fact $B_i(s,a,s') \leq 1$, we have the following tighter bound of confidence width.

**Corollary D.3.4.** *Conditioning on event $\mathcal{A}$, we have*

$$B_i(s,a,s') \leq \min\left\{ 4\sqrt{\frac{P(s'|s,a) \ln\left(\frac{T|S||A|}{\delta}\right)}{\max\{m_i(s,a),1\}}} + \frac{40 \ln\left(\frac{T|S||A|}{\delta}\right)}{3 \max\{m_i(s,a),1\}}, 1 \right\}$$

$$\leq \min\left\{ 4\sqrt{\frac{P(s'|s,a) \ln\left(\frac{T|S||A|}{\delta}\right)}{\max\{m_i(s,a),1\}}}, 1 \right\} + \min\left\{ \frac{40 \ln\left(\frac{T|S||A|}{\delta}\right)}{3 \max\{m_i(s,a),1\}}, 1 \right\}.$$

We often use the following two lemmas to deal with the small-probability event $\mathcal{A}^c$ when taking expectation.

**Lemma D.3.5.** *Suppose that a random variable $X$ satisfies the following conditions:*

- *Conditioning on event $\mathcal{E}$, $X < Y$ where $Y > 0$ is another random variable;*
- *$X < C$ holds always for some fixed $C \in \mathbb{R}+$.*

*Then, we have*

$$\mathbb{E}[X] \leq C \cdot \Pr[\mathcal{E}^c] + \mathbb{E}[Y].$$

*Proof.* By writing the random variable $X$ as $X \cdot \mathbb{I}\{\mathcal{E}\} + X \cdot \mathbb{I}\{\mathcal{E}^c\}$, and noting

$$X \cdot \mathbb{I}\{\mathcal{E}\} \leq Y \cdot \mathbb{I}\{\mathcal{E}\} \leq Y, \text{ and } X \cdot \mathbb{I}\{\mathcal{E}^c\} \leq C \cdot \mathbb{I}\{\mathcal{E}^c\},$$

we prove the statement after taking the expectations. $\qquad\square$

**Lemma D.3.6.** *Suppose that a random variable $X$ satisfies the following conditions:*

- *Conditioning on event $\mathcal{E}$, $X < Y$ where $Y > 0$ is another random variable;*
- *$X < C$ holds where $C$ is another random variable which ensures $\mathbb{E}[C|\mathcal{E}^c] \leq D$ for some fixed $D \in \mathbb{R}_+$.*

*Then, we have*

$$\mathbb{E}[X] \leq D \cdot \Pr[\mathcal{E}^c] + \mathbb{E}[Y].$$

*Proof.* By writing the random variable $X$ as $X \cdot \mathbb{I}\{\mathcal{E}\} + X \cdot \mathbb{I}\{\mathcal{E}^c\}$, and noting

$$X \cdot \mathbb{I}\{\mathcal{E}\} \leq Y \cdot \mathbb{I}\{\mathcal{E}\} \leq Y, \quad X \cdot \mathbb{I}\{\mathcal{E}^c\} \leq C \cdot \mathbb{I}\{\mathcal{E}^c\}, \quad \mathbb{E}[C \cdot \mathbb{I}\{\mathcal{E}^c\}] \leq \mathbb{E}[C|\mathcal{E}^c],$$

we prove the statement after taking the expectations. $\qquad\square$

**Lemma D.3.7.** *([Jin et al., 2020, Lemma 10]) With probability at least $1 - 2\delta$, we have for all $k = 0, \ldots L - 1$,*

$$\sum_{t=1}^{T} \sum_{s \in S_k, a \in A} \frac{q_t(s,a)}{\max\{1, m_{i(t)}(s,a)\}} = \mathcal{O}\left(|S_k||A| \ln T + \ln(L/\delta)\right) \tag{68}$$

*and*

$$\sum_{t=1}^{T} \sum_{s \in S_k, a \in A} \frac{q_t(s,a)}{\sqrt{\max\{1, m_{i(t)}(s,a)\}}} = \mathcal{O}\left(\sqrt{|S_k||A|T} + |S_k||A| \ln T + \ln(L/\delta)\right). \tag{69}$$

*Simultaneously, for all $k < h$, we have*

$$\sum_{t=1}^{T} \sum_{(u,v,w) \in T_k} \sum_{(x,y,z) \in T_h} q_t(u,v)\sqrt{\frac{P(w|u,v)}{\max\{1, m_{i(t)}(u,v)\}}} \cdot q_t(x,y|w)\sqrt{\frac{P(z|x,y)}{\max\{1, m_{i(t)}(x,y)\}}} \tag{70}$$
$$= \mathcal{O}\left((|A| \ln T + \ln(L/\delta)) \cdot \sqrt{|S_k||S_{k+1}||S_h||S_{h+1}|}\right).$$

*Proof.* Eq. (68) and Eq. (69) are from Jin et al. [2020]. For Eq. (70), by direct calculation we have

$$\sum_{t=1}^{T} \sum_{(u,v,w) \in T_k} \sum_{(x,y,z) \in T_h} q_t(u,v)\sqrt{\frac{P(w|u,v)}{\max\{1, m_{i(t)}(u,v)\}}} \cdot q_t(x,y|w)\sqrt{\frac{P(z|x,y)}{\max\{1, m_{i(t)}(x,y)\}}}$$

$$= \sum_{t=1}^{T} \sum_{(u,v,w) \in T_k} \sum_{(x,y,z) \in T_h} \sqrt{\frac{q_t(u,v)P(z|x,y)q_t(x,y|w)}{\max\{1, m_{i(t)}(u,v)\}}} \cdot \sqrt{\frac{q_t(u,v)P(w|u,v)q_t(x,y|w)}{\max\{1, m_{i(t)}(x,y)\}}}$$

$$\leq \sqrt{\sum_{t=1}^{T} \sum_{(u,v,w) \in T_k} \sum_{(x,y,z) \in T_h} \frac{q_t(u,v)P(z|x,y)q_t(x,y|w)}{\max\{1, m_{i(t)}(u,v)\}}} \cdot \sqrt{\sum_{t=1}^{T} \sum_{(u,v,w) \in T_k} \sum_{(x,y,z) \in T_h} \frac{q_t(u,v)P(w|u,v)q_t(x,y|w)}{\max\{1, m_{i(t)}(x,y)\}}}$$

$$\leq \sqrt{|S_{k+1}| \sum_{t=1}^{T} \sum_{u \in S_k} \sum_{a \in A} \frac{q_t(u,v)}{\max\{1, m_{i(t)}(u,v)\}}} \cdot \sqrt{|S_{h+1}| \sum_{t=1}^{T} \sum_{x \in S_h} \sum_{a \in A} \frac{q_t(x,y)}{\max\{1, m_{i(t)}(x,y)\}}}$$

$$\leq \mathcal{O}\left( (|A| \ln T + \ln(L/\delta)) \cdot \sqrt{|S_k| |S_{k+1}| |S_h| |S_{h+1}|} \right).$$

$\square$

**Lemma D.3.8.** *For all $k = 0, \ldots, L-1$, we have*

$$\mathbb{E}\left[ \sum_{t=1}^{T} \sum_{s \in S_k, a \in A} \frac{q_t(s,a)}{\max\{1, m_{i(t)}(s,a)\}} \right] = \mathcal{O}\left( |S_k||A| \ln T + |S_k||A| \right) \tag{71}$$

*and*

$$\mathbb{E}\left[ \sum_{t=1}^{T} \sum_{s \in S_k, a \in A} \frac{q_t(s,a)}{\sqrt{\max\{1, m_{i(t)}(s,a)\}}} \right] = \mathcal{O}\left( \sqrt{|S_k||A|T} + |S_k||A| \right). \tag{72}$$

*Proof.* For each state-action pair $(s,a)$, we have

$$\mathbb{E}\left[ \sum_{t=1}^{T} \frac{q_t(s,a)}{\max\{1, m_{i(t)}(s,a)\}} \right]$$

$$= \mathbb{E}\left[ \sum_{t=1}^{T} \frac{\mathbb{I}_t(s,a)}{\max\{1, m_{i(t)}(s,a)\}} \right] = \mathbb{E}\left[ \sum_{i=1}^{N} \sum_{t=t_i}^{t_{i+1}-1} \frac{\mathbb{I}_t(s,a)}{\max\{1, m_i(s,a)\}} \right]$$

$$= \mathbb{E}\left[ \sum_{i=1}^{N} \frac{m_{i+1}(s,a) - m_i(s,a)}{\max\{1, m_i(s,a)\}} \right]$$

$$\leq 2\mathbb{E}\left[ 1 + \int_{1}^{1+m_{N+1}(s,a)} \frac{dx}{x} \right] \leq 2\left( 2 \ln T + 1 \right)$$

where the second line follows from the definition of the indicator and occupancy measure $q_t$, and the last line applies the fact $m_{i+1}(s,a) \leq 2m_i(s,a)$ when $m_i(s,a) \geq 1$. Taking the summation over all state-action pairs at layer $k$ finishes the proof of Eq. (71).

Similarly, we have

$$\mathbb{E}\left[ \sum_{t=1}^{T} \frac{q_t(s,a)}{\sqrt{\max\{1, m_{i(t)}(s,a)\}}} \right]$$

$$= \mathbb{E}\left[ \sum_{t=1}^{T} \frac{\mathbb{I}_t(s,a)}{\sqrt{\max\{1, m_{i(t)}(s,a)\}}} \right] = \mathbb{E}\left[ \sum_{i=1}^{N} \sum_{t=t_i}^{t_{i+1}-1} \frac{\mathbb{I}_t(s,a)}{\sqrt{\max\{1, m_i(s,a)\}}} \right]$$

$$= \mathbb{E}\left[ \sum_{i=1}^{N} \frac{m_{i+1}(s,a) - m_i(s,a)}{\sqrt{\max\{1, m_i(s,a)\}}} \right]$$

$$\leq 2\mathbb{E}\left[ 1 + \int_{0}^{m_{N+1}(s,a)} \frac{dx}{\sqrt{x}} \right] \leq 2\left( 2\sqrt{m_{N+1}(s,a)} + 1 \right)$$

where $m_{N+1}(s,a)$ is the total number of visiting state-action pair $(s,a)$. Taking the summation over all state-action pairs of layer $k$ yields that

$$\mathbb{E}\left[ \sum_{s \in S_k} \sum_{a \in A} \sum_{t=1}^{T} \frac{q_t(s,a)}{\sqrt{\max\{1, m_{i(t)}(s,a)\}}} \right]$$

$$\leq \sum_{s \in S_k} \sum_{a \in A} 2\left( 2\sqrt{m_{N+1}(s,a)} + 1 \right) \leq 2\left( 2\sqrt{|S_k||A|T} + |S_k||A| \right)$$

where the last inequality follows from the Cauchy-Schwarz inequality.

$\square$

**Definition D.3.9.** *(Residual Term) We define the residual term $r_t(s,a)$ as*

$$r_t(s,a) = \frac{40}{3} \sum_{k=0}^{k(s)-1} \sum_{(u,v,w)\in T_k} q_t(u,v) \cdot \frac{P(w|u,v)\ln\left(\frac{T|S||A|}{\delta}\right)}{\max\{m_{i(t)}(u,v),1\}} \cdot q_t(s,a|w)$$

$$+ \sum_{k=0}^{k(s)-1} \sum_{h=k+1}^{k(s)-1} \sum_{(u,v,w)\in T_k} \sum_{(x,y,z)\in T_h} q_t(u,v)B_{i(t)}(u,v,w)q_t(x,y|w)B_{i(t)}(x,y,z) \tag{73}$$

$$+ \mathbb{I}\{\mathcal{A}^c\}.$$

*for all state-action pair $(s,a) \in S \times A$ and all episodes $t \in [T]$.*

**Lemma D.3.10.** *The following hold:*

$$|q_t(s,a) - \widehat{q}_t(s,a)| \le r_t(s,a) + 4 \sum_{k=0}^{k(s)-1} \sum_{(u,v,w)\in T_k} q_t(u,v)\sqrt{\frac{P(w|u,v)\ln\iota}{\max\{m_{i(t)}(u,v),1\}}} q_t(s,a|w)$$

*and*

$$\mathbb{E}\left[\sum_{t=1}^{T}\sum_{s\neq s_L}\sum_{a\in A} r_t(s,a)\right] = \mathcal{O}\left(L^2|S|^3|A|^2\ln^2\left(\frac{T|S||A|}{\delta}\right) + |S||A|T\cdot\delta\right).$$

*Proof.* For simplicity, we let $\iota = \frac{T|S||A|}{\delta}$ and assume $\delta \in (0,1)$. According to the Lemma D.3.1, conditioning on event $\mathcal{A}$, we have

$$|q_t(s,a) - \widehat{q}_t(s,a)| = \left|\sum_{k=0}^{k(s)-1} \sum_{(u,v,w)\in T_k} q_t(u,v)\left(P(w|u,v) - \bar{P}_{i(t)}(w|u,v)\right)\widehat{q}_t(s,a|w)\right|$$

$$\le \sum_{k=0}^{k(s)-1} \sum_{(u,v,w)\in T_k} q_t(u,v)\left|P(w|u,v) - \bar{P}_{i(t)}(w|u,v)\right|\widehat{q}_t(s,a|w)$$

$$\le \sum_{k=0}^{k(s)-1} \sum_{(u,v,w)\in T_k} q_t(u,v)B_{i(t)}(u,v,w)\widehat{q}_t(s,a|w)$$

Moreover, we apply Lemma D.3.1 again to conditional occupancy measure and obtain

$$|q_t(s,a|w) - \widehat{q}_t(s,a|w)| \le \sum_{h=k(w)}^{k(s)-1} \sum_{(x,y,z)\in T_h} q_t(x,y|w)B_{i(t)}(x,y,z)\widehat{q}_t(s,a|z)$$

$$\le \sum_{h=k(w)}^{k(s)-1} \sum_{(x,y,z)\in T_h} q_t(x,y|w)B_{i(t)}(x,y,z)$$

where the second line applies the fact $\widehat{q}_t(s,a|z) \le 1$.

Combining these inequalities yields (under the event $\mathcal{A}$)

$$|q_t(s,a) - \widehat{q}_t(s,a)|$$

$$\le \sum_{k=0}^{k(s)-1} \sum_{(u,v,w)\in T_k} q_t(u,v)B_{i(t)}(u,v,w)q_t(s,a|w)$$

$$+ \sum_{k=0}^{k(s)-1} \sum_{(u,v,w)\in T_k} q_t(u,v)B_{i(t)}(u,v,w)\left(\sum_{h=k(w)}^{k(s)-1} \sum_{(x,y,z)\in T_h} q_t(x,y|w)B_{i(t)}(x,y,z)\right)$$

$$\le 4 \sum_{k=0}^{k(s)-1} \sum_{(u,v,w)\in T_k} q_t(u,v)\sqrt{\frac{P(w|u,v)\ln\iota}{\max\{m_{i(t)}(u,v),1\}}} q_t(s,a|w)$$

$$+ \frac{40}{3} \sum_{k=0}^{k(s)-1} \sum_{(u,v,w)\in T_k} q_t(u,v) \cdot \frac{P(w|u,v)\ln\iota}{\max\left\{m_{i(t)}(u,v),1\right\}} \cdot q_t(s,a|w)$$

$$+ \sum_{k=0}^{k(s)-1} \sum_{h=k+1}^{k(s)-1} \sum_{(u,v,w)\in T_k} \sum_{(x,y,z)\in T_h} q_t(u,v)B_{i(t)}(u,v,w)q_t(x,y|w)B_{i(t)}(x,y,z)$$

where the second line follows from Lemma D.3.3.

On the other hand, $|q_t(s,a) - \widehat{q}_t(s,a)| \leq 1$ holds always. Combining the bounds of these two cases finishes the first statement.

Recall the definition of the residual terms, we decompose the following into three terms $\text{SUM}_1$, $\text{SUM}_2$ and $\text{SUM}_3$:

$$\mathbb{E}\left[\sum_{t=1}^{T}\sum_{s\neq s_L}\sum_{a\in A} r_t(s,a)\right]$$

$$= \frac{40}{3}\mathbb{E}\underbrace{\left[\sum_{t=1}^{T}\sum_{s\neq s_L}\sum_{a\in A}\sum_{k=0}^{k(s)-1}\sum_{(u,v,w)\in T_k} q_t(u,v)\cdot \frac{P(w|u,v)\ln\iota}{\max\left\{m_{i(t)}(u,v),1\right\}}\cdot q_t(s,a|w)\right]}_{\triangleq \text{SUM}_1}$$

$$+ \mathbb{E}\underbrace{\left[\sum_{t=1}^{T}\sum_{s\neq s_L}\sum_{a\in A}\mathbb{I}\{\mathcal{A}^c\}\right]}_{\triangleq \text{SUM}_2}$$

$$+ \mathbb{E}\underbrace{\left[\sum_{t=1}^{T}\sum_{s\neq s_L}\sum_{a\in A}\sum_{k=0}^{k(s)-1}\sum_{h=k+1}^{k(s)-1}\sum_{(u,v,w)\in T_k}\sum_{(x,y,z)\in T_h} q_t(u,v)B_{i(t)}(u,v,w)q_t(x,y|w)B_{i(t)}(x,y,z)\right]}_{\triangleq \text{SUM}_3}.$$

Then, we show that these terms are all logarithmic in $T$.

**SUM$_1$** By direct calculation, we have

$$\text{SUM}_1 = \frac{40}{3}\mathbb{E}\left[\sum_{t=1}^{T}\sum_{s\neq s_L}\sum_{a\in A}\sum_{k=0}^{k(s)-1}\sum_{(u,v,w)\in T_k} q_t(u,v)\cdot \frac{P(w|u,v)\ln\iota}{\max\left\{m_{i(t)}(u,v),1\right\}}\cdot q_t(s,a|w)\right]$$

$$= \frac{40}{3}\mathbb{E}\left[\sum_{t=1}^{T}\sum_{k=0}^{L-1}\sum_{(u,v,w)\in T_k} q_t(u,v)\cdot \frac{\ln\iota}{\max\left\{m_{i(t)}(u,v),1\right\}}\cdot\left(\sum_{s\neq s_L}\sum_{a\in A} P(w|u,v)q_t(s,a|w)\right)\right]$$

$$\leq \frac{40L}{3}\ln\iota\,\mathbb{E}\left[\sum_{t=1}^{T}\sum_{u\neq s_L}\sum_{v\in A}\cdot\frac{q_t(u,v)}{\max\left\{m_{i(t)}(u,v),1\right\}}\right]$$

$$= \frac{80L}{3}\ln\iota\left(\sum_{k=0}^{L-1} |S_k||A|\,(\ln T+1)\right) = \mathcal{O}\left(L|S||A|\ln^2\iota\right) \tag{74}$$

where the first line follows from the property of occupancy measures, and the last line applies Eq. (71) of Lemma D.3.8.

**SUM$_2$** According to the definition of event $\mathcal{A}$, we have

$$\text{SUM}_2 = \mathbb{E}\left[\sum_{t=1}^{T}\sum_{s\neq s_L}\sum_{a\in A}\mathbb{I}\{\mathcal{A}^c\}\right] = |S||A|T\cdot\mathbb{E}\left[\mathbb{I}\{\mathcal{A}^c\}\right] = |S||A|T\cdot\delta. \tag{75}$$

**SUM$_3$**  First, we consider the term inside the expectation bracket and show the following conditioning on event $\mathcal{A}$:

$$\sum_{t=1}^{T} \sum_{s \neq s_L} \sum_{a \in A} \sum_{k=0}^{k(s)-1} \sum_{h=k+1}^{k(s)-1} \sum_{(u,v,w) \in T_k} \sum_{(x,y,z) \in T_h} q_t(u,v) B_{i(t)}(u,v,w) q_t(x,y|w) B_{i(t)}(x,y,z)$$

$$\leq 4 \sum_{t=1}^{T} \sum_{s \neq s_L} \sum_{a \in A} \sum_{k=0}^{k(s)-1} \sum_{h=k+1}^{k(s)-1} \sum_{(u,v,w) \in T_k} \sum_{(x,y,z) \in T_h} q_t(u,v) \sqrt{\frac{P(w|u,v)\ln \iota}{\max\{m_{i(t)}(u,v),1\}}} q_t(x,y|w) B_{i(t)}(x,y,z)$$

$$+ \frac{40}{3} \sum_{t=1}^{T} \sum_{s \neq s_L} \sum_{a \in A} \sum_{k=0}^{k(s)-1} \sum_{h=k+1}^{k(s)-1} \sum_{(u,v,w) \in T_k} \sum_{(x,y,z) \in T_h} q_t(u,v) \left( \frac{P(w|u,v)\ln \iota}{\max\{m_{i(t)}(u,v),1\}} \right) q_t(x,y|w) B_{i(t)}(x,y,z)$$

$$\leq 16|S||A|\ln \iota \sum_{t=1}^{T} \sum_{k<h} \sum_{(u,v,w) \in T_k} \sum_{(x,y,z) \in T_h} q_t(u,v) \sqrt{\frac{P(w|u,v)}{\max\{m_{i(t)}(u,v),1\}}} q_t(x,y|w) \sqrt{\frac{P(z|x,y)}{\max\{m_{i(t)}(x,y),1\}}}$$

$$+ \frac{160|S||A|}{3} \sum_{t=1}^{T} \sum_{k<h} \sum_{(u,v,w) \in T_k} \sum_{(x,y,z) \in T_h} q_t(u,v) \sqrt{\frac{P(w|u,v)\ln \iota}{\max\{m_{i(t)}(u,v),1\}}} q_t(x,y|w) \min\left\{ \frac{P(z|x,y)\ln \iota}{\max\{m_{i(t)}(x,y),1\}}, 1 \right\}$$

$$+ \frac{40|S||A|}{3} \sum_{t=1}^{T} \sum_{k<h} \sum_{(u,v,w) \in T_k} \sum_{(x,y,z) \in T_h} q_t(u,v) \left( \frac{P(w|u,v)\ln \iota}{\max\{m_{i(t)}(u,v),1\}} \right) q_t(x,y|w)$$

where the second inequality follows from [Lemma D.3.3](#) and [Corollary D.3.4](#).

Then we consider bounding these three different terms with the help of previous analysis. According to [Eq. (70)](#) of [Lemma D.3.7](#), The first term is bounded with probability at least $1 - 2\delta'$:

$$16|S||A|\ln \iota \sum_{t=1}^{T} \sum_{k<h} \sum_{(u,v,w) \in T_k} \sum_{(x,y,z) \in T_h} q_t(u,v) \sqrt{\frac{P(w|u,v)}{\max\{m_{i(t)}(u,v),1\}}} q_t(x,y|w) \sqrt{\frac{P(z|x,y)}{\max\{m_{i(t)}(x,y),1\}}}$$

$$\leq 16|S||A|\ln \iota \cdot \mathcal{O}\left( (|A|\ln T + \ln(L/\delta')) \sum_{k<h} \sqrt{|S_k||S_{k+1}||S_h||S_{h+1}|} \right)$$

$$\leq 16|S||A|\ln \iota \cdot \mathcal{O}\left( (|A|\ln T + \ln(L/\delta')) \sum_{k<h} (|S_k||S_{k+1}| + |S_h||S_{h+1}|) \right)$$

$$\leq \mathcal{O}\left( (|A|\ln T + \ln(L/\delta')) L|S|^3|A|\ln \iota \right),$$

where the third line follows from the AM-GM inequality. Taking the expectation with $\delta' = \frac{L}{\iota}$, we have the expectation of the first term bounded by $\mathcal{O}\left( L|S|^3|A|^2\ln^2 \iota \right)$ using [Lemma D.3.5](#).

On the other hand, for the second term, we have

$$\frac{160|S||A|}{3} \sum_{t=1}^{T} \sum_{k<h} \sum_{(u,v,w) \in T_k} \sum_{(x,y,z) \in T_h} q_t(u,v) \sqrt{\frac{P(w|u,v)\ln \iota}{\max\{m_{i(t)}(u,v),1\}}} q_t(x,y|w) \min\left\{ \frac{P(z|x,y)\ln \iota}{\max\{m_{i(t)}(x,y),1\}}, 1 \right\}$$

$$\leq \frac{80|S||A|}{3} \sum_{t=1}^{T} \sum_{k<h} \sum_{(u,v,w) \in T_k} \sum_{(x,y,z) \in T_h} q_t(u,v) P(w|u,v) q_t(x,y|w) \left( \frac{P(z|x,y)\ln \iota}{\max\{m_{i(t)}(x,y),1\}} \right)$$

$$+ \frac{80|S||A|}{3} \sum_{t=1}^{T} \sum_{k<h} \sum_{(u,v,w) \in T_k} \sum_{(x,y,z) \in T_h} q_t(u,v) \frac{\ln \iota}{\max\{m_{i(t)}(u,v),1\}} q_t(x,y|w)$$

$$\leq \frac{80L|S||A|}{3} \ln \iota \sum_{t=1}^{T} \sum_{x \in S} \sum_{y \in A} \left( \frac{q_t(x,y)}{\max\{m_{i(t)}(x,y),1\}} \right)$$

$$+ \frac{80L|S|^2|A|}{3} \ln \iota \sum_{t=1}^{T} \sum_{u \neq s_L} \sum_{v \in A} \left( \frac{q_t(u,v)}{\max\{m_{i(t)}(u,v),1\}} \right)$$

$$\leq \frac{160L|S|^2|A|}{3} \ln \iota \sum_{t=1}^{T} \sum_{u \neq s_L} \sum_{v \in A} \frac{q_t(u,v)}{\max\left\{m_{i(t)}(u,v),1\right\}}$$

where the expectation of the final term is bounded $\mathcal{O}\left(L|S|^3|A|^2 \ln^2 \iota\right)$ with the help from Lemma D.3.8. Similarly, we have the expectation of the third term bounded by $\mathcal{O}\left(L|S|^3|A|^2 \ln \iota\right)$ following the same idea.

Therefore, we have $\text{SUM}_3$ bounded as

$$\begin{aligned}
\text{SUM}_3 &= \mathcal{O}\left(L|S|^3|A|^2 \ln^2 \iota + L|S|^3|A|^2 \ln^2 \iota + |S||A|T \cdot \delta\right) \\
&= \mathcal{O}\left(L|S|^3|A|^2 \ln^2 \iota + |S||A|T \cdot \delta\right)
\end{aligned} \tag{76}$$

where the $|S||A|T \cdot \delta$ comes from the range of $\text{SUM}_3$ and the probability of event $\mathcal{A}^c$.

Combining the bounds of $\text{SUM}_1$, $\text{SUM}_2$, and $\text{SUM}_3$ stated in Eq. (74), Eq. (75) and Eq. (76) finishes the proof. $\qquad \square$

**Corollary D.3.11.** *The following holds:*

$$|q_t(s,a) - u_t(s,a)| \leq 4r_t(s,a) + 16 \sum_{k=0}^{k(s)-1} \sum_{(u,v,w) \in T_k} q_t(u,v) \sqrt{\frac{P(w|u,v) \ln\left(\frac{T|S||A|}{\delta}\right)}{\max\left\{m_{i(t)}(u,v),1\right\}}} q_t(s,a|w).$$

*where $q_t$ is the true occupancy measure of episode $t$, and $u_t$ is the upper occupancy bound of episode $t$ associated with confidence set $\mathcal{P}_{i(t)}$ and policy $\pi_t$.*

*Proof.* Fix the state-action pair $(s,a)$ and episode $t$. Let $\widehat{P}$ be the transition in $\mathcal{P}_{i(t)}$ that realizes the maximum in the definition of $u_t(s,a)$, and $\widetilde{q}_t = q^{\widehat{P},\pi_t}$ bet the associated occupancy measure. Therefore, we have $\widetilde{q}_t(s,a) = u_t(s,a)$.

Conditioning on event $\mathcal{A}$, we have

$$\begin{aligned}
|q_t(s,a) - \widetilde{q}_t(s,a)| &= \left| \sum_{k=0}^{k(s)-1} \sum_{(u,v,w) \in T_k} q_t(u,v) \left(P(w|u,v) - \widehat{P}(w|u,v)\right) \widetilde{q}_t(s,a|w) \right| \\
&\leq \sum_{k=0}^{k(s)-1} \sum_{(u,v,w) \in T_k} q_t(u,v) \left| P(w|u,v) - \widehat{P}(w|u,v) \right| \widetilde{q}_t(s,a|w) \\
&\leq 2 \sum_{k=0}^{k(s)-1} \sum_{(u,v,w) \in T_k} q_t(u,v) B_{i(t)}(u,v,w) \widetilde{q}_t(s,a|w).
\end{aligned}$$

Moreover, we apply Lemma D.3.1 to terms $\widehat{q}_t(s,a|w)$ and obtain

$$\begin{aligned}
|q_t(s,a|w) - \widetilde{q}_t(s,a|w)| &\leq 2 \sum_{h=k(w)}^{k(s)-1} \sum_{(x,y,z) \in T_h} q_t(x,y|w) B_{i(t)}(x,y,z) \widetilde{q}_t(s,a|z) \\
&\leq 2 \sum_{h=k(w)}^{k(s)-1} \sum_{(x,y,z) \in T_h} q_t(x,y|w) B_{i(t)}(x,y,z)
\end{aligned}$$

where the second line uses $\widehat{q}_t(s,a|z) \leq 1$.

Combining these inequalities yields (under the event $\mathcal{A}$)

$$\begin{aligned}
&|q_t(s,a) - \widehat{q}_t(s,a)| \\
&\leq 4 \sum_{k=0}^{k(s)-1} \sum_{(u,v,w) \in T_k} q_t(u,v) B_{i(t)}(u,v,w) q_t(s,a|w)
\end{aligned}$$

$$+ 4 \sum_{k=0}^{k(s)-1} \sum_{(u,v,w) \in T_k} q_t(u,v) B_{i(t)}(u,v,w) \left( \sum_{h=k(w)}^{k(s)-1} \sum_{(x,y,z) \in T_h} q_t(x,y|w) B_{i(t)}(x,y,z) \right)$$

$$\leq 16 \sum_{k=0}^{k(s)-1} \sum_{(u,v,w) \in T_k} q_t(u,v) \sqrt{\frac{P(w|u,v) \ln \iota}{\max \{ m_{i(t)}(u,v), 1 \}}} q_t(s,a|w)$$

$$+ \frac{160}{3} \sum_{k=0}^{k(s)-1} \sum_{(u,v,w) \in T_k} q_t(u,v) \cdot \frac{P(w|u,v) \ln \iota}{\max \{ m_{i(t)}(u,v), 1 \}} \cdot q_t(s,a|w)$$

$$+ 4 \sum_{k=0}^{k(s)-1} \sum_{h=k+1}^{k(s)-1} \sum_{(u,v,w) \in T_k} \sum_{(x,y,z) \in T_h} q_t(u,v) B_{i(t)}(u,v,w) q_t(x,y|w) B_{i(t)}(x,y,z)$$

where the second line follows from Lemma D.3.3.

On the other hand, $|q_t(s,a) - \widetilde{q}_t(s,a)| \leq 1$ holds always. Combining the bounds of these two cases finishes the proof. $\qquad \square$

**Lemma D.3.12.** *Algorithm 1 ensures $N \leq 4|S||A| (\log T + 1)$ where $N$ is the number of epochs.*

*Proof.* For a fixed state-action pair $(s,a)$, let the $i_1 \leq i_2 \leq \ldots \leq i_k$ denotes the epochs that triggered by this state-action pair, that is

$$\{i_1, i_2, \ldots, i_k\} = \{i : i \in 1, \ldots N, m_i(s,a) \geq \max \{1, 2 \cdot m_{i-1}(s,a)\}\}.$$

Clearly, it holds that

$$1 = m_{i_1}(s,a), \text{ and } m_{i_\tau}(s,a) \geq 2 m_{i_{\tau-1}}(s,a) \tau \in 2, \ldots, k$$

which indicates that $m_{i_k}(s,a) \geq 2^{k-1}$. Combining with the fact that $m_{i_k}(s,a) \leq T$, we have

$$k = |\{i_1, i_2, \ldots, i_k\}| \leq 4 \log T + 4.$$

Taking the summation over all state-action pairs finishes the proof. $\qquad \square$