# OpenReview forum: "The best of both worlds: stochastic and adversarial episodic MDPs with unknown transition"
_NeurIPS.cc/2021/Conference — NeurIPS 2021 Oral_

### Official Review · Reviewer_VSsV · 2021-07-13

**Rating:** 7
**Confidence:** 3

**Summary:**

This paper studies the "best of both worlds" algorithm for episodic MDPs with unknown transition with stochastic and adversarial rewards, hence the paper resolves the open question in (Jin and Luo 2020). While the adversarial case alone with unknown transition is closed in (Jin et al. 2020), the paper shows that a simple variation of FTRL with a loss-shifting trick, optimism-based exploration bonus and restarting can also achieve poly-logarithmic regret in stochastic environments assuming the constant minimum-gap $\Delta_{min} = \min_{s, a \neq \pi^*(s)} V^*(s) - Q^*(s,a)$. The proposed algorithm and associated analysis enjoy more simplified analysis compared to the previous work in (Jin and Luo 2020). The simplified analysis is adapted from (Jin and Luo 2020) and (Simchowitz and Jamieson 2019). Main results are the the regret bound of $O(\sqrt{T})$ under adversarial setting, and $O(log^2 (T))$ under stochastic setting.

**Limitations And Societal Impact:**

Limitations are not separately addressed in the main paper. Here are a few of my questions:

- Do you think the regret bound can be improved to $O(log T)$ in the stochastic cases in (Simchowitz and Jamieson 2019)?

- Can you improve the regret bound by a factor of $|S|$ in the adversarial case?

**Main Review:**

In some sense, the contribution may be seen as a minor improvement from (Jin and Luo 2020) for known transition case. In fact, maybe one can first run reward-free exploration schemes to learn transition kernels and then execute any known algorithms for best-of-both-worlds. It would result in O(T^{2/3})-regret nevertheless. Hence getting $O(\sqrt{T})$ and $O(polylog(T))$ regrets in adversarial and stochastic cases respectively does not seem to be so trivial.

On the other hand, analysis seems not particularly novel but a variation of FTRL method (e.g., Jin et al. 2020) and logarithmic-regret for MDPs (e.g., Simchowitz and Jamieson 2019). Nevertheless, they are well-adapted to a challenging problem of achieving best of both worlds with unknown transition kernels.

Overall the paper is well-written and easy to follow. Comparisons to previous work are well discussed whenever needed. Main results and analysis are well-presented, hence I recommend the acceptance for this paper.



There are some points I want to be clarified.

- In Section 3, the main idea seems to add additional term (loss-shifting?) $g_\tau$. However, I do not see what conceptually plays the role of $g_\tau$ in Algorithm 1. (Eq (11) is, in my interpretation, simply an adjusted loss with standard exploration bonus). Could you be more specific about the connection?

- Line 231-234: I do not understand what it does mean by "This framework, however, does not allow the loss-shifting trick discussed in Section 3" (or why it has to be).

- Line 293-294 "In fact, ... complicated": at least, could you explain which improvement you have or can be made?



I also raise some other questions on the analysis

- Could you be more specific on your technical novelty? In which part of the analysis is different from (Jin et al. 2020) for adversarial case?

- How is the regret decomposition for the stochastic case different from (Simchowitz and Jamieson 2019)? (or are they the same?)

- Line 345-346 "In fact, the reason ... terms":  I don't understand what you mean by $Reg_T(\pi^o)$ cannot be directly bounded - isn't your theorem bounds it by $O(\sqrt{T})$?





Minor Comments

- Line 259: "in the bandit case" -> do you mean in the bandit-feedback case?

- Line 137: $\Delta(s,a)$ is not defined before used here.

- Line 177: "this natural and simple candidate" - do you mean the one used in (Zimmert and Seldin 2019)?

**Time Spent Reviewing:**

6

---

> ### Author Response · Authors · 2021-08-10
> **Response to Reviewer VSsV**
>
> We thank the reviewer for the valuable feedback.
>
> 1\. *maybe one can first run reward-free exploration schemes to learn transition kernels and then execute any known algorithms for best-of-both-worlds.*
>
> We want to emphasize that this completely defeats the purpose of obtaining best-of-both-world guarantees, as it results in $T^{2/3}$ regret for both the adversarial setting and the stochastic setting.
>
> 2\. *In Section 3, the main idea seems to add additional term (loss-shifting?) However, I do not see what conceptually plays the role of in Algorithm 1. (Eq (11) is, in my interpretation, simply an adjusted loss with standard exploration bonus). Could you be more specific about the connection?*
>
> The loss shifting technique does not require actually adding the loss-shifting function explicitly in the algorithm (as shown in Eq.(5)). This is only an analytic tool that enables us to apply several useful properties (e.g., Lemma A.1.3 for the known transition setting), which is critical to achieve the best-of-both-worlds guarantee.
>
> 3\. *Line 231-234: I do not understand what it does mean by "This framework, however, does not allow the loss-shifting trick discussed in Section 3" (or why it has to be).*
>
> As we explain immediately after that sentence, our loss-shifting function is defined with respect to a fixed transition, and the important invariant Eq.(5) (or Corollary A.1.2) would not hold if we use the standard framework which performs FTRL over the larger set $\Omega(\mathcal{P}_i)$. In other words, the standard method does not allow analyzing the regret using the imaginary additional loss-shifting function, forcing us to come up with a different framework.
>
> 4\. *Line 293-294 "In fact, ... complicated": at least, could you explain which improvement you have or can be made?*
>
> We are able to reduce the dependency of L for the stochastic world by carefully separating the state-action pairs into those that are visited often enough and those that are not.
>
> 5\. *Could you be more specific on your technical novelty? In which part of the analysis is different from (Jin et al. 2020) for adversarial case? How is the regret decomposition for the stochastic case different from (Simchowitz and Jamieson 2019)? (or are they the same?)*
>
> We reiterate our technical novelty stated in Lines 40-71 (and explained further in other sections): First, we propose a novel loss-shifting trick for the FTRL analysis for MDPs with fixed transition, which greatly simplifies the approach of Jin and Luo (2020) for the known transition case and lays the foundation for the analysis of unknown transition. Second, we design a framework for the unknown transition under adversarial losses that enables the application of the same loss-shifting trick. Third, we develop an analysis to show that in the stochastic world the transition estimation error of our algorithm is only polylogarithmic in T with the help of a new regret decomposition (Lemma D.1.1 and Corollary D.1.2) and several self-bounding terms (Appendix D.2).
>
> For the adversarial world, although our algorithm is different from Jin et al. (2020), the analysis is very similar to theirs, especially for bounding the ERR_1 term as we point out in Lines 310-311.
>
> However, the key challenge is on handling to the stochastic world.
> Although our regret decomposition is inspired by Simchowitz and Jamieson (2019), they are not the same. Our decomposition (Lemma D.1.1) holds generally and is agnostic to specific algorithms, while theirs is very specific to the so-called "strongly optimistic" algorithms (see their Appendix D). In fact, they do not really have an equality decomposition but only an upper bound after applying a certain clipping trick. Our decomposition does not rely on the clipping trick, and instead makes use of a novel quantity $q_\pi^\star(s,a)$, the probability of visiting a trajectory of the form $(s_0, \pi^\star(s_0)), (s_1, \pi^\star(s_1)), \ldots, (s_{k(s)-1}, \pi^\star(s_{k(s)-1})), (s,a)$ when executing policy $\pi_t$.
>
> 6\.*Line 345-346 "In fact, the reason ... terms": I don't understand what you mean by $\text{Reg}_T(\mathring{\pi})$ cannot be directly bounded - isn't your theorem bounds it by $\mathcal{O}(\sqrt{T})$?*
>
> Sorry for the confusion. Here, we mean that $\text{Reg}_T(\mathring{\pi})$ cannot be bounded  by $\text{polylog}(T)$ using our proposed techniques.
>
>
> 7\. *Line 259: "in the bandit case" -> do you mean in the bandit-feedback case?*
>
> Yes.
>
> 8\. *Line 137: $\Delta(s,a)$ is not defined before used here.*
>
> Actually, $\Delta(s,a)$ is defined in Condition (1) already.
>
> 9\. *Line 177: "this natural and simple candidate" - do you mean the one used in (Zimmert and Seldin 2019)?*
>
> Yes.
>
> 10\. *Do you think the regret bound can be improved to $O(\log T)$ in the stochastic cases in (Simchowitz and Jamieson2019)?*
>
> We believe that this is possible, but we do not know how to achieve this at the moment. The extra $\log(T)$ factor of our bound comes from the fact that we run $\log(T)$ independent instances of FTRL with a fixed transition, which is currently crucial for our loss-shifting technique.
>
> 11\. *Can you improve the regret bound by a factor of $S$ in the adversarial case?*
>
> Note that our bound is only $\sqrt{S}$ (not $S$) away from the best lower bound. Please see more discussions on this in our second response to reviewer o6kk.

---

### Official Review · Reviewer_WK45 · 2021-07-16

**Rating:** 8
**Confidence:** 3

**Summary:**

The authors consider the problem of stochastic and adversarial episodic MDPs with unknown transition, is an extension of the work of Jin and Luo (2020), with unknown transitions rather than known ones.

They propose an algorithm which is part of the FTRL framework, where they use some existing tools, such as the self-bounding technique as well as new tools such as the Loss-Shifting Technique.

They outperform the previous results from Jin and Luo (2020) for the problem of stochastic and adversarial episodic MDPs with known transition with a simpler analysis, and they derive new bounds in the setting with unknown transitions.

Both Full information and Bandit feedback are coinsidered.


**Limitations And Societal Impact:**

The limitations are supposedly discussed in section 1, but they could be clearer and extended. In particular, a clearer description of the suboptimality gaps of the provided bounds would be beneficial.

The societal impact is not super relevant for this type of theoretical problems, and in general achieving more robust and polyvalent algorithms has a positive impact.


**Main Review:**

This work is an improvement compared to previous works on several fronts. First, in the setting with fixed transitions, this new algorithm and its analysis are both more efficient and simpler to study.
In the setting with unknown transition, the results are novel.

Overall, the framework provided is very complete, as they study episodic MDPs in a Best of Both world setting, both with known and unknown transitions and both with bandit and full information feedback.

The algorithm proposed is part of the FTRL framework, where they use some  existing tools, such as the self-bounding technique as well as new tools such as the Loss-Shifting Technique, which is efficient and easier to analyse than the previously existing algorithm from Jin and Luo (2020).

The paper is quite well written, and the proofs are extensively detailled in the appendix. The amount of material presented is important for the length of the paper.
 A conclusion that discuss possible extensions would be a useful addition.

**Time Spent Reviewing:**

12

---

> ### Author Response · Authors · 2021-08-10
> **Response to Reviewer WK45**
>
> We thank the reviewer for the valuable feedback.  We will incorporate your minor comments on writing in the next version.

---

### Official Review · Reviewer_o6kk · 2021-07-16

**Rating:** 8
**Confidence:** 4

**Summary:**

This paper presents an algorithm for episodic Markov decision processes with unknown losses and transition kernel. The paper proves that this algorithm achieves order sqrt(T) regret bounds when losses are chosen by an oblivious adversary and order log^2*(T) when losses are stochastic. This is the first algorithm to achieve such best of both works regret guarantees when transitions are unknown.
The key technical innovations are (1) the use of loss-shifting technique in the MDP setting, (2) running FTRL in phases where the empirical transition model is fixed per phase, (3) new biased loss estimators that combine importance weighting using upper occupancy measures with   additional bonus terms to promote exploration and (4) carefully tuned regularizers for FTRL.

**Limitations And Societal Impact:**

I am not aware of any potential negative societal impact of this work and the authors have sufficiently discussed the technical limitations of their work.

**Main Review:**

Significance: Best of both worlds results are desirable and have received a good amount of attention in the RL / bandit theory community. This paper significantly advances the state of the art in this area by showing the first such results in MDPs with unknown transitions. Besides the actual regret bound, I think the proof techniques in this paper will be useful in other settings as well. I particularly liked the simpler analysis using shifted losses compared to prior works.

Clarity: The paper is very well written, illustrating the main technical challenges and clearly explaining how they were overcome. I think the paper does a particularly good job at sketching the intuition in the main paper while providing the verbose arguments with all details in the appendix.

Originality: Many of the technologies used in this paper have been used elsewhere before, including loss shifting, using phases where the empirical model is kept fixed and adding a reward bias to account for uncertainty in transition estimates (e.g. in UCBVI algorithms). I still think that the combination and adaptation of these tools to the setting addressed here is technically highly nontrivial and sufficiently novel.

Quality: I checked several technical lemmas in the proofs and did not find any issues. The main arguments also make sense to me, but I did not go through all lemmas in the appendix carefully.

Minor comments and typos:
- One log(T) factor seems to come from the fact that the algorithm runs log(T) independent instances of FTRL. If one could avoid that, would that allow us to achieve a log(T) instead of log^2(T) bound or are there other places that contribute the additional log-factor?
- I understand that it is not the main focus of the paper to push the best results for either case, adversarial or stochastic, but I wonder whether the techniques developed in this paper can shed light on the additional sqrt(S) / S factor in the presented upper-bounds compared to the lower bounds when transitions are stochastic & unknown?
- Line 342: challenging -> challenges

**Time Spent Reviewing:**

3

---

> ### Author Response · Authors · 2021-08-10
> **Response to Reviewer o6kk**
>
> We thank the reviewer for the valuable feedback.
>
> 1\. *One $\log(T)$ factor seems to come from the fact that the algorithm runs $\log(T)$ independent instances of FTRL. If one could avoid that, would that allow us to achieve a $\log(T)$ instead of $\log^2(T)$ bound or are there other places that contribute the additional log-factor?*
>
> Indeed, the fact that we run $\log(T)$ independent instances of FTRL is the only source of the extra $\log(T)$ factor in our regret bound. It is unclear to us how to get rid of it though, since having a fixed transition in each epoch is critical to our loss shifting technique.
>
> 2\. *I understand that it is not the main focus of the paper to push the best results for either case, adversarial or stochastic, but I wonder whether the techniques developed in this paper can shed light on the additional $\sqrt{S}$ / $S$ factor in the presented upper-bounds compared to the lower bounds when transitions are stochastic & unknown?*
>
> Unfortunately, no (or at least not that we are aware of). Our algorithm uses the same Bernstein-style confidence interval as UOB-REPS (Jin et al., 2020), and we at best achieve the same bound as theirs for the adversarial environment, which is $\sqrt{S}$ away from the best lower bound unfortunately. We in fact conjecture that the upper bound is tight already, given the lower bound of the reward-free exploration setting (see Jin et al. "Reward-Free Exploration for Reinforcement Learning"), which shows unavoidable extra $S$ dependence when the learner has little knowledge on the future reward function of interest.

---

> > ### Comment · Reviewer_o6kk · 2021-08-22
> > **Thank you**
> >
> > Thank you for responding to my questions. This is very interesting work!

---

### Official Review · Reviewer_yvyL · 2021-07-27

**Rating:** 7
**Confidence:** 3

**Summary:**

This paper studies the problem of sequential learning in episodic MDP with full and bandit feedback and unknown dynamics of the MDP. The paper provides the first algorithm with provable performance guarantees, that simultaneously ensure polylogarithmic regret in the stochastic setting and the \sqrt(T) order regret bound in the adversarial setting. This is the first result of this kind, previous works only answer this question for a setting when the transitions are known. The paper proposes running the FTRL-type algorithm over the set of all valid occupancy measures from the estimated transition matrix. It ensures the exploration by subtracting the exploration bonus from the loss and it gives the importance weighted loss estimator by dividing the observed loss by the upper bound on the occupancy measure. The algorithm is computationally efficient. The secondary contribution of the paper is a more simple analysis of the variation of the same problem where the transitions are known.




**Limitations And Societal Impact:**

There is no potential negative societal impact.

**Main Review:**

The paper is very well-written and studies an interesting problem. It provides new techniques:

- The choice and use of a shifting function. It is a very nice solution.

- The idea to move the use of exploration bonus from the search of the occupancy measure for the confidence set of transitions
to the alternative solution of substructing the exploration bonus from the loss. This trick looks potentially useful in other applications.

Remarks:

- The statement of lines 167 - 170 and equation (7), and later reasoning on the same topic at lines 295-300 suffer from lack of clarity. Reference to a more detailed explanation in the appendix is needed and also reformulate a thought more clearly.

- (1) comes initially from best of both worlds analysis for bandits.

- The message of lines 231-234 is not very clear. The shifting function as well can be computed using \mathcal{P}_i.

- It is not very clear why the best of both world problem in the full information setting requires special attention. The stochastic regime with full information is not very challenging.

**Time Spent Reviewing:**

15

---

> ### Author Response · Authors · 2021-08-10
> **Response to Reviewer yvyL**
>
> We thank the reviewer for the valuable feedback
>
> 1\. *The statement of lines 167 - 170 and equation (7), and later reasoning on the same topic at lines 295-300 suffer from lack of clarity. Reference to a more detailed explanation in the appendix is needed and also reformulate a thought more clearly.*
>
> Thanks for the suggestion. We will clarify these in the next version.
>
> 2\. *Eq.(1) comes initially from best of both worlds analysis for bandits.*
>
> Indeed. We will add the reference.
>
> 3\. *The message of lines 231-234 is not very clear. The shifting function as well can be computed using $\mathcal{P}_i$.*
>
> Our shifting function is defined with respect to one fixed transition, so we assume what the reviewer means is that we perform FTRL over $\mathcal{P}_i$, whose solution corresponds to one transition in $\mathcal{P}_i$, and then we define the shifting function using this particular transition. This does not work as it does not ensure the invariant condition Eq. (5) (when $\bar{P}$ is changed to $\mathcal{P}_i$). Please also see Corollary A.1.2. for more details on this invariant.
>
> 4\. *It is not very clear why the best of both world problem in the full information setting requires special attention. The stochastic regime with full information is not very challenging.*
>
> We agree that the stochastic regime with full information is not very challenging. However, for the corrupted stochastic setting (which we cover), it is harder to identify the nature of world, and achieving similar guarantees as our Theorem 4.1.1. does not appear to be easy.

---

### Decision · Program_Chairs · 2021-09-27

**Decision:**

Accept (Oral)

**Comment:**

This paper has been very well-received by the reviewers, who all appreciated the significance of the contribution and the high quality of both the writing and the technical content. This is an excellent paper that clearly deserves to be published at the conference.